# Gapped phases with non-invertible symmetries: (1+1)d

Lakshya Bhardwaj, Lea E. Bottini, Daniel Pajer and Sakura Schäfer-Nameki

Mathematical Institute, University of Oxford, Andrew Wiles Building,
Woodstock Road, Oxford, OX2 6GG, UK

## Abstract

We propose a general framework to characterize gapped infra-red (IR) phases of theories with non-invertible (or categorical) symmetries. In this paper we focus on (1+1)d gapped phases with fusion category symmetries. The approach that we propose uses the Symmetry Topological Field Theory (SymTFT) as a key input: associated to a field theory in $d$ spacetime dimensions, the SymTFT lives in one dimension higher and admits a gapped boundary, which realizes the categorical symmetries. It also admits a second, physical, boundary, which is generically not gapped. Upon interval compactification of the SymTFT by colliding the gapped and physical boundaries, we regain the original theory. In this paper, we realize gapped symmetric phases by choosing the physical boundary to be a gapped boundary condition as well. This set-up provides computational power to determine the number of vacua, the symmetry breaking pattern, and the action of the symmetry on the vacua. The SymTFT also manifestly encodes the order parameters for these gapped phases, thus providing a generalized, categorical Landau paradigm for (1+1)d gapped phases. We find that for non-invertible symmetries the order parameters involve multiplets containing both untwisted and twisted sector local operators, and hence can be interpreted as mixtures of conventional and string order parameters [1]. We also observe that spontaneous breaking of non-invertible symmetries can lead to vacua that are physically distinguishable: unlike the standard symmetries described by groups, non-invertible symmetries can have different actions on different vacua of an irreducible gapped phase. This leads to the presence of relative Euler terms between physically distinct vacua. Along with the physical description of symmetric gapped phases, we also provide a mathematical one as pivotal 2-functors whose source 2-category is the delooping of the fusion category characterizing the symmetry and the target 2-category is the Euler completion of 2-vector spaces.

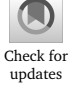

# 1  Introduction

Categorical symmetries have been a very active topic of research in both condensed matter and high-energy physics. Much recent progress [2–74] – or for recent overviews of this topic see [75–80] – has been on uncovering their structure, laying the foundations to study them in $d = D + 1$ spacetime dimensions with $D > 1$. The goal of this series of papers is to initiate a systematic analysis of the physical implications of such categorical symmetries. For a $d$ dimensional theory, the categorical symmetry $\mathcal{S}$ will be a fusion[1] $(d-1)$-category, which in first approximation corresponds to a set of topological defects of dimensions $0, \cdots, d-1$.

---

[1]Throughout this paper, we only work with finite symmetries, which is encoded in the fact that we only consider 'fusion' higher-categories. Moreover, we will only work with symmetry structures realizable in relativistic bosonic oriented theories.

Despite the many exciting developments – leading to some applications in particle physics – some truly path-breaking implications of categorical symmetries remain to be uncovered.

The goal of this series of papers is to explore such physical consequences and deliver on concrete physical implications that hinge on the existence of non-invertible symmetries. In [81] we proposed a program to classify the gapped phases with categorical symmetry $\mathcal{S}$. In this paper we apply this program to (1+1)d theories with fusion category symmetries. In [82] we will extend this to higher dimensions.

In (1+1)d, many aspects of gapped phases with fusion category symmetries have been discussed over the last few years. An axiomatic definition of 2d topological quantum field theories (TQFTs) with fusion category symmetries was provided in [83], and extended to include unoriented (i.e. time-reversal symmetric) TQFTs in [84]. Reference [85] discusses many interesting physical applications of categorical symmetries in 2d and some examples of TQFTs that can arise in the IR of systems with categorical symmetries. Their discussion includes the structure of vacua in the IR, but as will be discussed later in the paper, one can improve upon this description to provide full information of the TQFT, involving Euler terms and the action of the fusion category symmetry on the TQFT, along with the determination of order parameters for the associated gapped phase. Similar comments apply also to [86], which discusses many aspects and examples of gapped phases with categorical symmetries, involving a description of their boundaries. This reference also initiates a discussion of gapped phases with Tambara-Yamagami (TY) symmetries, which is a direction built upon in this paper by providing an explicit and detailed description of such phases. Another paper with interesting physical applications of categorical symmetries in 2d is [87], which also discusses many examples of 2d TQFTs with categorical symmetries. Defining data for correlation functions in 2d TQFTs with categorical symmetries were discussed in [88], which also discussed in detail a few examples of such TQFTs. There is overlap in our discussion of Ising SSB phase in section 6.3.1 with the discussion of regular TFT with Ising symmetry appearing in this reference. Lattice models for gapped phases with (non-anomalous) categorical symmetries are discussed in [89]. A paragraph in the introduction of [90] describes implicitly the structure of arbitrary unitary bosonic 2d TQFTs with categorical symmetries. This should be compared with the description appearing in section 3.1 of this paper, and the bulk of this paper can be viewed as explicitly fleshing out this structure with help from the tool of SymTFTs. Several examples of gapped phases with invertible but not necessarily abelian group symmetries have been studied in [91]. For other relevant works, focusing on classifying phases in 2d, the order parameters, or possible phase transitions, see [92–102].

The key innovations we propose going beyond the above works are:

- We use the SymTFT[2] to **classify $\mathcal{S}$-symmetric gapped phases**. We make a case that SymTFTs provide a systematic, comprehensive and computationally useful approach for performing the classification. We describe that the determination of an $\mathcal{S}$-symmetric (1+1)d gapped phase involves two steps:

    1. First of all, one needs to determine a non-symmetric 2d TQFT $\mathfrak{T}$ up to an overall Euler term, which in turn involves determining a set of vacua and relative Euler terms between the vacua.

    2. Furthermore, one has to specify how the $\mathcal{S}$ symmetry acts on the 2d TQFT $\mathfrak{T}$, which involves specifying which topological line defects of the 2d TQFT $\mathfrak{T}$ are implementing the symmetry $\mathcal{S}$.

---

[2]More details on the SymTFT are provided later in the introduction.

Mathematically, an $\mathcal{S}$-symmetric (1+1)d gapped phase is a 2-functor between two 2-categories (see the discussion around (32)). We describe how the SymTFT can be used to readily obtain all of this information regarding an arbitrary $\mathcal{S}$-symmetric gapped phase for any unitary fusion category $\mathcal{S}$. We apply these methods to explicitly classify (and determine the detailed structure of) gapped phases for all group symmetries with 't Hooft anomalies, for the non-invertible (i.e. not described by a group) $\mathsf{Rep}(S_3)$ symmetry, and for the non-invertible (and non-group-theoretical) Tambara-Yamagami $\mathsf{TY}(\mathbb{Z}_N)$ symmetries which includes as a special case $N = 2$ the Ising symmetry involving the Kramers-Wannier duality.

- We identify the (generalized) charges [58] of order parameters for arbitrary $\mathcal{S}$-symmetric gapped phases. In general, we find that such order parameters are mixtures of conventional order parameters (untwisted sector local operators) and string order parameters (twisted sector local operators, i.e. operators attached to topological lines), which are forced to coexist in a single irreducible multiplet due to the action of categorical symmetry on these local operators. This provides a completely systematic and computationally accessible **generalized version of the Landau paradigm** for gapped phases, in which the various gapped phases are distinguished by the generalized charges of the local operators that condense in that gapped phase. Extending this generalized Landau paradigm to include gapless phases and phase transitions is an important direction for future work. See [91, 103] for previous works along these lines.

- We uncover an interesting physical phenomenon tied closely to spontaneous breaking of non-invertible symmetries:

  **Spontaneous breaking of non-invertible symmetries can lead to physically distinguishable vacua, with different vacua having action of the symmetry on them.**

  This is in stark contrast to spontaneous breaking of invertible symmetries, i.e. conventional symmetries described by symmetry groups possibly with 't Hooft anomalies, where all the vacua participating in an irreducible gapped phase are acted upon in the same fashion by the symmetry.[3]

Let us note that the general idea of using SymTFTs to classify gapped phases has appeared in previous literature [103–105] (see also related works [106,107]). This work should be viewed as describing how to concretely carry out this idea for interesting non-invertible symmetries, and also describing the associated order parameters, along with an analysis of physical implications of spontaneous breaking of non-invertible symmetries, e.g. physical distinguishability of vacua.

**The Symmetry TFT.** Let us now briefly review the Symmetry TFT, which is the main tool used in this paper. When studying categorical symmetries, it is particularly useful to invoke the **symmetry topological field theory**, or SymTFT [108] (and for earlier works [109–111].

---

[3]Note that group symmetries with 't Hooft anomaly can give rise to vacua that can be physically distinguished, even if the symmetry acts on them in the same fashion. Such vacua are distinguished by the presence of different SPT phases for the unbroken group.

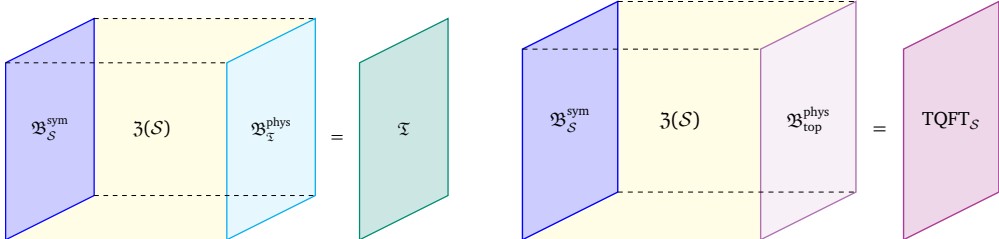

Figure 1: The basic SymTFT sandwich: (1) LHS: The $d$-dimensional theory $\mathfrak{T}$ on the RHS is constructed as the interval compactification of $d+1$-dimensional SymTFT $\mathfrak{Z}(\mathcal{S})$ on the LHS, with two boundary conditions. The gapped, i.e. topological, boundary $\mathfrak{B}_{\mathcal{S}}^{\text{sym}}$ is on the left and the physical, possibly non-topological, boundary $\mathfrak{B}_{\mathfrak{T}}^{\text{phys}}$ is on the right. (2) RHS: In this paper, we will focus on sandwich constructions for $\mathcal{S}$-symmetric TQFTs (denoted TQFT$_{\mathcal{S}}$) in which case the physical boundary $\mathfrak{B}_{\mathfrak{T}}^{\text{phys}}$ is also topological, $\mathfrak{B}_{\text{top}}^{\text{phys}}$ (though not necessarily the same as the symmetry boundary on the left).

Its utility is three-fold: it allows separating a field theory[4] from its symmetry and thus allows us to infer more general aspects of the symmetries themselves that do not depend on a specific QFT, it encodes all the **generalized charges** [58, 116] (i.e. local and extended operators that are charged under the categorical symmetry) and, finally, it provides a unified framework to study symmetries that are related by (generalized) gauging.

So far the SymTFT has been used to study symmetry-related questions of physical (not necessarily topological) QFTs, see the left hand side of figure 1. The SymTFT is a $(d + 1)$-dimensional TQFT $\mathfrak{Z}(\mathcal{S})$ for a $d$-dimensional theory $\mathfrak{T}$ with a categorical symmetry $\mathcal{S}$, which is topological and has two boundaries: a topological boundary $\mathfrak{B}_{\mathcal{S}}^{\text{sym}}$, which encodes the symmetries, and a not necessarily topological boundary $\mathfrak{B}_{\mathfrak{T}}^{\text{phys}}$, which depends on the QFT $\mathfrak{T}$. The topological defects of the SymTFT mathematically form the Drinfeld center $\mathcal{Z}(\mathcal{S})$ of the fusion higher-category $\mathcal{S}$.

Here we will specialize to gapped phases, which means that in the SymTFT description we are also imposing topological boundary conditions on $\mathfrak{B}^{\text{phys}}$, see the right hand side of figure 1. In this paper we flesh this proposal out in (1+1)d.

## 1.1 Proposal for classification of symmetric gapped phases

The goal of this work is to use the SymTFT formalism to classify gapped phases of theories with symmetry, starting with group-like symmetries (where we recover the known classifications) and later generalizing to categorical symmetries.

---

[4]Symmetry TFTs and sandwich constructions are continuum descriptions, and hence in their simplest form are only valid for QFTs. However, it is expected that there are lattice descriptions of SymTFTs and sandwich constructions, which would be applicable to quantum matter systems. In such a description, the result of the interval compactification would be lattice models such as (1+1)d anyonic chains of [112, 113] and their (2+1)d generalizations [54], and the role of SymTFTs would be played by the Levin-Wen string-net models [114] and their higher-dimensional generalizations. Such lattice versions of sandwich constructions will be presented in an upcoming work [115] which will be applied to obtain lattice models of $\mathcal{S}$-symmetric gapped phases discussed here. In any case, the framework of SymTFT will only be applied in this paper to study gapped phases captured by (Euclidean) topological quantum field theories, which admit a continuum description.

The procedure of classifying $\mathcal{S}$-symmetric $(1+1)$d gapped phases[5] by utilizing the SymTFT perspective requires the following steps:

1. SymTFT and its Topological Defects: Given a symmetry described by a unitary[6] fusion category $\mathcal{S}$, we identify the associated 3d SymTFT $\mathfrak{Z}(\mathcal{S})$ and the topological bulk lines living in $\mathfrak{Z}(\mathcal{S})$. These lines form a non-degenerate braided fusion category $\mathcal{Z}(\mathcal{S})$ known as the Drinfeld center of $\mathcal{S}$.

2. Lagrangian Algebras: We then classify all the irreducible topological boundary conditions of $\mathfrak{Z}(\mathcal{S})$, which are captured (modulo Euler terms) by the so-called Lagrangian algebras in $\mathcal{Z}(\mathcal{S})$ [119–121]. These characterize the topological lines in $\mathcal{Z}(\mathcal{S})$ which have Dirichlet boundary conditions.

3. Symmetry Boundary Condition: Since we are interested in classifying $\mathcal{S}$-symmetric gapped phases, we fix the symmetry boundary to be

$$\mathfrak{B}_{\mathcal{S}}^{\text{sym}} = \mathcal{A}_{\mathcal{S}}, \tag{1}$$

where $\mathcal{A}_{\mathcal{S}}$ is a Lagrangian algebra that realizes the symmetry $\mathcal{S}$ on the boundary. The Lagrangian specifies the set of topological lines in $\mathfrak{Z}(\mathcal{S})$, which end on the boundary $\mathfrak{B}_{\mathcal{S}}^{\text{sym}}$.

4. Physical Boundary Conditions: We then choose the physical boundary to also be a topological boundary condition specified by a Lagrangian algebra $\mathcal{A}_{\text{phys}}$

$$\mathfrak{B}^{\text{phys}} = \mathcal{A}_{\text{phys}}, \tag{2}$$

whereby the interval compactification of the SymTFT results in a 2d TQFT. Again, this specifies the lines in $\mathfrak{Z}(\mathcal{S})$ that can end on the physical boundary $\mathfrak{B}^{\text{phys}}$. By varying $\mathcal{A}_{\text{phys}}$, while keeping $\mathcal{A}_{\mathcal{S}}$ fixed, we move between different irreducible $\mathcal{S}$-symmetric phases.

5. Generalized Charges as Order Parameters: For an arbitrary $\mathcal{S}$-symmetric QFT $\mathfrak{T}$, the charges of local operators under $\mathcal{S}$ are captured by topological line defects of the SymTFT $\mathfrak{Z}(\mathcal{S})$ that can end on the corresponding physical boundary $\mathfrak{B}_{\mathfrak{T}}^{\text{phys}}$ [58]. Applying it to our topological context, the topological line defects that can end on a topological physical boundary $\mathfrak{B}^{\text{phys}} = \mathcal{A}_{\text{phys}}$ are precisely the lines that participate in the Lagrangian algebra $\mathcal{A}_{\text{phys}}$. In other words, the elements of $\mathcal{A}_{\text{phys}}$ describe charged local operators appearing in a (1+1)d $\mathcal{S}$-symmetric gapped phase, and hence describe the charges under $\mathcal{S}$ of order parameters for the gapped phase. This provides a **categorical or generalized Landau paradigm** describing gapped phases for an arbitrary categorical symmetry $\mathcal{S}$.

   There is a conceptually important difference between invertible and non-invertible order parameters: for non-invertible symmetries, twisted and untwisted sector operators can appear in the same multiplet. Thus order parameters can be both local operators (untwisted) or twisted sector or string-like order parameters!

6. Vacua: We employ this powerful SymTFT machinery to compute information about the resulting (1+1)d gapped phase. The number of vacua, for example, is easily determined by the number of the lines that can completely end on both boundaries, i.e. lines

---

[5]Throughout the paper, we refer to TQFTs in $d$ spacetime dimensions as being "$d$-dimensional" to emphasize the Euclidean nature of the spacetime involved. On the other hand, we refer to gapped phases in $d = D + 1$ spacetime dimensions as being $(D + 1)$-dimensional to emphasize the separate role played by the time direction.

[6]Note that this in particular means that we are provided a canonical spherical structure on the fusion[7] category $\mathcal{S}$, cf. proposition 9.5.1 of [118].

appearing in both Lagrangian algebras $\mathcal{A}_\mathcal{S}$ and $\mathcal{A}_{\text{phys}}$. We will depict this in terms of the following pared-down SymTFT picture:

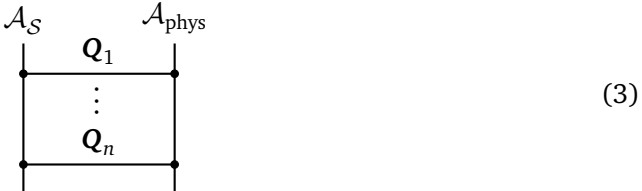

$$(3)$$

7. Action of the Symmetry $\mathcal{S}$: The action of the symmetry $\mathcal{S}$ on the (1+1)d gapped phase under discussion is specified by line operators $D_1^{(a)}$ of the associated 2d TQFT for each object $a \in \mathcal{S}$, i.e. which represent the fusion category $\mathcal{S}$ on the phases. The lines $D_1^{(a)}$ are determined as combinations of line operators of the 2d TQFT that act on the IR local operators realizing the order parameters according to their charges under $\mathcal{S}$.

8. Spontaneous Breaking of Non-Invertible Symmetries and Euler Terms: A notable phenomenon arises for (1+1)d gapped phases with non-invertible categorical symmetries, that does not occur for the usual group/invertible symmetries. The different vacua of a gapped phase with a non-invertible symmetry may be physically distinguishable as they may carry different Euler terms.[8] Such terms are encoded in the properties of interfaces (which are line defects in 2d) between different vacua, more precisely in the linking action of such interfaces on the vacua. An interface between two vacua with different Euler terms arises for a line operator $D_1^{(a)}$ implementing a non-invertible symmetry $a \in \mathcal{S}$ on the gapped phase. This means that the non-invertible symmetry is spontaneously broken as it relates different vacua. In other words, we learn that spontaneous breaking of non-invertible symmetries can lead to **physically distinguishable vacua**, with the symmetry acting differently on different vacua, and yet the whole set of vacua forms an irreducible representation of the symmetry.

This framework is applicable to **any (unitary) fusion category symmetry** $\mathcal{S}$. We will first revisit the group-symmetries described by a finite group $G$ possibly with a 't Hooft anomaly $\omega$, i.e. $\mathcal{S} = \text{Vec}_G^\omega$, and derive the expected structure of gapped phases in terms of spontaneous symmetry broken (SSB) phases and symmetry protected topological (SPT) phases from the SymTFT perspective.[9] The details are spelled out in section 3.

After this warm-up exercise, we then turn to the non-invertible group-theoretical symmetries,[10] i.e. fusion categories that are obtained by gauging a non-anomalous subgroup $H \leq G$. We focus on the $\text{Rep}(S_3)$ fusion category obtained by gauging the non-anomalous $G = S_3$ group symmetry. We determine all the $\text{Rep}(S_3)$-symmetric gapped phases, along with a description of the order parameters for each of the gapped phases. We compute in detail the number of vacua and the action of the non-invertible $\text{Rep}(S_3)$ symmetry on these vacua. There are a total of four irreducible gapped phases. In two of the gapped phases, the $\text{Rep}(S_3)$ symmetry is spontaneously broken but does not lead to physically distinguishable vacua, while in a third gapped phase, that we refer to as the $\text{Rep}(\boldsymbol{S_3})$ **SSB phase**, the action of the spontaneously broken $\text{Rep}(S_3)$ symmetry clearly treats the different vacua on a different footing, thus leading to

---

[8]It can happen in some instances that vacua coming from spontaneous breaking of an invertible symmetry are physically distinguishable, e.g. if $G$ and $H$ have a mixed anomaly the vacua from the SSB of $G$ can carry different SPT for $H$. However, the relative Euler terms in these cases are trivial.

[9]Strictly speaking SPT phases are already outside of the original Landau paradigm. These will be included as part of the generalized Landaau paradigm discussed in this paper, for which the order parameters are of string-type.

[10]Also sometimes called "non-intrinsically non-invertible" symmetries.


Table 1: Summary table of symmetries $\mathcal{S}$ considered in this paper, and their gapped IR phases, with some of the basic properties. The column order parameters indicates whether there is a standard untwisted sector local operator order parameter, or a twisted sector one (also known as string-type), i.e. an operator attached to a topological line. Mixed order parameter is a multiplet (under the action of symmetry $\mathcal{S}$) containing both untwisted and twisted sector local operators. Sometimes there are multiple order parameters having different properties: untwisted, string-type or mixed. SSB and SPT refer to spontaneous symmetry breaking and symmetry protected topological order, respectively.

| $\mathcal{S}$ | Phases | # Vacua | Order Parameters | Section |
|---|---|---|---|---|
| $\mathsf{Vec}_G$ | $G$ SSB | $\|G\|$ | untwisted | 4 |
| | $G$ SPT | 1 | string-type | 4 |
| | $(H, \beta)$ | $\|G:H\|$ | untwisted and string-type | 4 |
| $\mathsf{Rep}(S_3)$ | Trivial | 1 | string-type | 5, 5.3.1 |
| | $\mathbb{Z}_2$ SSB | 2 | string-type and mixed | 5, 5.3.2 |
| | $\mathsf{Rep}(S_3)/\mathbb{Z}_2$ SSB | 3 | string-type and mixed | 5, 5.3.3 |
| | $\mathsf{Rep}(S_3)$ SSB | 3 | string-type and mixed | 5, 5.3.4 |
| Ising | Ising SSB | 3 | untwisted and mixed | 6 |
| $\mathsf{TY}(\mathbb{Z}_N)$ | $(\mathbb{Z}_1, \mathbb{Z}_N)$ SSB | $N+1$ | untwisted and mixed | 7, 7.3.1 |
| | $(\mathbb{Z}_p, \mathbb{Z}_q)$ SSB | $p+q$ | untwisted, string-type and mixed | 7, 7.3.2 |

physically distinguishable vacua. Correspondingly, there are non-trivial relative Euler terms between the vacua in the $\mathsf{Rep}(S_3)$ SSB phase.

In the last two sections, we turn to intrinsically non-invertible, i.e. non-group-theoretical, symmetries: the Ising category, and its generalization to the Tambara-Yamagami categories $\mathsf{TY}(\mathbb{Z}_N)$ (when $N$ is not a perfect square). We again classify all possible gapped phases and describe the order parameters. We compute the number of vacua and the action of $\mathsf{TY}(\mathbb{Z}_N)$ on these vacua, along with the relative Euler terms forced by spontaneous breaking of $\mathsf{TY}(\mathbb{Z}_N)$ symmetries.

A summary of our results is shown in table 1, including the characterization of phases, number of vacua, and the type of order parameters.

## 2 (1+1)d gapped phases without symmetry

In this section, we review the classification of gapped phases in (1+1)d without imposing any symmetry constraints. This means that we identify two IR theories if they can be deformed into each other by arbitrary UV operators.[11] Later we will discuss gapped phases with symmetries, where we will identify two IR theories only if they can be deformed into each other by UV operators respecting that symmetry.

---

[11] It should be noted that we include IR phenomena in the infinite volume limit. If two IR theories can deformed into each other at finite volume, but cannot be deformed into each other at infinite volume, then the two IR theories lie in distinct phases.

## 2.1 Classification of gapped phases and TQFTs

The IR theories of interest are *unitary* 2d TQFTs, which were classified in [122]. According to this classification, a unitary 2d TQFT is described by a tuple of the form

$$(n|\lambda_1, \lambda_2, \cdots, \lambda_n), \qquad n \in \mathbb{Z}, \quad n > 0, \quad \lambda_i \in \mathbb{R}. \tag{4}$$

Such a 2d TQFT is not irreducible and decomposes into a collection of $n$ irreducible 2d TQFTs

$$(n|\lambda_1, \lambda_2, \cdots, \lambda_n) \cong (1|\lambda_1) \oplus (1|\lambda_2) \oplus \cdots \oplus (1|\lambda_n), \tag{5}$$

where the TQFT $(1|\lambda_i)$ has partition function

$$Z = e^{-\lambda_i \chi}, \tag{6}$$

where $\chi$ is the Euler characteristic of the 2d spacetime manifold under consideration. For this reason, we refer to the $\lambda_i$ in (4) as **Euler terms**.

The Euler terms can be smoothly deformed and fixed to be zero, from which we learn that (1+1)d gapped phases without symmetry are classified just by a positive integer $n$. We can recognize such a (1+1)d gapped phase as a collection of $n$ copies of the trivial gapped phase. Thus, there are no irreducible non-trivial gapped phases in (1+1)d. This just recovers the well-known fact that there is **no topological order in (1+1)d**.

Once we include symmetries into the game, it is sometimes not possible to tune all of the Euler terms to zero. For invertible symmetries, i.e. when the symmetry is described by a finite (0-form) symmetry group $G$, the Euler terms can all still be tuned to zero. However, as we will see, for non-invertible symmetries, the Euler terms cannot always be tuned to zero. It is for this reason that we do not pass to gapped phases in what follows, but study general 2d TQFTs.

Before moving on, let us note that the irreducible 2d TQFTs $(1|\lambda)$ are all invertible under the stacking operation and form the group $\mathbb{R}$

$$(1|\lambda) \boxtimes (1|\lambda') \cong (1|\lambda + \lambda'), \tag{7}$$

where $\boxtimes$ denotes stacking.

## 2.2 Properties of topological defects

We would like to capture the information (4) for a general 2d TQFT in terms of properties of topological defects of the TQFT.

### 2.2.1 Set of vacua

The positive integer $n$ in (4) captures the **number of vacua** of the TQFT. This can also be recognized as the dimension of the vector space $V_{S^1}$ that the TQFT assigns to a circle $S^1$. Physically, $V_{S^1}$ describes ground states of the IR theory on $S^1$. Using the state-operator correspondence, we can also regard $V_{S^1}$ as the vector space formed by topological local operators of the TQFT.

There is however a crucial difference between vacua[12] and arbitrary ground states on $S^1$. The distinction can be characterized in terms of a product operation on $V_{S^1}$, which converts it into an algebra. When $V_{S^1}$ is viewed as the space of local operators, the product operation corresponds to fusion/OPE of local operators

$$\underset{\mathcal{O}_1}{\bullet} \quad \underset{\mathcal{O}_2}{\bullet} \quad = \quad \underset{\mathcal{O}_1 \mathcal{O}_2}{\bullet}. \tag{8}$$

---

[12]Physically, vacua describe possible infinite volume limits of the $(1+1)$-dimensional system obeying cluster decomposition for correlation functions of local operators.

When $V_{S^1}$ is viewed as the space of states, the product operation corresponds to evaluating the TQFT on a pair of pants.

For a unitary 2d TQFT, the algebra structure on $V_{S^1}$ is such that there is a unique basis,

$$\{v_1, v_2, \cdots, v_n\} \in V_{S^1},\tag{9}$$

in which the product takes the form

$$v_i v_j = \delta_{ij} v_i,\tag{10}$$

where $\delta_{ij}$ is the Kronecker delta (see [123] for an explanation). The states $v_i$ are referred to as the **vacua** of the TQFT. In other words, the vacua describe a basis of idempotents in $V_{S^1}$. It should be particularly noted that vacua form a set of $n$ elements, while ground states on $S^1$ form an $n$-dimensional vector space.

### 2.2.2 Line operators

Let us now describe the topological line operators in a 2d TQFT associated to the data (4). Any such line defect can be expressed as a sum of irreducible unit line operators $\mathbf{1}_{ij}$ transitioning between vacua $i$ and $j$, with the vacuum $i$ lying on its left and the vacuum $j$ lying on its right

$$i \quad \Big\uparrow \quad j$$
$$\mathbf{1}_{ij}\tag{11}$$

For $j = i$, $\mathbf{1}_{ii}$ describes the identity line operator in the vacuum $i$. The full identity line operator $\mathbf{1}$ of the TQFT, after accounting for all the $n$ vacua, is

$$\mathbf{1} = \bigoplus_{i=1}^{n} \mathbf{1}_{ii}.\tag{12}$$

The fusion of these topological line operators is quite straightforward

$$\mathbf{1}_{ij} \otimes \mathbf{1}_{kl} = \delta_{jk} \mathbf{1}_{il},\tag{13}$$

which can be diagrammatically represented as

$$i \quad \Big\uparrow \quad j \quad \Big\uparrow \quad k \quad = \quad i \quad \Big\uparrow \quad k$$
$$\mathbf{1}_{ij} \qquad \mathbf{1}_{jk} \qquad\qquad\qquad \mathbf{1}_{ik}\tag{14}$$

## 2.3 Euler terms from linking action of line operators

The Euler terms $\lambda_i$ are encoded in the linking action of line operators $\mathbf{1}_{ij}$ on the vacua of the TQFT. The action takes the form

$$\mathbf{1}_{ij}: \quad v_i \to e^{-(\lambda_j - \lambda_i)} v_j,\tag{15}$$

which can be represented diagrammatically as

$$\left(\bigcirc i\,\Big\uparrow\right) \quad j \quad = \quad e^{-(\lambda_j - \lambda_i)} \quad j\,.$$
$$\mathbf{1}_{ij}\tag{16}$$

Note that these linking actions only capture the **relative Euler terms** between the vacua, but not the overall Euler term.

It is easy to deduce (16) as follows. An arbitrary linking action takes the form

$$\mathbf{1}_{ij}: \quad v_i \to \lambda_{ij} v_j\,, \qquad \lambda_{ij} \in \mathbb{R},\ \lambda_{ij} > 0\,. \tag{17}$$

The numbers $\lambda_{ij}$ have to respect the fusion of $\mathbf{1}_{ij}$ lines, and thus satisfy

$$\lambda_{ij} \lambda_{jk} = \lambda_{ik}\,. \tag{18}$$

Moreover, we have

$$\lambda_{ii} = 1\,, \tag{19}$$

since $\mathbf{1}_{ii}$ is the identity line in the vacuum $i$. This implies that

$$\lambda_{ji} = \lambda_{ij}^{-1}\,. \tag{20}$$

Now consider the partition function of the vacuum $v_i$ on a sphere $S^2$. Using (6), it can be expressed as

$$\bigcirc\, i \quad = \quad e^{-2\lambda_i}\,. \tag{21}$$

Let us create a line $\mathbf{1}_{ji}$ at the south pole of $S^2$, which gives

$$\mathbf{1}_{ji}\,\bigcirc\, \begin{matrix} i \\ j \end{matrix} \quad = \quad e^{-2\lambda_i} \lambda_{ji}^{-1}\,. \tag{22}$$

Sweeping the $\mathbf{1}_{ji}$ line across the sphere and contracting it at the north pole gives rise to an expression for the $S^2$ partition function of the vacua $j$ in terms of $\lambda_{ij}$

$$\bigcirc\, j \quad = \quad e^{-2\lambda_i} \lambda_{ji}^{-1} \lambda_{ij}\,. \tag{23}$$

Equating the RHS with $e^{-2\lambda_j}$ we obtain

$$\lambda_{ij} = e^{-(\lambda_j - \lambda_i)}\,, \tag{24}$$

as desired.

## 3 (1+1)d gapped phases with symmetry: General structure

Now we introduce a finite symmetry $\mathcal{S}$. Since we are in 2d, we take $\mathcal{S}$ to be described by a unitary fusion category.[13] The simple objects of this category describe different symmetries, which may be invertible or non-invertible. Each simple object is associated with a topological line operator in the theory.

We want to study gapped phases in (1+1)d carrying $\mathcal{S}$ symmetry. Such phases are obtained by identifying IR gapped theories up to UV deformations preserving $\mathcal{S}$. This is a finer equivalence relation than the one considered in the previous section, where we did not impose any symmetry: two $\mathcal{S}$-symmetric phases may become the same phase if one forgets about $\mathcal{S}$ and allows deformations that break the $\mathcal{S}$ symmetry.

---

[13]In this paper, we often drop the adjective 'unitary' for brevity.

### 3.1 2d TQFTs with symmetry

#### 3.1.1 Physical description: Choice of line operators implementing symmetries

Let us understand the structure of an $\mathcal{S}$-symmetric (1+1)d gapped phase in more detail. Such phases are obtained as deformation classes of $\mathcal{S}$-symmetric 2d TQFTs. The most fundamental data of an $\mathcal{S}$-symmetric 2d TQFT is an underlying 2d TQFT (without symmetry) specified by data of the form (4). The $\mathcal{S}$ symmetry is realized on this 2d TQFT by choosing line and local operators that reproduce all of the properties related to $\mathcal{S}$. That is, given a simple object $a$ of $\mathcal{S}$, we have a line operator[14]

$$D_1^{(a)} \cong \bigoplus_{i,j} n_{ij}^{(a)} \mathbf{1}_{ij}, \qquad n_{ij}^{(a)} \in \mathbb{Z}, \ n_{ij}^{(a)} \geq 0, \tag{25}$$

of the 2d TQFT. These line operators have to satisfy fusion rules of $\mathcal{S}$, i.e.

$$D_1^{(a)} \otimes D_1^{(b)} \cong \bigoplus_c N_c^{ab} D_1^{(c)}. \tag{26}$$

The morphisms of the symmetry $\mathcal{S}$ are represented by topological local operators living at the junctions of the above topological line operators. That is, given a morphism

$$\mu: \quad a \otimes b \to c, \tag{27}$$

where $a$, $b$ and $c$ are simple objects of $\mathcal{S}$, we have a topological local operator $D_0^{(\mu)}$ lying at the junction of $D_1^{(a)}$, $D_1^{(b)}$ and $D_1^{(c)}$

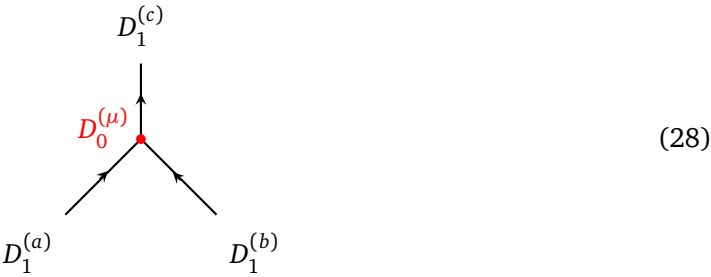

$$\tag{28}$$

The choices of these topological local operators have to be made such that the F-moves of $\mathcal{S}$ are respected. That is, the topological operators of the 2d TQFT satisfy:

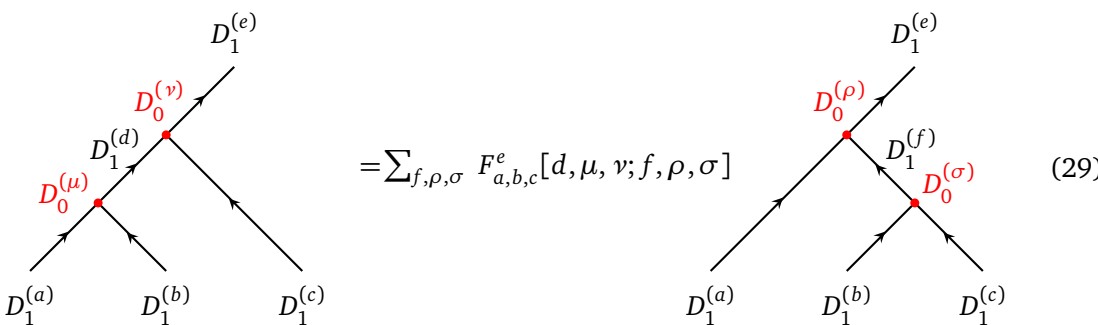

$$\tag{29}$$

mimicking the F-moves of the fusion category $\mathcal{S}$.

---

[14]We will denote topological defects of dimension $k$ by $D_k$.

Furthermore, the quantum dimensions of the objects of the category also have to be obeyed by the corresponding topological line operators

$$
\bigcirc_{D_1^{(a)}} \quad = \quad \dim(a)\,.
\tag{30}
$$

This requirement combined with the requirement (26) often enforces non-trivial linking actions for the unit lines $\mathbf{1}_{ij}$ participating in (25), implying the existence of non-trivial relative Euler terms.

### 3.1.2 Mathematical description: Classification of certain 2-functors

All of the above information is neatly captured mathematically as follows. Let $B\mathcal{S}$ be the 2-category obtained by delooping[15] the fusion category $\mathcal{S}$. Also, denote the 2-category formed by 2d TQFTs as[16]

$$
2\text{-Vec}^{\odot}\,.
\tag{31}
$$

$\mathcal{S}$-symmetric 2d TQFTs are then classified by pivotal 2-functors

$$
\varphi^{(2)}: \quad B\mathcal{S} \to 2\text{-Vec}^{\odot}\,.
\tag{32}
$$

Expanding the above description a little bit makes the connection to the above physical discussion clear. Such a 2-functor $\varphi^{(2)}$ assigns the sole object of $B\mathcal{S}$ to an object, namely a 2d TQFT $\mathfrak{T}$, of $2\text{-Vec}^{\odot}$. The 2-functor then descends to a pivotal monoidal functor

$$
\varphi^{(1)}: \quad \mathcal{S} \to \operatorname{End}(\mathfrak{T})\,,
\tag{33}
$$

where $\operatorname{End}(\mathfrak{T})$ is the endomorphism category of $\mathfrak{T}$, which is the multi-fusion category formed by topological line operators of the 2d TQFT $\mathfrak{T}$. In particular, we have

$$
D_1^{(a)} = \varphi^{(1)}(a)\,,
\tag{34}
$$

for any simple object $a \in \mathcal{S}$, and

$$
D_0^{(\mu)} = \varphi^{(1)}(\mu)\,,
\tag{35}
$$

for any morphism $\mu$ of the form (27). The fact that the lines $D_1^{(a)}$ obey the fusion rules and the local operators $D_0^{(\mu)}$ obey the F-symbols of $\mathcal{S}$ is encoded in the fact that $\varphi^{(1)}$ is a **monoidal** functor. Note that we also require $\varphi^{(1)}$ to be compatible with the **pivotal** structure, and in particular to preserve the quantum dimensions of objects of $\mathcal{S}$, as in (30).

## 3.2 From symmetric TQFTs to symmetric gapped phases

It is easy to pass from $\mathcal{S}$-symmetric 2d TQFTs to $\mathcal{S}$-symmetric (1+1)d gapped phases. Any irreducible $\mathcal{S}$-symmetric 2d TQFT $\mathfrak{T}_{\mathcal{S}}$ lies in a one-parameter family of irreducible $\mathcal{S}$-symmetric 2d TQFTs

$$
\mathfrak{T}_{\mathcal{S}} \in \left\{ \mathfrak{T}_{\mathcal{S}}^{\lambda} \,|\, \lambda \in \mathbb{R} \right\}\,,
\tag{36}
$$

---

[15]A delooping of a fusion $n$-category $\mathcal{C}$ is an $(n+1)$-category with a single object whose endomorphisms form the $n$-category $\mathcal{C}$.

[16]This 2-category is closely related to the 2-category $2\text{-Vec}$ of 2-vector spaces, whose objects are finite semisimple abelian categories. The operation $\odot$ is the Euler completion. See section 2.5 of [124] for more details.

such that the 2d TQFT $\mathfrak{T}^\lambda$ underlying $\mathfrak{T}_{\mathcal{S}}^\lambda$ is specified by the data

$$(n|\lambda + \lambda_1, \lambda + \lambda_2, \cdots, \lambda + \lambda_n), \tag{37}$$

if the data associated to the underlying 2d TQFT $\mathfrak{T}$ for $\mathfrak{T}_{\mathcal{S}}$ is

$$(n|\lambda_1, \lambda_2, \cdots, \lambda_n). \tag{38}$$

Said differently, we can express the 2d TQFT $\mathfrak{T}^\lambda$ as the stacking of $\mathfrak{T}$ with the invertible 2d TQFT $(1|\lambda)$

$$\mathfrak{T}^\lambda \cong (1|\lambda) \boxtimes \mathfrak{T}. \tag{39}$$

In fact, this $(1|\lambda)$ factor does not interact with the symmetry. That is, we can express $\mathfrak{T}_{\mathcal{S}}^\lambda$ as a stacking of $\mathfrak{T}_{\mathcal{S}}$ with $(1|\lambda)$

$$\mathfrak{T}_{\mathcal{S}}^\lambda \cong (1|\lambda) \boxtimes \mathfrak{T}_{\mathcal{S}}. \tag{40}$$

In other words, the **overall Euler term** is totally decoupled from $\mathcal{S}$ and can be tuned to zero. That is, the $\mathcal{S}$-symmetric irreducible 2d TQFTs $\mathfrak{T}_{\mathcal{S}}^\lambda$ for different $\lambda$ describe the same $\mathcal{S}$-symmetric irreducible (1+1)d gapped phase. Thus, only **relative Euler terms** between different vacua enter into the description of an irreducible $\mathcal{S}$-symmetric (1+1)d gapped phase.

Note that the relative Euler terms are precisely captured by the linking action of vacuum changing line operators $\mathbf{1}_{ij}$ of the underlying 2d TQFT. So all the information of an $\mathcal{S}$-symmetric (1+1)d gapped phase can be captured in the properties of topological defects of an $\mathcal{S}$-symmetric 2d TQFT.

## 3.3 SPT phases

Interesting classes of $\mathcal{S}$-symmetric (1+1)d gapped phases are symmetry protected topological (SPT) phases. The underlying (non-symmetric) gapped phase for an SPT phase is a trivial phase carrying a single vacuum, specified by $n = 1$.

The mathematical description (33) is quite useful for their characterization. The input that the underlying phase is trivial means that we can choose the underlying 2d TQFT to also be trivial (by removing the overall Euler term). That is, the image of the 2-functor $\varphi^{(2)}$ at the level of objects is

$$\mathsf{T} = \mathsf{Vec} \in 2\text{-}\mathsf{Vec}^{\odot}. \tag{41}$$

We have

$$\mathrm{End}(\mathsf{T}) = \mathrm{End}(\mathsf{Vec}) = \mathsf{Vec}, \tag{42}$$

and hence the $\mathcal{S}$-SPT phases are classified by monoidal functors

$$\varphi^{(1)}: \quad \mathcal{S} \to \mathsf{Vec}. \tag{43}$$

Such functors for which the target category is $\mathsf{Vec}$ are called **fiber functors**. Thus, $\mathcal{S}$-SPT phases are classified by fiber functors for the fusion category $\mathcal{S}$ (see also [86]).

**Non-anomalous $G$ symmetry.** We can now consider various examples. First of all, for an invertible symmetry described by a finite group $G$ without 't Hooft anomaly, the corresponding fusion category is

$$\mathcal{S} = \mathsf{Vec}_G, \tag{44}$$

and the fiber functors,

$$\mathsf{Vec}_G \to \mathsf{Vec}, \tag{45}$$

are known to be classified by group cohomology (see example 4.2 in [58] for a proof)

$$H^2(G, U(1)), \tag{46}$$

which is well-known classification of (1+1)d $G$-SPT phases.

**Anomalous $G$ symmetry.** If we now introduce a 't Hooft anomaly

$$0 \neq \omega \in H^3(G, U(1)), \tag{47}$$

the corresponding fusion category is

$$\mathcal{S} = \mathsf{Vec}_G^\omega, \tag{48}$$

with $\omega$ being a non-trivial associator. There are now no fiber functors of the form

$$\mathsf{Vec}_G^\omega \to \mathsf{Vec}, \tag{49}$$

as Vec has a trivial associator and it is not possible to represent a non-trivial associator on it. This reproduces the well-known result that a 't Hooft anomaly can be diagnosed as the absence of a trivial/SPT phase.

**Rep$(G)$ symmetry.** Consider the symmetry

$$\mathcal{S} = \mathsf{Rep}(G), \tag{50}$$

where $\mathsf{Rep}(G)$ is the fusion category formed by finite-dimensional representations of a finite group $G$. This is a non-invertible symmetry if $G$ is non-abelian. In this case, there is always at least one fiber functor

$$\varphi_{\mathrm{triv}}^{(1)}: \quad \mathsf{Rep}(G) \to \mathsf{Vec}, \tag{51}$$

which maps a $G$-representation $R$ as follows

$$R \mapsto V, \tag{52}$$

where $V$ is the underlying vector space of the representation $R$. We will refer to the resulting $\mathsf{Rep}(G)$-symmetric phase as the **trivial $\mathsf{Rep}(G)$-symmetric phase**.

It should be noted that there may be other SPT phases for a $\mathsf{Rep}(G)$ symmetry, even when it is non-invertible. Also, note that when $\mathsf{Rep}(G)$ is non-invertible, there isn't a canonical way to define a stacking operation on $\mathsf{Rep}(G)$ symmetric gapped phases. So the trivial phase given by $\varphi_{\mathrm{triv}}^{(1)}$ should not be regarded as a phase that is identity for the stacking operation.

**$\mathsf{TY}(\mathbb{Z}_N)$ symmetry.** It is also possible to have non-invertible symmetries that do not admit any SPT phases. Such symmetries are associated to fusion categories $\mathcal{S}$ not admitting any fiber functors. A class of examples of such symmetries is

$$\mathcal{S} = \mathsf{TY}(\mathbb{Z}_N), \tag{53}$$

where $\mathsf{TY}(\mathbb{Z}_N)$ is the Tambara-Yamagami fusion category based on the group $\mathbb{Z}_N$. We choose the bicharacter $\chi$ and the sign $\tau$ as in (329). This includes the Ising fusion category as we have

$$\mathsf{TY}(\mathbb{Z}_2) = \mathsf{Ising}. \tag{54}$$

The non-existence of the fiber functor will be made clear from the SymTFT approach that we will discuss below.

## 3.4 Order parameters

In this subsection, we describe a **Landau-type** characterization of $\mathcal{S}$-symmetric gapped phases. Different phases are characterized by different order parameters, which are UV local operators carrying non-trivial charges under the $\mathcal{S}$-symmetry that acquire a non-zero vacuum expectation value (vev).

### 3.4.1 Generalized charges

We begin by describing how a symmetry $\mathcal{S}$ acts on local operators. In general, local operators form irreducible multiplets under the action of $\mathcal{S}$. It should be noted that generally, a multiplet contains twisted sector local operators, i.e. local operators living at the ends of non-trivial topological line operators generating the symmetry $\mathcal{S}$. Although for invertible symmetries, an irreducible multiplet only carries untwisted sector local operators or twisted sector local operators, for non-invertible symmetries a single irreducible multiplet may carry both untwisted and twisted sector local operators.

The full action of $\mathcal{S}$ on an irreducible multiplet $\mathcal{M}$ is captured in a **generalized charge** or an $\mathcal{S}$-charge $Q$ carried by $\mathcal{M}$. Let us describe the information of $Q$ in more detail. The multiplet $\mathcal{M}$ involves vector spaces $V_{\mathcal{M}}^{(a)}$ of local operators living in $a$-twisted sectors[17] for each simple object $a \in \mathcal{S}$

$$ a \quad\longleftarrow\quad \bullet \quad \mathcal{O}_\mu^{(a)} \in V_{\mathcal{M}}^{(a)} \tag{55} $$

with $\mathcal{O}_\mu^{(a)}$ for different $\mu$ being a basis of $V_{\mathcal{M}}^{(a)}$. The action of $\mathcal{S}$ is described as

$$ = \sum_{c,\sigma,d,\nu,\rho} Q_{c,\sigma}^{a,\mu}(b)[d,\nu,\rho] \tag{56} $$

where the coefficients $Q_{c,\sigma}^{a,\mu}(b)[d,\nu,\rho]$ capture the information of the $\mathcal{S}$-charge $Q$. The generalized charges $Q$ are constrained by demanding that composing two symmetries and acting by the composition, is the same as first acting by the two symmetries individually and then composing them. Schematically, this is the requirement that the **rectangle identity** is satisfied, applying the action of $\mathcal{S}$ to different lines in the fusion diagram figure 2.

**Linking Action.** Often times we will need to understand the linking action of $\mathcal{S}$ lines on the local operators $\mathcal{O}_\mu^{(a)}$

$$ = \tag{57} $$

where we have chosen a basis for the quadrivalent junction such that it is a product of two trivalent junctions. This is often also referred to as the lasso action and gives rise to a concept called the tube algebra. See [26,85,125] for discussions of this in the physics literature.

---

[17]If some multiplet $\mathcal{M}$ does not include $b$-twisted sector local operators for some simple object $b \in \mathcal{S}$, then we have $V_{\mathcal{M}}^{(b)} = 0$.

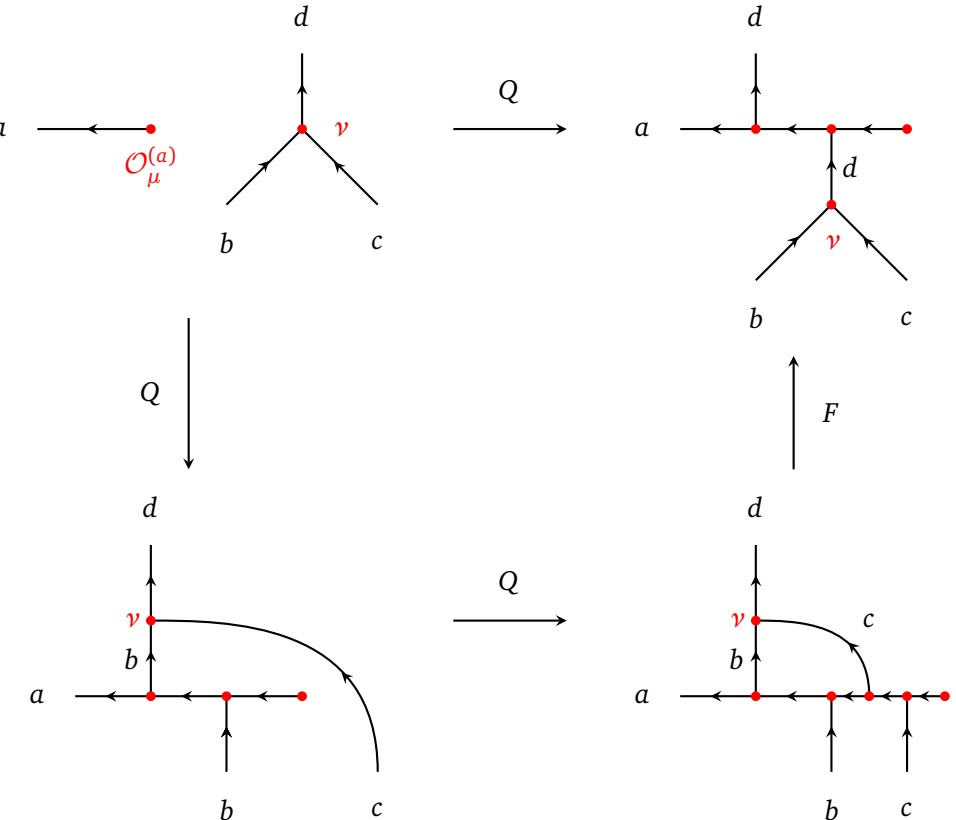

Figure 2: Rectangle identity: the square needs to commute for for all values of $\{a, b, c, \mu, \nu\}$. The arrows in the rectangle diagram denote the coefficients arising in each move. As is clear from the diagram, three of the moves involve $Q$-symbols, while one of them involves $F$-symbols. We have omitted the sum over internal topological lines, which can be reinstated straightforwardly using the definitions of $Q$ and $F$.

We can now use (56) to express the right hand side of (57) as

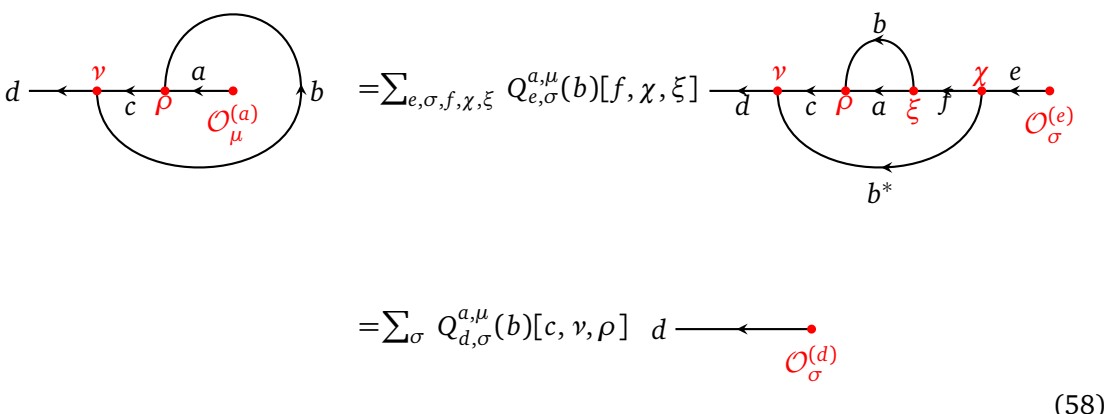

$$= \sum_{\sigma} Q^{a,\mu}_{d,\sigma}(b)[c, \nu, \rho] \quad d \xrightarrow{\hspace{2cm}} \mathcal{O}^{(d)}_{\sigma}$$

$$(58)$$

**Mathematical structure.** Mathematically, generalized charges for a symmetry described by a fusion category $\mathcal{S}$ are characterized by the **Drinfeld center** $\mathcal{Z}(\mathcal{S})$ of $\mathcal{S}$. The Drinfeld center $\mathcal{Z}(\mathcal{S})$ is a non-degenerate braided fusion category obtained from the data of the fusion category $\mathcal{S}$. An object of $\mathcal{Z}(\mathcal{S})$ is a pair

$$(Z, \beta), \tag{59}$$

where $Z$ is an object of $\mathcal{S}$, which we can express as a combination of simple objects $a$ of $\mathcal{S}$

$$Z \cong \bigoplus_a n^a a, \qquad n^a \in \mathbb{Z}, \ n^a \geq 0, \tag{60}$$

and $\beta$ is a collection of morphisms in $\mathcal{S}$, known as **half-braidings**.

A simple object $(Z, \beta)$ of $\mathcal{Z}(\mathcal{S})$ is a generalized charge for an irreducible multiplet $\mathcal{M}$ of local operators, such that a number $n^a$ appearing in the decomposition (60) of $Z$ is the dimension of the vector space of $a$-twisted sector local operators involved in $\mathcal{M}$, i.e.

$$n^a = \dim\left(V_{\mathcal{M}}^{(a)}\right). \tag{61}$$

The coefficients $Q_{c,\sigma}^{a,\mu}(b)[d, \nu, \rho]$ are captured in the information of half-braidings $\beta$. See section 4.3 of [58] for more details.

### 3.4.2 Landau characterization of symmetric gapped phases

Now consider an $\mathcal{S}$-symmetric UV theory $\mathfrak{T}_{\text{UV}}$ that is gapped and focus on the flow to an irreducible $\mathcal{S}$-symmetric gapped phase $\mathfrak{T}_{\text{IR}}$ in the IR containing $n$ vacua. A scalar local operator $\mathcal{O}_{\text{UV}}$ in an irreducible $\mathcal{S}$-multiplet $\mathcal{M}_{\text{UV}}$ of local operators of $\mathfrak{T}_{\text{UV}}$ may acquire a non-zero vacuum expectation value (vev), or in other words **condense**, in at least one of the $n$ vacua. If this happens, any other local operator in the same multiplet $\mathcal{M}_{\text{UV}}$ also acquires a non-zero vev[18] in at least one of the $n$ vacua. In such a situation, we say that the multiplet $\mathcal{M}_{\text{UV}}$ is an **order parameter** for the gapped phase $\mathfrak{T}_{\text{IR}}$.

Such a multiplet $\mathcal{M}_{\text{UV}}$ flows to a multiplet $\mathcal{M}_{\text{IR}}$ of topological local operators in the TQFT $\mathfrak{T}_{\text{IR}}$ carrying the same generalized charge $\boldsymbol{Q}$ under the $\mathcal{S}$ symmetry. In fact, an $\mathcal{S}$-symmetric (1+1)d gapped phase can be characterized by the spectrum of generalized charges realized within it. Given an irreducible generalized charge $\boldsymbol{Q}_A$, an $\mathcal{S}$-symmetric gapped phase $\mathfrak{T}_{\text{IR}}$ realizes $n_A \geq 0$ number of multiplets carrying the charge $\boldsymbol{Q}_A$, and we can characterize $\mathfrak{T}_{\text{IR}}$ in terms of the set of these non-negative integers $\{n_A\}$ for various values of $A$.

This provides a Landau-type classification of gapped phases with non-invertible symmetries. One can promote these ideas to a generalized Landau paradigm, with order parameters carrying generalized charges characterizing $\mathcal{S}$-symmetric gapped phases, and tunings of these order parameters characterizing second-order phase transitions between these gapped phases. The IR theories describing such phase transitions are $\mathcal{S}$-symmetric 2d conformal field theories (CFTs).

**Structure of an IR multiplet realizing a generalized charge.** Let $\mathcal{M}$ be an irreducible multiplet of local operators in an irreducible $\mathcal{S}$-symmetric (1+1)d gapped phase $\mathfrak{T}_{\text{IR}}$ carrying a generalized charge $\boldsymbol{Q}$ under $\mathcal{S}$. The various local operators $\mathcal{O}_\mu^{(a)} \in V_{\mathcal{M}}^{(a)}$ (55) participating in $\mathcal{M}$ are realized by topological local operators in $\mathfrak{T}_{\text{IR}}$. For fixed $a$, the operators $\mathcal{O}_\mu^{(a)}$ for different $\mu$ are linearly independent topological local operators in the space $\text{Hom}\left(\mathbf{1}, D_1^{(a)}\right)$ of topological local operators living at the end of topological line operator $D_1^{(a)}$ implementing the $a$ symmetry in $\mathfrak{T}_{\text{IR}}$. This can be expressed as

$$\text{Hom}\left(\mathbf{1}, D_1^{(a)}\right) \cong \bigoplus_i \mathbb{C}^{n_{ii}^{(a)}}, \tag{62}$$

---

[18]We only consider symmetries $\mathcal{S}$ that commute with spacetime symmetries. As such, the spacetime quantum numbers of all the local operators in an $\mathcal{S}$-multiplet are the same. Consequently, since $\mathcal{O}_{\text{UV}}$ is a scalar, all other local operators in $\mathcal{M}_{\text{UV}}$ are also scalars, and hence can acquire vevs.

using the coefficients $n_{ij}^{(a)}$ appearing in the description (25) of $D_1^{(a)}$. These operators have to satisfy the equation (56) for the action of line operators $D_1^{(b)}$. It should thus be clear that only very specific generalized charges can arise in a particular irreducible $\mathcal{S}$-symmetric gapped phase.

For the subspace $V_{\mathcal{M}}^{(\mathrm{id})}$ of untwisted sector local operators, we can express the participating operators $\mathcal{O}_{\mu}^{(\mathrm{id})}$ in terms of the vacua $v_i$

$$\mathcal{O}_{\mu}^{(\mathrm{id})} = \sum_i \alpha_{\mu}^i v_i, \qquad \alpha_{\mu}^i \in \mathbb{C}. \tag{63}$$

If $\alpha_{\mu}^i \neq 0$, then we can say that the operator $\mathcal{O}_{\mu}^{(\mathrm{id})}$ acquires a non-zero vev in the vacuum $v_i$.

### 3.4.3 Spontaneous breaking of non-invertible symmetries

Similar to the case of invertible group-like symmetries, we have a notion of spontaneous breaking for general non-invertible symmetries. There are two equivalent ways to characterize spontaneous breaking of a symmetry $a \in \mathcal{S}$ in a vacuum $v_i$ of an irreducible $\mathcal{S}$-symmetric (1+1)d gapped phase $\mathfrak{T}_{\mathrm{IR}}$:

- The line operator $D_1^{(a)}$ (25) implementing the symmetry $a$ has the property that the coefficient

$$n_{ij}^{(a)} \geq 1, \tag{64}$$

for some $j \neq i$. That is, the symmetry $a$ is spontaneously broken in vacuum $v_i$ if its (linking) action on $v_i$ produces another vacuum $v_j$.

- There exists an untwisted local operator $\mathcal{O}$ charged non-trivially under $a$ that acquires a non-zero vev in the vacuum $v_i$ (see above).

An interesting physical phenomena may occur when non-invertible symmetries are spontaneously broken. As is well-known, all the vacua of an irreducible gapped phase for an invertible symmetry (possibly with a 't Hooft anomaly) are acted upon equally by the broken symmetries, and it is impossible to physically differentiate between any two vacua involved in the gapped phase solely by looking at their symmetry transformation properties. This no longer holds true for irreducible gapped phases with non-invertible symmetries. In such a gapped phase, the action of the non-invertible symmetry may act differently on different vacua, thus physically distinguishing them. A necessary requirement for such physically distinguishable vacua to arise is spontaneous breaking of some non-invertible symmetry. However, it should be noted that this is not sufficient: one may obtain vacua with the same actions of symmetry even if a non-invertible symmetry is spontaneously broken. This will be explicitly shown later in examples. The physical distinction between vacua can be equivalently characterized in terms of the presence of relative Euler terms between them, which are enforced by the spontaneous breaking of some non-invertible symmetry.

## 3.5 $\mathcal{S}$-symmetric phases from symmetry TFTs

In this section, we have so far discussed various objects of interest in the study of $\mathcal{S}$-symmetric (1+1)d gapped phases, but we are yet to provide a computational handle on these objects. In this subsection, we describe how the Symmetry TFT (SymTFT) can be used to carry out these computations. Symmetry TFTs can also be used to obtain a full classification of all the possible $\mathcal{S}$-symmetric gapped phases and generalized charges.

**Review: The SymTFT setup.** We begin by reviewing the setup of SymTFTs in full generality. Consider a theory $\mathfrak{T}$ in $d$ spacetime dimensions carrying $\mathcal{S}$ symmetry, where $\mathcal{S}$ is some fusion $(d-1)$-category. Then, we can express $\mathfrak{T}$ as an interval compactification, as shown in figure 1.

The main components of these constructions are as follows:

- $\mathfrak{Z}(\mathcal{S})$ is a TQFT (without symmetry) in $(d+1)$ spacetime dimensions.

- $\mathfrak{B}_{\mathcal{S}}^{\text{sym}}$ is a topological boundary condition of $\mathfrak{Z}(\mathcal{S})$ such that topological defects living on $\mathfrak{B}_{\mathcal{S}}^{\text{sym}}$ (and unattached to topological defects of the bulk theory $\mathfrak{Z}(\mathcal{S})$) form the fusion $(d-1)$-category $\mathcal{S}$.

- $\mathfrak{B}_{\mathfrak{T}}^{\text{phys}}$ is a boundary condition of $\mathfrak{Z}(\mathcal{S})$ capturing the information of the theory $\mathfrak{T}$. This boundary is (non-)topological if the theory $\mathfrak{T}$ is (non-)topological.

This setup is also known as the **sandwich construction**.

As noted above, if $\mathfrak{T}$ is a TQFT, then $\mathfrak{B}_{\mathfrak{T}}^{\text{phys}}$ is a topological boundary condition of $\mathfrak{Z}(\mathcal{S})$. As such, **irreducible $\mathcal{S}$-symmetric TQFTs are classified by irreducible topological boundary conditions of $\mathfrak{Z}(\mathcal{S})$**. It is this correspondence that we wish to exploit further to understand $\mathcal{S}$-symmetric gapped phases in (1+1)d.

We restrict our attention to spacetime dimension $d = 1+1$ from this point on, as that is the case of interest in this paper. However, the extension to higher dimensions fits equally well into this framework and will be discussed in a follow-up paper [82]. For $d = 1+1$, the SymTFTs are 3d TQFTs which are obtained by applying the **Turaev-Viro-Barrett-Westbury construction** with input fusion category $\mathcal{S}$. The key information relevant to our analysis is that of topological line operators of $\mathfrak{Z}(\mathcal{S})$. These line operators form precisely the Drinfeld center $\mathcal{Z}(\mathcal{S})$ of $\mathcal{S}$ which, as discussed in previous subsection, captures generalized charges of multiplets formed under the action of $\mathcal{S}$ by local operators in an $\mathcal{S}$-symmetric 2d theory.

**Topological boundary conditions of SymTFT and Lagrangian algebras.** The irreducible topological boundary conditions of the SymTFT $\mathfrak{Z}(\mathcal{S})$ are captured (modulo Euler terms) by Lagrangian algebras in the Drinfeld Center $\mathcal{Z}(\mathcal{S})$ [119, 121]. A Lagrangian algebra $\mathcal{A}$ can be expressed as

$$\mathcal{A} = \bigoplus_A n_A \mathbf{Q}_A, \qquad n_A \in \mathbb{Z}_{\geq 0},\tag{65}$$

where $\mathbf{Q}_A$ are simple objects of $\mathcal{Z}(\mathcal{S})$, and the algebra has to satisfy the dimension constraint

$$\dim(\mathcal{A}) := \sum_{A \in \mathcal{A}} n_A \dim(\mathbf{Q}_A) = \dim^2(\mathcal{S}),\tag{66}$$

where the dimension of a line is its quantum dimension, and the dimension of a category is its total quantum dimension, which can be expressed as

$$\dim(\mathcal{S}) = \sqrt{\sum_a \dim^2(a)},\tag{67}$$

where the sum is over all simple objects $a$ of $\mathcal{S}$. Along with the above condition on quantum dimensions, $\mathcal{A}$ has to satisfy further constraints in order to be a Lagrangian algebra. In particular

$$n_A n_B \leq \sum_C N_{AB}^C n_C,\tag{68}$$

where $N_{AB}^C$ is the coefficient specifying the fusion $A \otimes B \to C$ in $\mathcal{S}$. Let us note here another useful condition, which is that all $\mathbf{Q}_A$ participating with non-zero coefficients in the Lagrangian algebra $\mathcal{A}$ must be **bosons**, i.e. their spins must be trivial. For more details see [126].

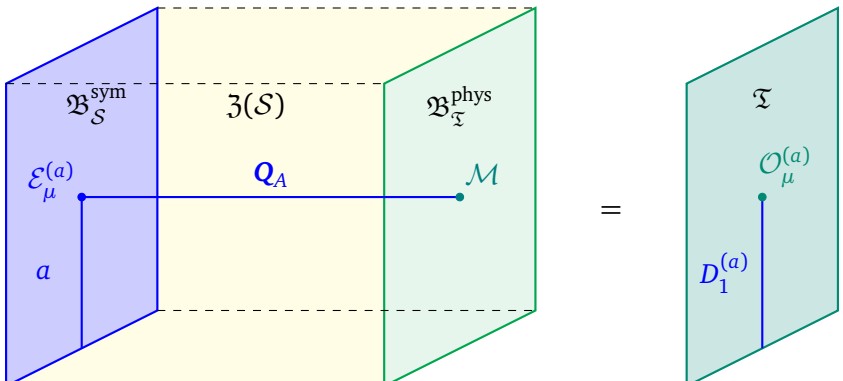

Figure 3: The sandwich construction of a multiplet $\mathcal{M}$ of local operators carrying generalized charge $Q_A$ in an $\mathcal{S}$-symmetric 2d theory $\mathfrak{T}$ involves interval compactification of the corresponding bulk topological line $Q_A$ ending along the physical boundary $\mathfrak{B}^{\text{phys}}_{\mathfrak{T}}$ along a local operator $\mathcal{M}$. Along the symmetry boundary $\mathfrak{B}^{\text{sym}}_{\mathcal{S}}$, the bulk line $Q_A$ can be attached to a boundary line $a$ along local operators $\mathcal{E}^{(a)}_\mu$. After the interval compactification, these local operators $\mathcal{E}^{(a)}_\mu$ become $a$-twisted sector local operators $\mathcal{O}^{(a)}_\mu$ in the multiplet $\mathcal{M}$, living at the end of topological line $D^{(a)}_1$ implementing $a$ symmetry in $\mathfrak{T}$.

The sandwich construction of an irreducible $\mathcal{S}$-symmetric (1+1)d gapped phase involves two Lagrangian algebras. The first one denoted $\mathcal{A}_{\mathcal{S}}$ describes the symmetry boundary $\mathfrak{B}^{\text{sym}}_{\mathcal{S}}$, while the second denote $\mathcal{A}_{\text{phys}}$ describes the physical boundary $\mathfrak{B}^{\text{phys}}$. Often, for convenience of notation, we do not differentiate between a Lagrangian algebra and the topological boundary associated to it, in particular making the following identifications

$$\mathfrak{B}^{\text{sym}}_{\mathcal{S}} = \mathcal{A}_{\mathcal{S}}, \qquad \mathfrak{B}^{\text{phys}} = \mathcal{A}_{\text{phys}}. \tag{69}$$

While discussing different $\mathcal{S}$-symmetric gapped phases, we keep $\mathcal{A}_{\mathcal{S}}$ fixed and vary $\mathcal{A}_{\text{phys}}$, and thus every $\mathcal{S}$-symmetric gapped phase is associated to a Lagrangian algebra $\mathcal{A}_{\text{phys}}$.[19]

**Order parameters from SymTFT.** The information of a Lagrangian algebra $\mathcal{A}$ encodes within it the generalized charges of the order parameters associated to the $\mathcal{S}$-symmetric gapped phase described by $\mathcal{A}$.

In order to describe these, recall first that the generalized charges for $\mathcal{S}$ are captured by line defects of $\mathfrak{Z}(\mathcal{S})$ [58]. This is the reason we chose above the same notation $Q_A$ for both irreducible generalized charges of $\mathcal{S}$ and simple line defects of $\mathfrak{Z}(\mathcal{S})$. The relationship between the two becomes clear by compactifying the line operator $Q_A$ along the interval as in figure 3.

A multiplet $\mathcal{M}$ transforming in the generalized charge $Q_A$ is realized as a single local operator on the physical boundary $\mathfrak{B}^{\text{phys}}_{\mathfrak{T}}$ that is attached to the corresponding bulk topological line $Q_A$. There are various possible partial ends $\mathcal{E}^{(a)}_\mu$ of $Q_A$ along a boundary topological line $a \in \mathcal{S}$, which after interval compactification descend to the operators $\mathcal{O}^{(a)}_\mu$ living in the multiplet $\mathcal{M}$ that are in $a$-twisted sector for the symmetry $\mathcal{S}$, or in other words are realized at the end of topological line operator $D^{(a)}_1$ implementing the symmetry $a$ in the $\mathcal{S}$-symmetric theory $\mathfrak{T}$.

---

[19]However, it should be noted that this correspondence is not canonical. We can have two different topological boundary conditions $\mathcal{A}_{\mathcal{S}}$ and $\mathcal{A}'_{\mathcal{S}}$ of $\mathfrak{Z}(\mathcal{S})$ whose topological defects form the same fusion category $\mathcal{S}$. Choosing $\mathcal{A}'_{\mathcal{S}}$ as the symmetry boundary provides a different one-to-one correspondence between Lagrangian algebras and $\mathcal{S}$-symmetric gapped phases. An example is provided by boundaries of the SymTFT for $S_3$ symmetry, as discussed around (193).

The action of $\mathcal{S}$ on $\mathcal{O}_\mu^{(a)}$ is obtained from the action of $\mathcal{S}$ on the partial ends $\mathcal{E}_\mu^{(a)}$, which is a property entirely of the symmetry boundary $\mathfrak{B}_\mathcal{S}^{\text{sym}}$.

The bulk topological line operators that can completely end on a topological boundary are precisely the ones entering the Lagrangian algebra corresponding to the boundary. Moreover, the number of linearly independent ends of a bulk line is given by its coefficient in the Lagrangian algebra. Applying it to the physical boundary, let us say the decomposition of $\mathcal{A}_{\text{phys}}$ is

$$\mathcal{A}^{\text{phys}} = \bigoplus_A n_A^{\text{phys}} \mathbf{Q}_A. \tag{70}$$

Then, the generalized charges carried by local operators in the $\mathcal{S}$-symmetric gapped phase $\mathfrak{T}(\mathcal{A}^{\text{phys}})$ corresponding to $\mathcal{A}^{\text{phys}}$ are $\mathbf{Q}_A$ with $n_A^{\text{phys}} \neq 0$. These are then the generalized charges of the order parameters for the $\mathcal{S}$-symmetric gapped phase $\mathfrak{T}(\mathcal{A}^{\text{phys}})$. Moreover, each order parameter has $n_A^{\text{phys}}$ possible IR images (i.e. IR multiplets with the same generalized charge $\mathbf{Q}_A$) that it can flow to.

**Number of vacua from the SymTFT.** We can determine all of the data associated to an irreducible $\mathcal{S}$-symmetric (1+1)d gapped phase from its SymTFT construction. To begin with, the number of vacua involved in the gapped phase $\mathfrak{T}(\mathcal{A}_{\text{phys}})$ is easily determined as follows.

Let $V_{S^1}$ be the space of (untwisted) topological local operators of the $\mathfrak{T}(\mathcal{A}_{\text{phys}})$. A basis for this space is obtained by compactifying bulk line operators $\mathbf{Q}_A$ such that they completely end on both boundaries $\mathfrak{B}_\mathcal{S}^{\text{sym}}$ and $\mathfrak{B}_\mathfrak{T}^{\text{phys}}$, without involving any non-trivial boundary topological line operators. This basis is in general different from the basis provided by the vacua. However, we can determine the number of vacua by counting the number of local operators produced in this way. If the decomposition of the Lagrangian algebra $\mathcal{A}_\mathcal{S}$ is

$$\mathcal{A}^{\text{sym}} = \bigoplus_A n_A^{\text{sym}} \mathbf{Q}_A, \tag{71}$$

then the number of vacua $n$ is equal to

$$n = \sum_A n_A^{\text{sym}} n_A^{\text{phys}}. \tag{72}$$

Every simple bulk line $\mathbf{Q}_A$ gives rise to a subspace $V_A \subseteq V_{S^1}$ whose dimension is $n_A^{\text{sym}} n_A^{\text{phys}}$. We will depict these configurations by a simplified SymTFT picture as follows

$$
\begin{array}{cc}
\mathcal{A}^{\text{sym}} & \mathcal{A}_{\text{phys}} \\
\end{array}
\begin{array}{c}
\mathbf{Q}_1 \\
\vdots \\
\mathbf{Q}_n
\end{array}
\tag{73}
$$

**Relative Euler terms from SymTFT.** To determine the relative Euler terms, we need to understand the linking action of symmetry line operators on the vacua, which descends from the action of topological lines living on $\mathfrak{B}_\mathcal{S}^{\text{sym}}$ on the ends of bulk lines $\mathbf{Q}_A$ along $\mathfrak{B}_\mathcal{S}^{\text{sym}}$.

To do this, we determine the vacua as elements of $V_{S^1}$, which requires us to identify the algebra structure on $V_{S^1}$. Since we are using the SymTFT picture, the natural basis to use is provided by local operators in vector spaces $V_A$ descending from simple bulk lines. In the text, we will combine various consistency conditions to bootstrap the product,

$$\mathcal{O}_i \mathcal{O}_j = \sum_k \alpha_{ij}^k \mathcal{O}_k, \tag{74}$$

on such local operators:

- The product on local operators needs to be consistent with the product on bulk lines.

- The product needs to be consistent with the action of topological line operators living on both boundaries $\mathfrak{B}^{\text{sym}}_{\mathcal{S}}$ and $\mathfrak{B}^{\text{phys}}$.

- The product needs to be associative.

- Finally, we can rescale the local operators to put the product in a nice desired form.

Once the vacua are determined, the linking action of $\mathcal{S}$ on them is readily obtained, from which we can read the decomposition (25) of symmetry line operators in terms of irreducible unit line operators $\mathbf{1}_{ij}$. In particular, implementing the quantum dimension of the symmetry line operators forces some of the unit line operators $\mathbf{1}_{ij}$ to have non-trivial linking actions on vacua, which precisely capture the relative Euler terms.

## 4 Revisiting (1+1)d gapped phases with group symmetries

In this section and in the next sections, we implement the procedure discussed in the previous section to understand and classify (1+1)d gapped phases with symmetry. In this section, we revisit the classic case of invertible symmetries described by a finite (0-form) group $G$, possibly with a 't Hooft anomaly

$$\omega \in H^3(G, U(1)).\tag{75}$$

We hope that putting this very well-explored case into the general framework will familiarize the reader with the SymTFT approach, and thus make the subsequent generalizations to non-invertible symmetric phases more transparent.

Before diving into the details, let us connect to the categorical notation for such symmetries. The fusion category associated to such an invertible symmetry and 't Hooft anomaly (75) is

$$\mathcal{S} = \mathsf{Vec}^{\omega}_G,\tag{76}$$

which is the category formed by $G$-graded (complex, finite-dimensional) vector spaces, with their associator provided by $\omega$. The simple objects of $\mathcal{S}$ are labeled by group elements

$$g, \qquad g \in G,\tag{77}$$

with fusion given by group multiplication

$$g_1 \otimes g_2 \cong g_1 g_2,\tag{78}$$

and $F$-symbols given by $\omega$ as

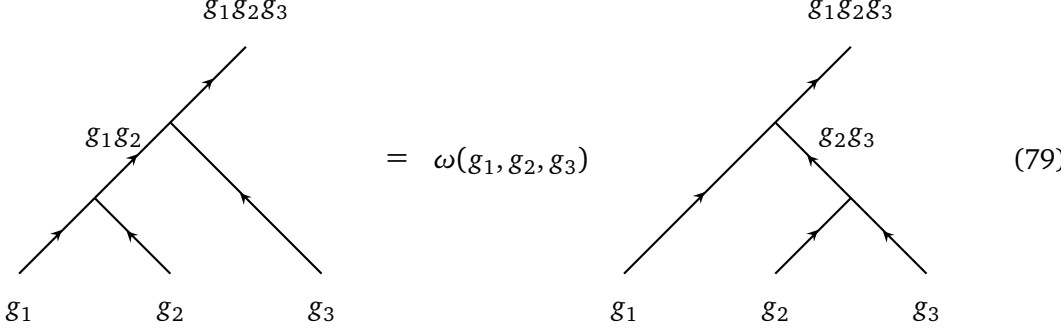

$$\tag{79}$$

It should be noted that the quantum dimensions of all simple objects are trivial

$$\dim(g) = 1, \qquad \forall\, g \in G.\tag{80}$$

## 4.1 Classification: SSB and SPT

**Expected classification.** The expected classification of such phases is that they are mixtures of spontaneous symmetry breaking (SSB) and symmetry protected topological (SPT) phases. This will indeed be reproduced by the SymTFT analysis below, and hence the invertible symmetries serve as a cross-check for our general proposal.

In more detail, such irreducible (1+1)d gapped phases are classified by two pieces of data:

1. A subgroup $H \leq G$, on which the anomaly trivializes

$$\omega|_H = 0 \in H^3(H, U(1)). \tag{81}$$

   Physically, $H$ is the subgroup left spontaneously unbroken in one of the vacua $v$. The total number $n$ of vacua is the number of $H$ cosets in $G$.

2. An element

$$\beta \in H^2(H, U(1)), \tag{82}$$

   which describes the SPT phase carried by $H$ in the vacuum $v$.

The vacua can be parameterized by $H$-cosets. An element $g \in G$ sends the vacuum $v$ to the coset containing element $g$, and thus $g$ is spontaneously broken in $v$ if $g \notin H$. The spontaneously unbroken symmetry in a vacuum corresponding to coset $gH$ is the subgroup

$$gHg^{-1} \subseteq G, \tag{83}$$

which is isomorphic to $H$ but is in general a different subgroup of $G$. The SPT phase for $gHg^{-1}$ in this vacuum is obtained from $\beta$ by applying the isomorphism map.

The line operator implementing $g$ symmetry in the IR is expressed as

$$D_1^{(g)} = \bigoplus_i \mathbf{1}_{ig(i)}, \tag{84}$$

where the sum is over all vacua $i$ and $g(i)$ is the vacuum obtained by acting $g$ on the $i$-th vacuum.

The above expression for $D_1^{(g)}$ implies that there are no relative Euler terms in the IR theory. If the linking action for any unit line is non-trivial

$$\mathbf{1}_{ij}: \quad v_i \rightarrow \lambda_{ij} v_j, \qquad \lambda_{ij} \neq 1, \tag{85}$$

then we cannot satisfy the condition (80), which now translates to the requirement that the linking action of $D_1^{(g)}$ leaves invariant the sum of all vacua

$$D_1^{(g)}: \quad 1 = \sum_i v_i \rightarrow \sum_i v_i = 1, \tag{86}$$

where 1 denotes the identity local operator in the IR.

**Classification via SymTFT.** We can easily reproduce the above classification via the SymTFT approach. The Turaev-Viro construction based on $\mathrm{Vec}_G^\omega$ leads to the 3d **Dijkgraaf-Witten (DW) gauge theory** with gauge group $G$ and DW twist $\omega$. This is thus the SymTFT $\mathcal{Z}\left(\mathrm{Vec}_G^\omega\right)$. Note that this 3d theory can be obtained by beginning with the trivial 3d theory, and gauging the $G$ symmetry that acts trivially on the theory, with discrete torsion $\omega$.

The symmetry boundary $\mathcal{B}_{\mathrm{Vec}_G^\omega}^{\mathrm{sym}}$ carrying the $\omega$-anomalous $G$-symmetry is the **Dirichlet** boundary condition for the $G$ gauge fields in the 3d bulk. Indeed, on a Dirichlet boundary, the

$G$ gauge symmetry of the 3d bulk becomes a $G$ global symmetry, and the twist $\omega$ appears as 't Hooft anomaly for this $G$ global symmetry. Thus, the Dirichlet boundary condition carries the $\text{Vec}_G^\omega$ symmetry and hence can be used as the symmetry boundary $\mathcal{B}^{\text{sym}}_{\text{Vec}_G^\omega}$.

A general fact of every 3d SymTFT $\mathfrak{Z}(\mathcal{S})$ is that any arbitrary topological boundary condition of $\mathfrak{Z}(\mathcal{S})$ can be obtained by gauging the $\mathcal{S}$ symmetry of $\mathfrak{B}^{\text{sym}}_{\mathcal{S}}$ (this goes via the correspondence between module categories over $\mathcal{S}$ and Lagrangian algebras in $\mathcal{Z}(\mathcal{S})$, see [120, 127]). Let us apply this fact to the case (76). We can only gauge a subgroup $H$ of the $G$ symmetry on which the 't Hooft anomaly trivializes (81), and the $H$ gauging involves a choice of discrete torsion, which is an element $\beta$ as in (82).

Thus, the topological boundary conditions of the SymTFT $\mathfrak{Z}\left(\text{Vec}_G^\omega\right)$ are also classified by the same data entering the expected classification of (1+1)d gapped phases with $G$ symmetry carrying 't Hooft anomaly $\omega$. According to our general approach, topological boundary conditions of a 3d SymTFT $\mathfrak{Z}(\mathcal{S})$ classify all (1+1)d gapped phases with $\mathcal{S}$ symmetry. Thus, the SymTFT classification matches the expected classification. It is also possible to reproduce the detailed structure of vacua and the action of $G$-symmetry on them using the SymTFT approach. We do it on a case-by-case basis in the examples discussed later in this section.

**$G$ SSB phase.**   Let us discuss two special types of phases in more detail. The first one corresponds to

$$H = 1, \qquad \beta = 0, \tag{87}$$

in which all symmetries $g \in G$ are spontaneously broken in all vacua. There are a total of $|G|$ number of vacua which can be parameterized by group elements. The $g$ line is identified as

$$D_1^{(g)} = \bigoplus_{g'} \mathbf{1}_{g',gg'}, \tag{88}$$

where $\mathbf{1}_{g',gg'}$ is the unit line operator between vacua $v_g$ and $v_{gg'}$.

In terms of the SymTFT, this phase is constructed by choosing the physical boundary to be the same as the symmetry boundary

$$\mathcal{B}^{\text{phys}}_{\text{T}} = \mathcal{B}^{\text{sym}}_{\text{Vec}_G^\omega}. \tag{89}$$

**$G$ SPT phases.**   The opposite extreme corresponds to choosing

$$H = G, \tag{90}$$

which is only possible if the anomaly vanishes

$$\omega = 0. \tag{91}$$

We can now choose any

$$\beta \in H^2(G, U(1)), \tag{92}$$

and the resulting gapped phase is known as the SPT phase protected by $G$ symmetry. Such a phase has a single vacuum and all the $g$ lines are trivial

$$D_1^{(g)} \cong \mathbf{1}. \tag{93}$$

The information of $\beta$ is encoded in the choice of junction local operators between $D_1^{(g)}$ lines (see example 4.2 in [58] for a proof).

In terms of the SymTFT, we choose the physical boundary to be a Neumann boundary condition for $G$ gauge fields in the 3d bulk. The various choices are parameterized by $\beta$, which is the discrete torsion along the boundary.

## 4.2 Order parameters and generalized charges

A class of order parameters is well-known. These are untwisted sector local operators transforming in irreducible representations of $G$. Different phases can be parameterized by the presence or absence of such order parameters. For example, the $G$ SSB phase contains all the order parameters, i.e. for every irreducible representation $R$ of $G$, there exists a topological local operator[20] in the IR theory transforming in $R$.

On the other hand, all of these order parameters are absent in $G$ SPT phases. Yet, the different $G$ SPT phases can be distinguished in terms of unconventional order parameters, known as string order parameters. These are twisted sector local operators, whose charges distinguish the different SPT phases. More details will be discussed in the examples later in this section. General gapped phases exhibiting partial SSB exhibit both untwisted and twisted sector order parameters. For relevant previous works discussing the use of string order parameters see e.g. [128–135].

**Line defects of the SymTFT.** In terms of the SymTFT, the various possible charges carried by untwisted and twisted local operators are encoded in the topological line defects of the DW theory $\mathfrak{Z}\left(\mathrm{Vec}_G^\omega\right)$. These can be understood by computing the Drinfeld center $\mathcal{Z}\left(\mathrm{Vec}_G^\omega\right)$ (see [58] for a recent detailed discussion). The simple lines are

$$Q_{[g],R}, \tag{94}$$

labeled by a conjugacy class $[g]$ of $G$ and an irreducible representation $R$ of a twisted group algebra $\mathbb{C}[H_g, \omega_g]$, where $H_g \subseteq G$ is the centralizer of an element $g \in [g]$ and

$$\omega_g \in H^2(G, U(1)), \tag{95}$$

is obtained from $\omega$ by taking the slant product with respect to $g$. A representative of $\omega_g$ can be obtained in terms of a representative of $\omega$ as

$$\omega_g(h_1, h_2) = \frac{\omega(g, h_1, h_2)\omega(h_1, h_2, g)}{\omega(h_1, g, h_2)}, \qquad h_1, h_2 \in H_g. \tag{96}$$

The twisted group algebra $\mathbb{C}[H_g, \omega_g]$ can then be defined in terms of a basis of vectors

$$V_h, \qquad h \in H_g, \tag{97}$$

with product

$$V_{h_1} V_{h_2} = \omega_g(h_1, h_2) V_{h_1 h_2}. \tag{98}$$

For the trivial conjugacy class $[g] = [\mathrm{id}]$, the slant product vanishes $\omega_{\mathrm{id}} = 0$, and $R$ are irreducible representations of $G$. Physically, these lines $Q_{[\mathrm{id}],R}$ are the **Wilson lines** for the 3d $G$ gauge theory. On the other hand, a line $Q_{[g],1}$ for a non-trivial conjugacy class $[g]$ but a trivial representation is a **vortex line** around which we have a holonomy for the $G$ gauge fields.[21] The remaining lines $Q_{[g],R}$ are mixed vortex-Wilson lines obtained by dressing the vortex lines with Wilson lines.

---

[20]In fact, there is a $\dim(R)$ number of such operators.

[21]Sometimes such lines are also referred to magnetic lines, but we do not use this terminology as it has the danger of confusion with 't Hooft defects describing worldvolumes of probe magnetic monopoles. The 't Hooft defects are codimension-3 in spacetime while the vortex defects being considered here are codimension-2 in spacetime. As we are in spacetime dimension 3, line defects are codimension-2, and indeed the line defects under study precisely induce vortex configurations for gauge fields.

**Associated multiplets.** Let us describe the multiplets of local operators carrying generalized charges $Q_{[g],R}$. The usual charges correspond to the Wilson lines $Q_{[\mathrm{id}],R}$, describing untwisted sector local operators transforming in representation $R$

$$
\mathcal{O} \in R \quad = \quad g \cdot \mathcal{O} \in R \tag{99}
$$

On the other hand, the vortex lines $Q_{[g],1}$ describe local operators living in twisted sectors for elements $g \in [g]$, i.e. attached to $g$-lines. There is a single local operator (up to multiplication by a complex number) $\mathcal{O}_g$ in each twisted sector, and the action of $G$ is just to permute these operators

$$
g \quad \mathcal{O}_g \quad = \quad g \quad \mathcal{O}_{g'^{-1}gg'} \tag{100}
$$

In particular, note that the centralizer $H_g$ of $g \in [g]$ leaves $\mathcal{O}_g$ invariant. Finally, a mixed vortex-Wilson line $Q_{[g],R}$ also describes the generalized charge of a multiplet of local operators living in twisted sectors for elements $g \in [g]$, but now there is a non-trivial action of $H_g$ on local operators living in the $g$-twisted sector. Such operators form the irreducible representation $R$ of the twisted group algebra $\mathbb{C}[H_g, \omega_g]$

$$
g \quad \mathcal{O} \in R \quad h \in H_g \quad = \quad g \quad h \cdot \mathcal{O} \in R \tag{101}
$$

The fact that $R$ is a representation of the twisted group algebra is necessary to satisfy the consistency condition arising by requiring the action of a product of two group elements to be the same as the product of the individual actions of the two group elements

$$
g \quad \mathcal{O} \quad h_1 \quad h_2 \quad = \quad g \quad h_1 \cdot (h_2 \cdot \mathcal{O}) \quad h_1 \quad h_2
$$

$$
\| \tag{102}
$$

$$
g \quad \mathcal{O} \quad h_1 h_2 \quad = \quad g \quad (h_1 h_2) \cdot \mathcal{O} \quad h_1 h_2 \quad = \quad \omega_g(h_1, h_2) \quad g \quad (h_1 h_2) \cdot \mathcal{O} \quad h_1 \quad h_2
$$

where we have used the fact that composition of $h_1$ and $h_2$ lines on a $g$ line has an extra factor of $\omega_g$ arising due to the presence of the associator $\omega$.

**Order parameters for $G$ SSB phase.** We can use the SymTFT to obtain which of the above generalized charges are carried by order parameters for various phases. First of all, consider the $G$ SSB phase for which the physical boundary condition is the same as the symmetry boundary condition (89). This boundary is the Dirichlet boundary condition for the $G$ gauge fields, and hence all Wilson lines of the 3d gauge theory can end along it. This can also be seen from the Lagrangian algebra, which is [58]

$$\mathcal{A}_{\mathrm{Dir}} = \bigoplus_R \dim(R) \boldsymbol{Q}_{[\mathrm{id}],\boldsymbol{R}} \,. \tag{103}$$

Thus the order parameters for the $G$ SSB phase carry precisely the usual untwisted charges $\boldsymbol{Q}_{[\mathrm{id}],\boldsymbol{R}}$, recovering the classic result.

**Order parameters for $G$ SPT phases.** As another example, consider the case $\omega = 0$, which allows for the presence of $G$ SPT phases. In particular, we have a trivial SPT phase for which the group cohomology class $\beta$ appearing in (92) is trivial, $\beta = 0$. The corresponding boundary condition for the SymTFT is pure Neumann without any discrete torsion. This means the vortex line defects in the 3d bulk become vortex point defects on the 2d boundary, i.e. the vortex lines can end. The corresponding Lagrangian algebra is

$$\mathcal{A}_{\mathrm{Neu}} = \bigoplus_{[g]} \boldsymbol{Q}_{[g],\boldsymbol{1}} \,, \tag{104}$$

where the sum is over all conjugacy classes in $G$. Thus the order parameters are twisted sector operators of the type discussed around (100). Non-trivial SPT phases similarly only involve twisted sector operators as order parameters, but now they carry non-trivial charges under centralizer subgroups. String order parameters for $G$ SPT phases have been previously discussed in [132–135].

## 4.3 Example: $\mathbb{Z}_2$ symmetry

Let us now discuss the possible gapped phases for some examples of $(G, \omega)$ by using their SymTFT construction. Consider the simplest non-trivial finite group

$$G = \mathbb{Z}_2 = \{\mathrm{id}, P\} \,, \qquad \omega = 0 \,. \tag{105}$$

This is a textbook example for which it is well-known that there are only two possible (1+1)d gapped phases: the $\mathbb{Z}_2$ SSB phase and the trivial $\mathbb{Z}_2$ phase.

The simple topological line operators of the associated SymTFT, which is the 3d $\mathbb{Z}_2$ DW gauge theory without a twist, also referred to as the toric code, are

$$\boldsymbol{Q}_{[\mathrm{id}],+} \equiv \boldsymbol{1} \,, \qquad \boldsymbol{Q}_{[\mathrm{id}],-} \equiv \boldsymbol{e} \,, \qquad \boldsymbol{Q}_{[P],+} \equiv \boldsymbol{m} \,, \qquad \boldsymbol{Q}_{[P],-} \equiv \boldsymbol{f} \,, \tag{106}$$

where $+$ and $-$ denote respectively the trivial and non-trivial irreducible representations of $\mathbb{Z}_2$, and we have also labeled the lines by their usual names $\boldsymbol{e}$, $\boldsymbol{m}$ and $\boldsymbol{f}$. The fusion rules of the lines are described by the group law of $\mathbb{Z}_2 \times \mathbb{Z}_2$.

The quantum dimensions of all the above lines are 1, while only $\boldsymbol{1}$, $\boldsymbol{e}$, and $\boldsymbol{m}$ are bosons. Thus, the only possible Lagrangian algebras are

$$\mathcal{A}_{\mathrm{Dir}} = \boldsymbol{1} \oplus \boldsymbol{e} \,, \qquad \mathcal{A}_{\mathrm{Neu}} = \boldsymbol{1} \oplus \boldsymbol{m} \,, \tag{107}$$

from which we at least recover that there are two irreducible $\mathbb{Z}_2$ symmetric gapped phases. More information about these phases require a more detailed analysis considered below.

### 4.3.1  $\mathbb{Z}_2$ SSB phase

Let us pick

$$\mathfrak{B}^{\text{phys}} = \mathcal{A}_{\text{Dir}}. \tag{108}$$

This 2d TQFT contains two untwisted local operators because the bulk lines **1** and $e$ can end on both boundaries, and both have a single possible end on each boundary (as the coefficient of these lines in the Lagrangian algebras is 1). We denote this diagrammatically as a schematic depiction of the SymTFT, where we have suppressed a spacetime dimension:

$$\tag{109}$$

The **1** line constructs the identity local operator 1 in the 2d TQFT, while the $e$ line constructs a non-identity local operator that we denote as $\mathcal{O}$.

The presence of two untwisted local operators implies that the resulting phase has two vacua. These should be idempotents in the algebra of untwisted local operators. In order to determine them, we need to determine the product structure on $\{1, \mathcal{O}\}$. Since 1 is the identity operator, we only need to determine $\mathcal{O}^2$. This product is highly constrained by the fusion of bulk lines

$$e \otimes e \cong \mathbf{1}, \tag{110}$$

which implies that the square of $\mathcal{O}$ must be proportional to the identity operator

$$\mathcal{O}^2 = \alpha, \qquad \alpha \in \mathbb{C}, \ \alpha \neq 0. \tag{111}$$

We can now rescale $\mathcal{O}$ to set the product to be

$$\mathcal{O}^2 = 1. \tag{112}$$

The two vacua $v_0, v_1$ are now straightforwardly determined to be

$$v_0 = \frac{1 + \mathcal{O}}{2}, \qquad v_1 = \frac{1 - \mathcal{O}}{2}, \tag{113}$$

which indeed satisfy

$$\begin{aligned} v_0 v_0 &= v_0, \\ v_1 v_1 &= v_1, \\ v_0 v_1 &= 0. \end{aligned} \tag{114}$$

As already argued before, there are no relative Euler terms for phases with invertible symmetries.

Above we have completely determined the structure of 2d TQFT (up to an overall Euler term) without symmetry. Now we would like to determine how the $\mathbb{Z}_2$ symmetry acts on this 2d TQFT. In particular, we would like to determine the line operator $D_1^{(P)}$ in the 2d TQFT implementing the $\mathbb{Z}_2$ symmetry. For this purpose, we can utilize the linking action of $D_1^{(P)}$ on the untwisted sector local operators

$$\tag{115}$$

where in the first step we have used the action (99). We denote this linking action succinctly as

$$D_1^{(P)}: \quad \mathcal{O} \to -\mathcal{O}. \tag{116}$$

The linking action on the identity operator is simply the quantum dimension of $P$

$$D_1^{(P)}: \quad 1 \to 1. \tag{117}$$

Combining these two linking actions, we obtain the linking actions on the vacua to be

$$D_1^{(P)}: \quad v_0 \to v_1, \qquad v_1 \to v_0, \tag{118}$$

i.e. the $\mathbb{Z}_2$ symmetry of the 2d TQFT exchanges the two vacua. This means that $\mathbb{Z}_2$ is spontaneously broken. We can use this linking action to determine $D_1^{(P)}$ to be

$$D_1^{(P)} \cong \mathbf{1}_{01} \oplus \mathbf{1}_{10}. \tag{119}$$

It is straightforward to see that the crucial $\mathbb{Z}_2$ relation is satisfied

$$D_1^{(P)} \otimes D_1^{(P)} \cong \mathbf{1}_{00} \oplus \mathbf{1}_{11} = \mathbf{1}. \tag{120}$$

The IR image of any order parameter for this phase is the untwisted sector local operator $\mathcal{O}$. It carries a non-trivial charge under $\mathbb{Z}_2$ and can be expressed in terms of the two vacua as

$$\mathcal{O} = v_0 - v_1. \tag{121}$$

Note that the two vacua $v_0$ and $v_1$ are physically indistinguishable. That is, we can relabel $v_0$ as $v_1$ and vice-versa, and the phase will look exactly the same, with the $\mathbb{Z}_2$ symmetry acting as the exchange

$$v_0 \longleftrightarrow v_1. \tag{122}$$

The indistinguishability of the vacua is tied to the fact that there is no relative Euler term between the two.

### 4.3.2 Trivial phase

Let us now pick

$$\mathfrak{B}^{\text{phys}} = \mathcal{A}_{\text{Neu}}. \tag{123}$$

We now have only a single bulk line that can end on both boundaries

$$
\begin{array}{cc}
\mathcal{A}_{\text{Dir}} & \mathcal{A}_{\text{Neu}} \\
\vrule \quad & \quad \vrule \\
\bullet \!\!\!-\!\!\!-\!\!\!\mathbf{1}\!\!\!-\!\!\!-\!\!\!\bullet \\
\vrule \quad & \quad \vrule
\end{array}
\tag{124}
$$

and hence the only untwisted local operator in the resulting 2d TQFT is the identity local operator 1. Thus, the 2d TQFT has a single vacuum.

The linking action of $D_1^{(P)}$ is trivial

$$D_1^{(P)}: \quad 1 \to 1, \tag{125}$$

which identifies $D_1^{(P)}$ with the identity line

$$D_1^{(P)} \cong \mathbf{1}, \tag{126}$$

which clearly satisfies the $\mathbb{Z}_2$ multiplication law

$$D_1^{(P)} \otimes D_1^{(P)} \cong \mathbf{1}. \tag{127}$$

As the underlying 2d TQFT and the action of $\mathbb{Z}_2$ are both trivial, we can identify it with the trivial $\mathbb{Z}_2$ phase in which $\mathbb{Z}_2$ symmetry is not broken spontaneously.

There are no non-trivial untwisted sector order parameters. However, there are twisted sector local operators acting as order parameters for this phase. Such order parameters are also known as string order parameters. In the specific case at hand, the string order parameters carry generalized charge $\mathbf{Q}_{[P],+} \equiv \mathbf{m}$. In terms of the SymTFT, they arise from the bulk line $\mathbf{m}$, which ends along $\mathfrak{B}^{\mathrm{phys}}$ but not along $\mathfrak{B}^{\mathrm{sym}}$, leaving instead a residual $P$ line along $\mathfrak{B}^{\mathrm{sym}}$, as shown in the following figure:

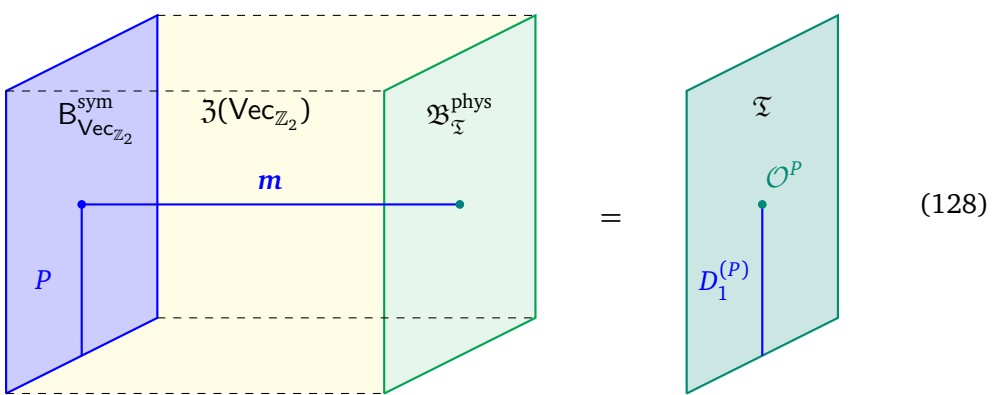

$$\tag{128}$$

The resulting local operator $\mathcal{O}^P$ lives at the end of line operator $D_1^{(P)}$ implementing the $\mathbb{Z}_2$ symmetry and is uncharged under the $\mathbb{Z}_2$ action, which means that it carries the generalized charge $\mathbf{Q}_{[P],+}$.

In the IR, an order parameter $\mathcal{O}^P$ flows to a non-zero topological local operator. Correspondingly $\mathbf{m}$ can end along the IR physical boundary $\mathcal{A}_{\mathrm{Neu}}$. On the other hand, as we discussed above, $D_1^{(P)}$ flows to the identity line. Combining the two together, we can describe the IR image $\mathcal{O}_{\mathrm{IR}}^P$ of $\mathcal{O}^P$ as an operator living at the end of an identity line, or in other words a genuine local operator. Since the genuine local operators in the IR form a one-dimensional vector space, by rescaling $\mathcal{O}^P$, we can identify $\mathcal{O}_{\mathrm{IR}}^P$ with the identity local operator,

$$\mathcal{O}_{\mathrm{IR}}^P = 1, \tag{129}$$

regarded as living at the end of the $D_1^{(P)}$ line.

## 4.4 Example: $\mathbb{Z}_2$ symmetry with anomaly

As we have

$$H^3(\mathbb{Z}_2, U(1)) = \mathbb{Z}_2, \tag{130}$$

we can have a non-trivial 't Hooft anomaly,

$$\omega \neq 0, \tag{131}$$

for a $\mathbb{Z}_2$ symmetry in 2d. Let us now study gapped phases in the presence of an anomaly and contrast it with the non-anomalous case discussed above. We will find only a single irreducible gapped phase, in which $\mathbb{Z}_2$ is spontaneously broken. We will work with a representative for $\omega$ whose only non-trivial element is

$$\omega(P, P, P) = -1. \tag{132}$$

Note that the slant product $\omega_g$ of $\omega$ must be trivial for both $g = \text{id}, P$ as

$$H^2(\mathbb{Z}_2, U(1)) = 0 \,. \tag{133}$$

The simple topological line operators of the associated SymTFT, which is the 3d $\mathbb{Z}_2$ DW gauge theory with a non-trivial twist, also referred to as the double semion model, are thus

$$Q_{[\text{id}],+} \equiv \mathbf{1} \,, \qquad Q_{[\text{id}],-} \equiv s\bar{s} \,, \qquad Q_{[P],+} \equiv s \,, \qquad Q_{[P],-} \equiv \bar{s} \,, \tag{134}$$

where $+$ and $-$ denote respectively the trivial and non-trivial irreducible representations of $\mathbb{Z}_2$, and we have also labeled the lines by their usual names $s$ for semion, $\bar{s}$ for antisemion and $s\bar{s}$ for the boson constructed as a semion-antisemion pair. The fusion rules of the lines are described by the group law of $\mathbb{Z}_2 \times \mathbb{Z}_2$.

### 4.4.1 $\mathbb{Z}_2$ SSB phase

Since we have only one non-trivial boson, there is only a single Lagrangian algebra

$$\mathcal{A}_{\text{Dir}} = \mathbf{1} \oplus s\bar{s} \,, \tag{135}$$

which is the symmetry boundary condition. Thus we have a single irreducible gapped phase corresponding to choosing

$$\mathsf{B}^{\text{phys}} = \mathsf{B}^{\text{sym}}_{\text{Vec}^{\omega}_{\mathbb{Z}_2}} = \mathcal{A}_{\text{Dir}} \,. \tag{136}$$

The resulting 2d TQFT contains two untwisted local operators because the bulk lines $\mathbf{1}$ and $s\bar{s}$ can end on both boundaries, and both have a single possible end on each boundary

$$
\begin{array}{c}
\mathcal{A}_{\text{Dir}} \qquad \mathcal{A}_{\text{Dir}} \\[4pt]
\vphantom{x} \\
\hspace{1em} s\bar{s} \\
\hline
\hspace{1em} \mathbf{1} \\
\end{array}
\tag{137}
$$

The $s\bar{s}$ line constructs a non-identity local operator that we denote as $\mathcal{O}$. This operator transforms in a non-trivial irreducible representation of $\mathbb{Z}_2$ and is the IR image of any order parameter for this phase.

The presence of two untwisted local operators implies that the resulting phase has two vacua $v_0, v_1$. Their determination is the same as for the $\mathbb{Z}_2$ SSB phase for $\omega = 0$

$$v_0 = \frac{1 + \mathcal{O}}{2} \,, \qquad v_1 = \frac{1 - \mathcal{O}}{2} \,, \tag{138}$$

and the $\mathbb{Z}_2$ line operator is

$$D_1^{(P)} \cong \mathbf{1}_{01} \oplus \mathbf{1}_{10} \,. \tag{139}$$

The 't Hooft anomaly $\omega$ has to be realized by an F-move of the following form

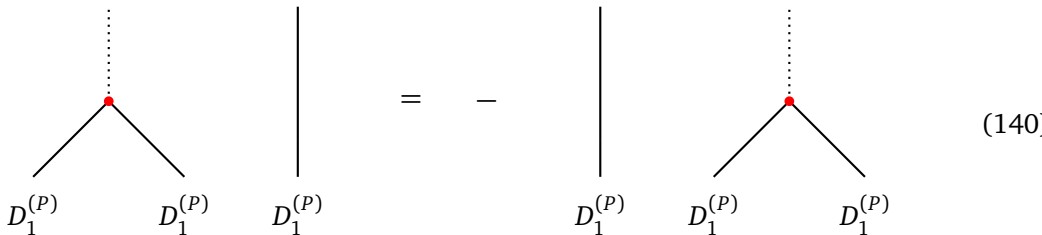

$$\tag{140}$$

where we have drawn the identity line operator as a dotted line. This constrains the choice of junction local operator shown in red above (up to multiplication by a non-zero complex number) to be

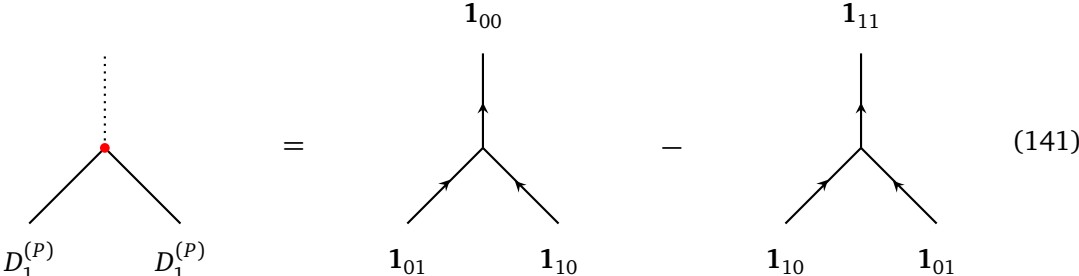

$$(141)$$

Indeed, using this choice of junction operator, we can now easily compute and verify the relation capturing the 't Hooft anomaly

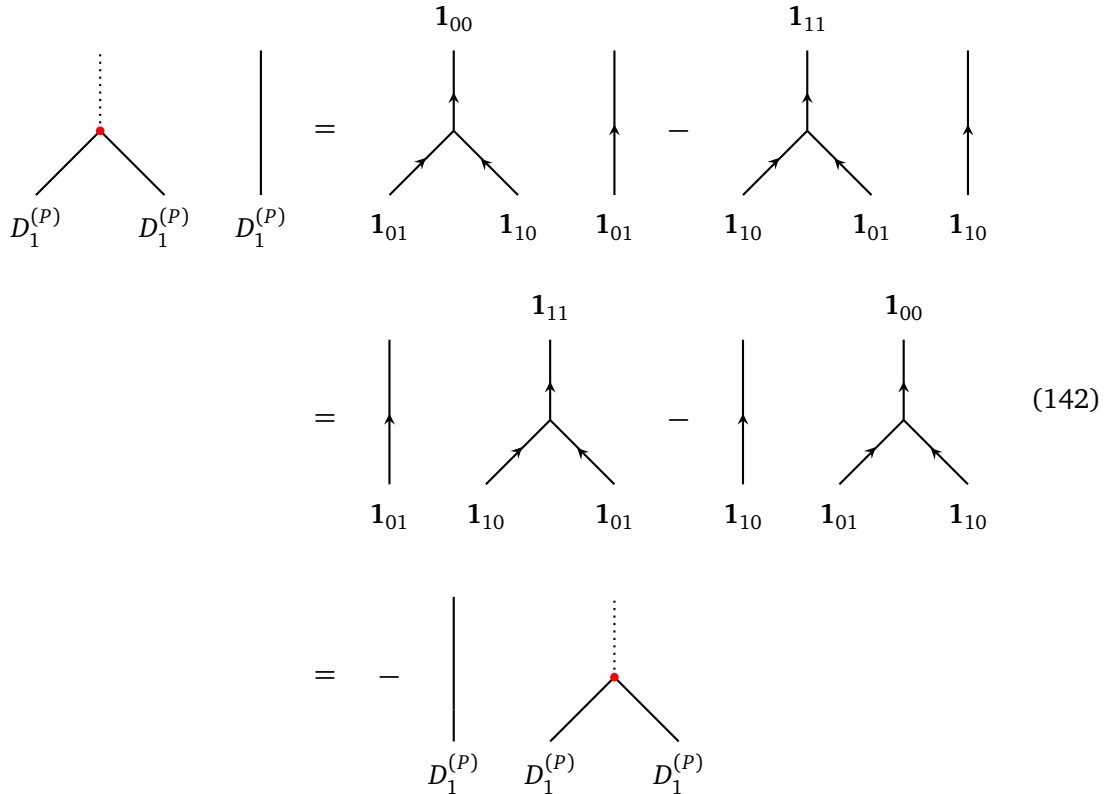

$$(142)$$

Again the two vacua are physically indistinguishable, just like the $\mathbb{Z}_2$ SSB phase for $\omega = 0$. This is again tied to the fact that there is no relative Euler term between the two vacua.

### 4.4.2 Absence of $\mathbb{Z}_2$ SPT phases

Above, using the SymTFT we did not find any trivial or non-trivial SPT phases, i.e. phases with a single vacuum. This can be seen without using SymTFT as follows. The $\mathbb{Z}_2$ symmetry in such a hypothetical phase must be realized as

$$D_1^{(P)} \cong \mathbf{1}, \tag{143}$$

but then it is impossible to satisfy the relationship (140) as it becomes a relation

$$\bullet \quad = \quad - \quad \bullet \tag{144}$$

on the junction local operator, which implies that the junction operator vanishes, leading to a contradiction.

## 4.5 Example: $\mathbb{Z}_2 \times \mathbb{Z}_2$ symmetry

We now consider

$$G = \mathbb{Z}_2 \times \mathbb{Z}_2 = \{\mathrm{id}, S, C, V\}, \tag{145}$$

where $S$, $C$ and $V$ are three order two elements. Let us consider the non-anomalous case $\omega = 0$. This is a particularly interesting case because it is the simplest setup exhibiting a non-trivial SPT phase. The second group cohomology is

$$H^2(\mathbb{Z}_2 \times \mathbb{Z}_2, U(1)) = \mathbb{Z}_2, \tag{146}$$

and the SPT phase corresponds to the non-trivial element of this group.

There are 16 simple topological lines of the SymTFT $\mathfrak{Z}(\mathrm{Vec}_{\mathbb{Z}_2 \times \mathbb{Z}_2})$, that we label as

$$Q_{g,s,s'}, \qquad g \in \mathbb{Z}_2 \times \mathbb{Z}_2, \quad s, s' \in \{+, -\}, \tag{147}$$

where the group element $g$ describes the choice of a conjugacy class and $s, s'$ are signs specifying irreducible representations of the centralizer, which is the full group $\mathbb{Z}_2 \times \mathbb{Z}_2$. The fusion rules of the lines are described by the group law of $\mathbb{Z}_2^4$

$$Q_{g_1,s_1,s_1'} \otimes Q_{g_2,s_2,s_2'} \cong Q_{g_1 g_2, s_1 s_2, s_1' s_2'}. \tag{148}$$

All these lines have quantum dimension 1, and the bosons are

$$Q_{\mathrm{id},s,s'}, \qquad Q_{g,+,+}, \qquad Q_{S,+,-}, \qquad Q_{C,-,+}, \qquad Q_{V,-,-}. \tag{149}$$

The possible Lagrangian algebras are

$$
\begin{aligned}
\mathcal{A}_{\mathrm{Dir}} &= \bigoplus_{s,s'} Q_{\mathrm{id},s,s'}, \\
\mathcal{A}_{\mathrm{Neu}} &= \bigoplus_{g} Q_{g,+,+}, \\
\mathcal{A}_{\mathrm{Neu}(S)} &= Q_{\mathrm{id},+,+} \oplus Q_{S,+,+} \oplus Q_{\mathrm{id},+,-} \oplus Q_{S,+,-}, \\
\mathcal{A}_{\mathrm{Neu}(C)} &= Q_{\mathrm{id},+,+} \oplus Q_{C,+,+} \oplus Q_{\mathrm{id},-,+} \oplus Q_{C,-,+}, \\
\mathcal{A}_{\mathrm{Neu}(V)} &= Q_{\mathrm{id},+,+} \oplus Q_{V,+,+} \oplus Q_{\mathrm{id},-,-} \oplus Q_{V,-,-}, \\
\mathcal{A}_{\mathrm{Neu, Tor}} &= Q_{\mathrm{id},+,+} \oplus Q_{S,+,-} \oplus Q_{C,-,+} \oplus Q_{V,-,-},
\end{aligned}
\tag{150}
$$

where $\mathcal{A}_{\mathrm{Dir}}$ and $\mathcal{A}_{\mathrm{Neu}}$ are Dirichlet and Neumann boundary conditions on $\mathbb{Z}_2 \times \mathbb{Z}_2$ gauge fields in the 3d bulk; $\mathcal{A}_{\mathrm{Neu}(i)}$ for $i \in \{S, C, V\}$ are boundaries where only gauge fields for the $\mathbb{Z}_2$ subgroup generated by $i$ are Neumann; and finally $\mathcal{A}_{\mathrm{Neu, Tor}}$ is the Neumann boundary condition on whole of $\mathbb{Z}_2 \times \mathbb{Z}_2$ in the presence of discrete torsion given by the non-trivial element of (146). Let us discuss the corresponding $\mathbb{Z}_2 \times \mathbb{Z}_2$ symmetric gapped phases.

### 4.5.1 $\mathbb{Z}_2 \times \mathbb{Z}_2$ SSB phase

Let us consider

$$\mathfrak{B}^{\mathrm{phys}} = \mathcal{A}_{\mathrm{Dir}}. \tag{151}$$

We obtain four untwisted local operators in the 2d theory from the ending lines $Q_{\mathrm{id},s,s'}$:

$$
\begin{array}{cc}
\mathcal{A}_{\mathrm{Dir}} & \mathcal{A}_{\mathrm{Dir}} \\
\rule{0pt}{0pt} & \\
Q_{\mathrm{id},-,-} & \\
Q_{\mathrm{id},+,-} & \\
Q_{\mathrm{id},-,+} & \\
Q_{\mathrm{id},+,+} &
\end{array}
\tag{152}
$$

Hence the resulting gapped phase has four vacua. We label these four operators as

$$
\mathcal{O}_{s,s'}, \qquad \mathcal{O}_{+,+} \equiv 1. \tag{153}
$$

The operator $\mathcal{O}_{s,s'}$ descends from the bulk line $Q_{\mathrm{id},s,s'}$.

Using the fusion rules of the bulk lines, associativity of the product, and rescaling of the operators, the product structure on the above local operators is

$$
\mathcal{O}_{s_1,s_1'} \mathcal{O}_{s_2,s_2'} = \mathcal{O}_{s_1 s_2, s_1' s_2'}. \tag{154}
$$

Finding the orthogonal idempotents, the vacua are determined to be

$$
\begin{aligned}
v_0 &= \frac{\sum_{s,s'} \mathcal{O}_{s,s'}}{4}, \\
v_1 &= \frac{\sum_{s,s'} s\,\mathcal{O}_{s,s'}}{4}, \\
v_2 &= \frac{\sum_{s,s'} s'\mathcal{O}_{s,s'}}{4}, \\
v_3 &= \frac{\sum_{s,s'} ss'\mathcal{O}_{s,s'}}{4}.
\end{aligned}
\tag{155}
$$

Now let us determine the line operators implementing the $\mathbb{Z}_2 \times \mathbb{Z}_2$ symmetry. The linking action of $S$ on the local operators is

$$
D_1^{(S)}: \quad \mathcal{O}_{s,s'} \to s\mathcal{O}_{s,s'}, \tag{156}
$$

implying the following linking action on vacua

$$
D_1^{(S)}: \quad v_0 \leftrightarrow v_1, \qquad v_2 \leftrightarrow v_3, \tag{157}
$$

which determines $D_1^{(S)}$ to be

$$
D_1^{(S)} \cong \mathbf{1}_{01} \oplus \mathbf{1}_{10} \oplus \mathbf{1}_{23} \oplus \mathbf{1}_{32}. \tag{158}
$$

Similarly, the linking action of $C$ on the local operators is

$$
D_1^{(C)}: \quad \mathcal{O}_{s,s'} \to s'\mathcal{O}_{s,s'}, \tag{159}
$$

implying the following linking action on vacua

$$
D_1^{(C)}: \quad v_0 \leftrightarrow v_2, \qquad v_1 \leftrightarrow v_3, \tag{160}
$$

which determines $D_1^{(C)}$ to be

$$D_1^{(C)} \cong \mathbf{1}_{02} \oplus \mathbf{1}_{20} \oplus \mathbf{1}_{13} \oplus \mathbf{1}_{31}. \tag{161}$$

The line $D_1^{(V)}$ is similarly determined to be

$$D_1^{(V)} \cong \mathbf{1}_{03} \oplus \mathbf{1}_{30} \oplus \mathbf{1}_{12} \oplus \mathbf{1}_{21}. \tag{162}$$

These lines indeed satisfy the $\mathbb{Z}_2 \times \mathbb{Z}_2$ group multiplication law. We thus see that all of the elements of $G = \mathbb{Z}_2 \times \mathbb{Z}_2$ are spontaneously broken in all vacua. The (IR images of the) order parameters are the local operators $\mathcal{O}_{s,s'}$ discussed above.

The four vacua are clearly physically indistinguishable.

### 4.5.2 $\mathbb{Z}_2$ SSB phases

Let us now consider

$$\mathfrak{B}^{\text{phys}} = \mathcal{A}_{\text{Neu}(i)}, \tag{163}$$

for any $i \in \{S, C, V\}$. These describe phases in which the $\mathbb{Z}_2$ subgroup generated by $i$ is *not* broken spontaneously. The analysis is similar for all three cases. For concreteness, we discuss only one of the three cases below

$$\mathfrak{B}^{\text{phys}} = \mathcal{A}_{\text{Neu}(S)}. \tag{164}$$

There are two untwisted local operators arising as

$$
\begin{array}{cc}
\mathcal{A}_{\text{Dir}} & \mathcal{A}_{\text{Neu}(S)} \\
\mathbf{Q}_{\text{id},+,-} & \\
\mathbf{Q}_{\text{id},+,+} &
\end{array}
\tag{165}
$$

and hence there are two gapped vacua. Let us denote the non-trivial operator descending from $\mathbf{Q}_{\text{id},+,-}$ as $\mathcal{O}$. The product on the operators can be fixed to be

$$\mathcal{O}^2 = 1, \tag{166}$$

from which we compute the vacua to be

$$v_0 = \frac{1 + \mathcal{O}}{2}, \qquad v_1 = \frac{1 - \mathcal{O}}{2}. \tag{167}$$

The $\mathbb{Z}_2 \times \mathbb{Z}_2$ symmetry lines have the following linking action on $\mathcal{O}$

$$
\begin{aligned}
D_1^{(S)} &: & \mathcal{O} &\to \mathcal{O}, \\
D_1^{(C)} &: & \mathcal{O} &\to -\mathcal{O}, \\
D_1^{(V)} &: & \mathcal{O} &\to -\mathcal{O},
\end{aligned}
\tag{168}
$$

which means the linking action on the vacua is

$$
\begin{aligned}
D_1^{(S)} &: & v_0 &\to v_0, & v_1 &\to v_1, \\
D_1^{(C)} &: & v_0 &\to v_1, & v_1 &\to v_0, \\
D_1^{(V)} &: & v_0 &\to v_1, & v_1 &\to v_0,
\end{aligned}
\tag{169}
$$

using which we can recognize the symmetry lines as

$$\begin{aligned}
D_1^{(S)} &\cong \mathbf{1}_{00} \oplus \mathbf{1}_{11} = \mathbf{1}\,, \\
D_1^{(C)} &\cong \mathbf{1}_{01} \oplus \mathbf{1}_{10}\,, \\
D_1^{(V)} &\cong \mathbf{1}_{01} \oplus \mathbf{1}_{10}\,.
\end{aligned} \tag{170}$$

Thus, $S$ is spontaneously unbroken in both vacua, while $C$ and $V$ are spontaneously broken in both vacua.

The generalized charges of the order parameters correspond to elements of the Lagrangian algebra $\mathcal{A}_{\mathrm{Neu}(S)}$. These are of three types, and the presence of all three is connected to the appearance of this phase in the IR:

- An order parameter of the first type is an untwisted sector local operator charged non-trivially under $C, V$, whose IR image can be identified with the operator $\mathcal{O}$ discussed above.

- An order parameter of the second type is a string order parameter. It is a local operator in the twisted sector for $S$ but charged trivially under $\mathbb{Z}_2 \times \mathbb{Z}_2$. Its IR image is the identity local operator, regarded as the end-point of line operator $D_1^{(S)} \cong \mathbf{1}$.

- An order parameter of the third type is also a string order parameter. It is a local operator in the twisted sector for $S$ but charged non-trivially under $C, V$. Its IR image is the operator $\mathcal{O}$, regarded as the end-point of line operator $D_1^{(S)} \cong \mathbf{1}$.

Note that the two vacua are physically indistinguishable: in both of them we have one $\mathbb{Z}_2$ that is unbroken.

### 4.5.3 Trivial phase

Let us consider

$$\mathfrak{B}^{\mathrm{phys}} = \mathcal{A}_{\mathrm{Neu}}\,. \tag{171}$$

There is only a single untwisted local operator in the 2d TQFT, as no non-identity bulk line can completely end on both boundaries. Thus we have a theory with a single vacuum. All the symmetry lines are trivial

$$D_1^{(i)} \cong \mathbf{1}\,, \qquad i \in \{S, C, V\}\,, \tag{172}$$

and hence there is no spontaneously broken symmetry. The junction operators between the symmetry lines are all equal to identity operators

$$
\begin{array}{c}
D_1^{(k)} \\
\big| \\
\bullet \\
\diagup \quad \diagdown \\
D_1^{(i)} \quad D_1^{(j)}
\end{array}
\qquad = \qquad \bullet\; 1\,, \tag{173}
$$

where $i, j, k$ obey the $\mathbb{Z}_2 \times \mathbb{Z}_2$ group law. This is the trivial $\mathbb{Z}_2 \times \mathbb{Z}_2$ phase in which all symmetry lines and their junctions are realized trivially with the underlying phase obtained after forgetting symmetry also being trivial.

The order parameters are twisted sector local operators which are uncharged under $\mathbb{Z}_2 \times \mathbb{Z}_2$. Their IR image is the identity local operator, regarded as the endpoint of corresponding line operators $D_1^{(i)} \cong \mathbf{1}$.

### 4.5.4 $\mathbb{Z}_2 \times \mathbb{Z}_2$ SPT phase

The choice

$$\mathfrak{B}^{\text{phys}} = \mathcal{A}_{\text{Neu, Tor}}, \tag{174}$$

yields quite similar results to the trivial $\mathbb{Z}_2$ phase discussed above. There is again only a single untwisted local operator in the 2d TQFT, as no non-identity bulk line can completely end on both boundaries. Thus we again have a theory with a single vacuum. All the symmetry lines are again trivial

$$D_1^{(i)} \cong \mathbf{1}, \qquad i \in \{S, C, V\}, \tag{175}$$

and hence there is no spontaneously broken symmetry.

However, the junction operators between the symmetry lines are not all equal to identity operators. The non-trivial junctions are

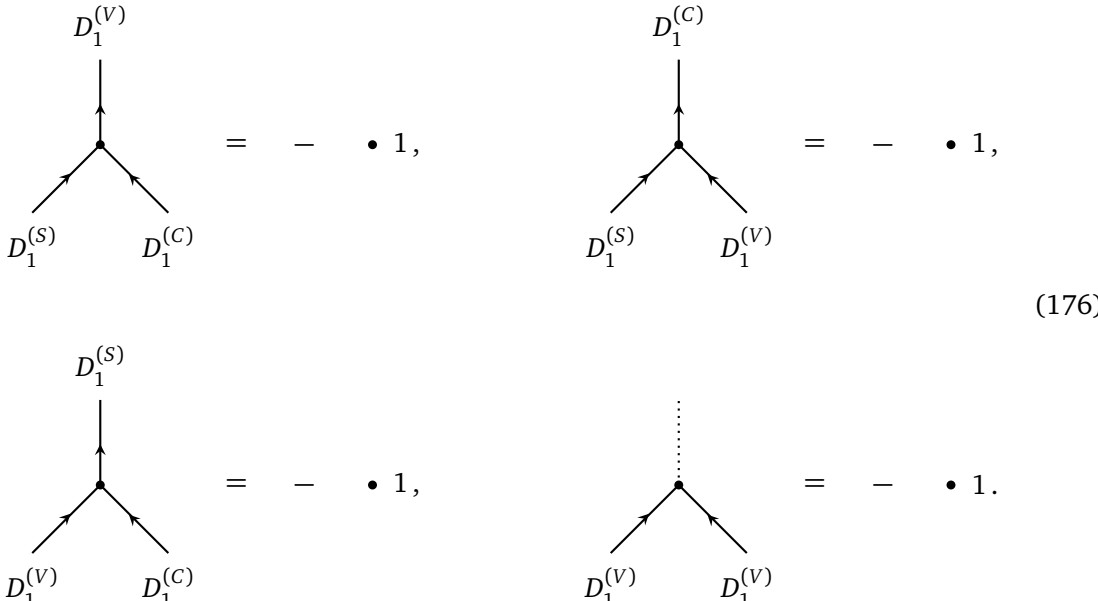

$$\tag{176}$$

Again, the dotted line indicates the identity line $D_1^{(\text{id})} = \mathbf{1}$. This is crucial for realizing the order parameters in the IR TQFT. From the Lagrangian algebra, we see that the order parameters are twisted sector operators that are non-trivially charged under $\mathbb{Z}_2 \times \mathbb{Z}_2$ in a specific way. In more detail, we have

- An $S$-twisted sector local operator charged under $C, V$.

- A $C$-twisted sector local operator charged under $S, V$.

- A $V$-twisted sector local operator charged under $S, C$.

The IR images of all these local operators can be identified with the identity local operator, regarded as the endpoint of line operators $D_1^{(i)} \cong \mathbf{1}$. For a discussion of the $\mathbb{Z}_2 \times \mathbb{Z}_2$ SPT phase and string order parameters from a lattice model perspective see [129, 130].

Let us describe how the non-trivial charges are realized as a consequence of the presence of non-trivial junction operators between symmetry lines. Without loss of generality, we consider the case of $S$-twisted sector operator (referred to as $\mathcal{O}_S$ below), as the other cases are similar. To begin with, the definition of $\mathcal{O}_S$ is

$$D_1^{(S)} \xleftarrow{\quad\quad} \bullet_{\mathcal{O}_S} \quad = \quad \bullet_1 \tag{177}$$

We need to show that we have the relation

$$
D_1^{(S)} \xrightarrow{\hspace{1cm}} \bullet\ \big|_{\mathcal{O}_S}^{D_1^{(C)}} \quad = \quad - \quad D_1^{(S)} \xrightarrow{\hspace{0.5cm}}\Big|_{\mathcal{O}_S}^{D_1^{(C)}} \bullet
\tag{178}
$$

meaning that $\mathcal{O}_S$ is charged under $C$. In order to see this, let us evaluate the two sides separately. The left-hand side is simply

$$
D_1^{(S)} \xrightarrow{\hspace{1cm}} \bullet\ \big|_{\mathcal{O}_S}^{D_1^{(C)}} \quad = \quad \underset{1}{\bullet}
\tag{179}
$$

by using the above definition of $\mathcal{O}_S$ and the fact that $D_1^{(C)}$ is the identity line. On the other hand, the right-hand side is

$$
D_1^{(S)} \xrightarrow{\hspace{0.5cm}}\Big|_{\mathcal{O}_S}^{D_1^{(C)}}\bullet \quad = \quad D_1^{(S)} \xrightarrow{}\overset{D_1^{(C)}}{\underset{D_1^{(C)}}{\bullet}}\overset{D_1^{(V)}\quad D_1^{(S)}}{\longleftarrow}\underset{\mathcal{O}_S}{\bullet} \quad = \quad \underset{-1}{\bullet}\quad\underset{1}{\bullet}\quad\underset{1}{\bullet}
$$

$$
= \quad - \quad \underset{1}{\bullet}
\tag{180}
$$

where we have resolved the quadrivalent junction into two trivalent junctions, one of which is non-trivial by the above-discussed assignments. We have thus shown explicitly the relation (178). One could have also resolved the quadrivalent junction in the opposite order, but that leads to the same result

$$
D_1^{(S)} \xrightarrow{\hspace{0.5cm}}\Big|_{\mathcal{O}_S}^{D_1^{(C)}}\bullet \quad = \quad D_1^{(S)} \xrightarrow{}\overset{D_1^{(C)}}{\bullet}\overset{D_1^{(V)}\ \ D_1^{(S)}}{\longleftarrow}\underset{D_1^{(C)}}{\underset{\mathcal{O}_S}{\bullet}} \quad = \quad \underset{1}{\bullet}\quad\underset{-1}{\bullet}\quad\underset{1}{\bullet}
$$

$$
= \quad - \quad \underset{1}{\bullet}
\tag{181}
$$

In a similar way, the reader can see that the following relationship holds

$$
D_1^{(S)} \xrightarrow{\hspace{1cm}} \bullet\ \big|_{\mathcal{O}_S}^{D_1^{(V)}} \quad = \quad - \quad D_1^{(S)} \xrightarrow{\hspace{0.5cm}}\Big|_{\mathcal{O}_S}^{D_1^{(V)}} \bullet
\tag{182}
$$

## 4.6  Example: $S_3$ symmetry

We now consider the symmetry group $S_3$, namely the non-abelian group formed by permutations of three objects. We will consider the non-anomalous case $\omega = 0$. The gapped phases in this group-symmetry example has been studied using different methods in [91]. This symmetry group is rather interesting because it is the simplest case in which we obtain a non-invertible symmetry after gauging an invertible symmetry: gauging all of $S_3$ leads to non-invertible $\mathrm{Rep}(S_3)$ symmetry. We will study the $\mathrm{Rep}(S_3)$-symmetric gapped phases in the next section, which will be related by gauging to the $S_3$-symmetric phases studied below.

We label the elements of $S_3$ as

$$S_3 = \left\{ \mathrm{id}, a, a^2, b, ab, a^2 b \right\} , \tag{183}$$

where $a$ is an order 3 element implementing a cyclic permutation, and $b$ is an order 2 element implementing a transposition, i.e.

$$a^3 = b^2 = \mathrm{id} . \tag{184}$$

The non-abelian nature of the group is encoded in the following relation

$$ab = ba^2 . \tag{185}$$

The conjugacy classes are

$$[\mathrm{id}] = \{\mathrm{id}\} , \qquad [a] = \{a, a^2\} , \qquad [b] = \{b, ab, a^2 b\} , \tag{186}$$

with corresponding centralizers

$$H_{\mathrm{id}} = S_3 , \qquad H_a = \mathbb{Z}_3 = \{\mathrm{id}, a, a^2\} , \qquad H_b = \mathbb{Z}_2 = \{\mathrm{id}, b\} . \tag{187}$$

Consequently, the simple topological lines of the SymTFT $\mathfrak{Z}(\mathrm{Vec}_{S_3})$ are labeled by the conjugacy class and representation of the centralizer,

$$\begin{aligned}
&Q_{[\mathrm{id}],1} , &&Q_{[\mathrm{id}],P} , &&Q_{[\mathrm{id}],E} , \\
&Q_{[a],1} , &&Q_{[a],\omega} , &&Q_{[a],\omega^2} , \\
&Q_{[b],+} , &&Q_{[b],-} .
\end{aligned} \tag{188}$$

Here we denote by 1 the trivial representation of $S_3$ or $\mathbb{Z}_3$; by $P$ the sign representation of $S_3$, which is a one-dimensional representation on which $a$ acts trivially and $b$ acts by a sign; by $E$ the standard representation of $S_3$, which is a two-dimensional irreducible representation of $S_3$ that admits a basis $\{V_1, V_2\}$ such that the action of $S_3$ is

$$\begin{aligned}
a: \quad & V_1 \to \omega V_1 , \qquad V_2 \to \omega^2 V_2 , \\
b: \quad & V_1 \leftrightarrow V_2 ,
\end{aligned} \tag{189}$$

where $\omega = e^{2\pi i/3}$.

The fusion rules of the lines are

$$
\begin{aligned}
Q_{[\text{id}],P} \otimes Q_{[\text{id}],P} &\cong Q_{[\text{id}],1} \,, \\
Q_{[\text{id}],P} \otimes Q_{[\text{id}],E} &\cong Q_{[\text{id}],E} \,, \\
Q_{[\text{id}],E} \otimes Q_{[\text{id}],E} &\cong Q_{[\text{id}],1} \oplus Q_{[\text{id}],P} \oplus Q_{[\text{id}],E} \,, \\
Q_{[\text{id}],P} \otimes Q_{[a],\omega^i} &\cong Q_{[a],\omega^i} \,, \qquad i \in \{0,1,2\} \,, \\
Q_{[\text{id}],P} \otimes Q_{[b],s} &\cong Q_{[b],-s} \,, \qquad s \in \{+,-\} \,, \\
Q_{[\text{id}],E} \otimes Q_{[a],\omega^i} &\cong Q_{[a],\omega^{i+1}} \oplus Q_{[a],\omega^{i+2}} \,, \\
Q_{[a],\omega^i} \otimes Q_{[a],\omega^i} &\cong Q_{[a],\omega^i} \oplus Q_{[\text{id}],1} \oplus Q_{[\text{id}],P} \,, \\
Q_{[a],\omega^i} \otimes Q_{[a],\omega^{i+1}} &\cong Q_{[a],\omega^{i+2}} \oplus Q_{[\text{id}],E} \,, \\
Q_{[a],\omega^i} \otimes Q_{[b],s} &\cong Q_{[b],+} \oplus Q_{[b],-} \,, \\
Q_{[b],s} \otimes Q_{[b],s} &\cong \bigoplus_{i=0}^{2} Q_{[a],\omega^i} \oplus Q_{[\text{id}],1} \oplus Q_{[\text{id}],E} \,, \\
Q_{[b],s} \otimes Q_{[b],-s} &\cong \bigoplus_{i=0}^{2} Q_{[a],\omega^i} \oplus Q_{[\text{id}],P} \oplus Q_{[\text{id}],E} \,.
\end{aligned}
\tag{190}
$$

The quantum dimensions of the lines are

$$
\dim(Q_{[\text{id}],P}) = 1 \,, \qquad \dim(Q_{[\text{id}],E}) = 2 \,, \qquad \dim(Q_{[a],\omega^i}) = 2 \,, \qquad \dim(Q_{[b],s}) = 3 \,, \tag{191}
$$

and the bosons are

$$
Q_{[\text{id}],P} \,, \qquad Q_{[\text{id}],E} \,, \qquad Q_{[a],1} \,, \qquad Q_{[b],+} \,. \tag{192}
$$

The Lagrangian algebras are

$$
\begin{aligned}
\mathcal{A}_{\text{Dir}} &= Q_{[\text{id}],1} \oplus Q_{[\text{id}],P} \oplus 2Q_{[\text{id}],E} \,, \\
\mathcal{A}_{\text{Neu}} &= Q_{[\text{id}],1} \oplus Q_{[a],1} \oplus Q_{[b],+} \,, \\
\mathcal{A}_{\text{Neu}(\mathbb{Z}_2)} &= Q_{[\text{id}],1} \oplus Q_{[\text{id}],E} \oplus Q_{[b],+} \,, \\
\mathcal{A}_{\text{Neu}(\mathbb{Z}_3)} &= Q_{[\text{id}],1} \oplus Q_{[\text{id}],P} \oplus 2Q_{[a],1} \,.
\end{aligned}
\tag{193}
$$

Here $\mathcal{A}_{\text{Neu}}$ corresponds to Neumann boundary condition on the full $S_3$ gauge fields, and $\mathcal{A}_{\text{Neu}(\mathbb{Z}_2)}$, $\mathcal{A}_{\text{Neu}(\mathbb{Z}_3)}$ correspond respectively to Neumann boundary conditions on $\mathbb{Z}_2 \subseteq S_3$ and $\mathbb{Z}_3 \subseteq S_3$ gauge fields.

For the $S_3$ group symmetry we choose the symmetry boundary to be

$$
\mathfrak{B}^{\text{sym}} = \mathcal{A}_{\text{Dir}} \,. \tag{194}
$$

Note that we could have equivalently chosen the symmetry boundary to be $\mathcal{A}_{\text{Neu}(\mathbb{Z}_3)}$.

### 4.6.1 $S_3$ SSB phase

Let us begin by considering the physical boundary to be the same as the symmetry boundary:

$$
\mathfrak{B}^{\text{phys}} = \mathcal{A}_{\text{Dir}} \,. \tag{195}
$$

Since the coefficient of $Q_{[\text{id}],E}$ in the Lagrangian algebra $\mathcal{A}_{\text{Dir}}$ is 2, there are two linearly independent ends for the bulk line $Q_{[\text{id}],E}$ on the corresponding boundary. Thus, we have two ends

of $\boldsymbol{Q}_{[\text{id}],E}$ on both boundaries. We denote this situation as

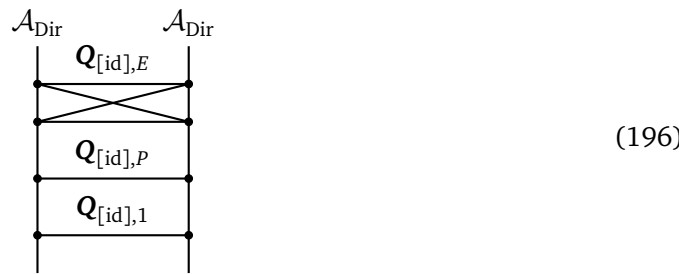

$$(196)$$

Thus, there are four possible compactifications of the $\boldsymbol{Q}_{[\text{id}],E}$ line and a single compactification for the other two lines. Overall, there are a total of 6 untwisted sector local operators, and hence the resulting gapped phase has 6 vacua.

Let us label the four operators descending from $\boldsymbol{Q}_{[\text{id}],E}$ as

$$\mathcal{O}_{i,j}, \qquad i,j \in \{1,2\}, \tag{197}$$

and the operator descending from $\boldsymbol{Q}_{[\text{id}],P}$ as $\mathcal{O}_P$. There are $S_3$ symmetries on both the left and right boundaries, with the action

$$
\begin{aligned}
a_L: \quad & \mathcal{O}_{i,j} \to \omega^i \mathcal{O}_{i,j}, & \mathcal{O}_P \to \mathcal{O}_P, \\
a_R: \quad & \mathcal{O}_{i,j} \to \omega^j \mathcal{O}_{i,j}, & \mathcal{O}_P \to \mathcal{O}_P, \\
b_L: \quad & \mathcal{O}_{i,j} \to \mathcal{O}_{[1+i]_2,j}, & \mathcal{O}_P \to -\mathcal{O}_P, \\
b_R: \quad & \mathcal{O}_{i,j} \to \mathcal{O}_{i,[1+j]_2}, & \mathcal{O}_P \to -\mathcal{O}_P,
\end{aligned}
\tag{198}
$$

where by $[\cdots]_2$ we mean modulo 2. The product of these operators has to obey both the $S_3$ actions. This lets us easily deduce the product to be as follows.

From the fusion

$$\boldsymbol{Q}_{[\text{id}],P} \otimes \boldsymbol{Q}_{[\text{id}],P} \cong \boldsymbol{Q}_{[\text{id}],1}, \tag{199}$$

we can fix the square of $\mathcal{O}_P$ to be $\mathcal{O}_P^2 = \alpha$, which we can rescale to be

$$\mathcal{O}_P^2 = 1. \tag{200}$$

From the fusion

$$\boldsymbol{Q}_{[\text{id}],P} \otimes \boldsymbol{Q}_{[\text{id}],E} \cong \boldsymbol{Q}_{[\text{id}],E}, \tag{201}$$

we can fix the product $\mathcal{O}_P \mathcal{O}_{1,1} = \beta \mathcal{O}_{1,1}$ where other $\mathcal{O}_{i,j}$ do not appear on the RHS because both the LHS and the RHS need to have the same transformation properties under the left and right $\mathbb{Z}_3 \subseteq S_3$ symmetries. Using associativity with (200), we find $\beta = \pm 1$, which can be fixed by rescaling $\mathcal{O}_P$ by a sign so that we have

$$\mathcal{O}_P \mathcal{O}_{1,1} = \mathcal{O}_{1,1}. \tag{202}$$

Acting by $\mathbb{Z}_2 \subseteq S_3$ symmetries on both boundaries on the above equation, we deduce

$$\mathcal{O}_P \mathcal{O}_{i,j} = (-1)^{[i+j]_2} \mathcal{O}_{i,j}. \tag{203}$$

Similarly, combining the fusion

$$\boldsymbol{Q}_{[\text{id}],E} \otimes \boldsymbol{Q}_{[\text{id}],E} \cong \boldsymbol{Q}_{[\text{id}],1} \oplus \boldsymbol{Q}_{[\text{id}],P} \oplus \boldsymbol{Q}_{[\text{id}],E}, \tag{204}$$

with the left and right $\mathbb{Z}_3$ symmetries, we deduce that

$$
\begin{aligned}
\mathcal{O}_{i,i} \mathcal{O}_{[1+i]_2,i} &= 0, \\
\mathcal{O}_{i,i} \mathcal{O}_{i,[1+i]_2} &= 0,
\end{aligned}
\tag{205}
$$

because the result should contain local operators on which left $\mathbb{Z}_3$ symmetry acts non-trivially while the right $\mathbb{Z}_3$ symmetry acts trivially, but we do not have such operators. We also deduce that

$$\mathcal{O}_{1,1}\mathcal{O}_{1,1} = \mathcal{O}_{2,2}\,, \tag{206}$$

where the coefficient on the RHS has been fixed to 1 by rescaling $\mathcal{O}_{1,1}$. Applying left and right $\mathbb{Z}_2$ symmetries on the above equation we deduce

$$\mathcal{O}_{i,j}\mathcal{O}_{i,j} = \mathcal{O}_{[1+i]_2,[1+j]_2}\,. \tag{207}$$

Finally, in a similar fashion as above, we find $\mathcal{O}_{1,1}\mathcal{O}_{2,2} = \gamma + \delta\mathcal{O}_P$. Imposing associativity with (202) implies $\gamma = \delta$ and imposing associativity with (206) implies that $2\gamma = 1$, thus leading to

$$\mathcal{O}_{1,1}\mathcal{O}_{2,2} = \frac{1+\mathcal{O}_P}{2}\,. \tag{208}$$

Applying the left $\mathbb{Z}_2$ symmetry on the above equation we deduce

$$\mathcal{O}_{2,1}\mathcal{O}_{1,2} = \frac{1-\mathcal{O}_P}{2}\,. \tag{209}$$

The vacua $v_k$ need to satisfy the relation

$$v_k \cdot v_l = \delta_{kl}v_l\,, \tag{210}$$

i.e. they are orthogonal idempotents. We can make a general ansatz for

$$v_0 = x_1 + x_P\mathcal{O}_P + \sum_{i,j}x_{i,j}\mathcal{O}_{i,j}\,, \qquad x_1, x_P, x_{i,j} \in \mathbb{C}\,, \tag{211}$$

and then successively solve for $v_1, v_2$ etc. We can solve for $v_0^2 = v_0$ first, and then require orthogonality with $v_1$. Not all solutions for $v_0$ will allow for the same number of solutions for $v_1$. Successively picking vacua $v_k$ with the largest solution space for $v_{k+1}$ (orthogonal and idempotent) results in 6 solutions:

$$\begin{aligned}
v_0 &= \frac{1+\mathcal{O}_P+2\mathcal{O}_{1,1}+2\mathcal{O}_{2,2}}{6}\,,\\
v_1 &= \frac{1+\mathcal{O}_P+2\omega\mathcal{O}_{1,1}+2\omega^2\mathcal{O}_{2,2}}{6}\,,\\
v_2 &= \frac{1+\mathcal{O}_P+2\omega^2\mathcal{O}_{1,1}+2\omega\mathcal{O}_{2,2}}{6}\,,\\
v_3 &= \frac{1-\mathcal{O}_P+2\mathcal{O}_{1,2}+2\mathcal{O}_{2,1}}{6}\,,\\
v_4 &= \frac{1-\mathcal{O}_P+2\omega\mathcal{O}_{1,2}+2\omega^2\mathcal{O}_{2,1}}{6}\,,\\
v_5 &= \frac{1-\mathcal{O}_P+2\omega^2\mathcal{O}_{1,2}+2\omega\mathcal{O}_{2,1}}{6}\,.
\end{aligned} \tag{212}$$

To track the action of $G = S_3$ symmetry on these vacua, recall that the symmetry boundary is on the left, so the left $S_3$ symmetry is the $S_3$ symmetry we are choosing to be the symmetry of the 2d TQFT. From the action of left $S_3$ symmetry on the operators, we can deduce the lines implementing the $S_3$ symmetry in the 2d TQFT as

$$\begin{aligned}
D_1^{(a)} &\cong \mathbf{1}_{01} \oplus \mathbf{1}_{12} \oplus \mathbf{1}_{20} \oplus \mathbf{1}_{34} \oplus \mathbf{1}_{45} \oplus \mathbf{1}_{53}\,,\\
D_1^{(b)} &\cong \mathbf{1}_{03} \oplus \mathbf{1}_{15} \oplus \mathbf{1}_{24} \oplus \mathbf{1}_{30} \oplus \mathbf{1}_{51} \oplus \mathbf{1}_{42}\,.
\end{aligned} \tag{213}$$

As a check, one can verify these satisfy the $S_3$ fusion rules. For example

$$D_1^{(a)} \otimes D_1^{(b)} \cong D_1^{(b)} \otimes D_1^{(a^2)} \cong \mathbf{1}_{05} \oplus \mathbf{1}_{14} \oplus \mathbf{1}_{23} \oplus \mathbf{1}_{32} \oplus \mathbf{1}_{41} \oplus \mathbf{1}_{50}. \tag{214}$$

Thus the $S_3$ symmetry is broken spontaneously in all the 6 vacua. All these vacua are physically indistinguishable.

The (IR images of the) order parameters are precisely the untwisted local operators discussed above: $\mathcal{O}_P$ transforming in the sign representation $P$ of $S_3$, $\mathcal{O}_{1,1}, \mathcal{O}_{2,1}$ transforming in the standard representation $E$ of $S_3$, and $\mathcal{O}_{1,2}, \mathcal{O}_{2,2}$ also transforming in the standard representation $E$ of $S_3$. Note that there are two possible IR images for an order parameter transforming in $E$ representation of $S_3$. This is correlated with the fact that the topological line defect describing the generalized charge $\mathbf{Q}_{[\mathrm{id},E]}$ enters in the Lagrangian algebra $\mathcal{A}_{\mathrm{phys}} = \mathcal{A}_{\mathrm{Dir}}$ with coefficient 2.

### 4.6.2 $\mathbb{Z}_3$ SSB phase

Let us now consider

$$\mathfrak{B}^{\mathrm{phys}} = \mathcal{A}_{\mathrm{Neu}(\mathbb{Z}_2)}. \tag{215}$$

The various compactifications of bulk lines are

$$
\begin{array}{cc}
\mathcal{A}_{\mathrm{Dir}} & \mathcal{A}_{\mathrm{Neu}(\mathbb{Z}_2)} \\
\end{array}
$$

$$
\boxed{
\begin{array}{c}
\mathbf{Q}_{[\mathrm{id}],E} \\
\\
\mathbf{Q}_{[\mathrm{id}],1}
\end{array}
}
\tag{216}
$$

Note that there are two possible compactifications of $\mathbf{Q}_{[\mathrm{id}],E}$ because it has two possible ends along $\mathcal{A}_{\mathrm{Dir}}$ and one possible end along $\mathcal{A}_{\mathrm{Neu}(\mathbb{Z}_2)}$. In total, we obtain three untwisted sector local operators, and hence 3 vacua, in the resulting 2d TQFT.

We denote the local operators obtained from $\mathbf{Q}_{[\mathrm{id}],E}$ compactifications as

$$\mathcal{O}_i, \qquad i \in \{1, 2\}, \tag{217}$$

which form the $E$ representation under the $S_3$ symmetry of the 2d TQFT. Combining the fusion (204) with the action of $S_3$ symmetry, along with associativity and suitable rescalings, we obtain the product rules

$$
\begin{aligned}
\mathcal{O}_1 \mathcal{O}_1 &= \mathcal{O}_2, \\
\mathcal{O}_2 \mathcal{O}_2 &= \mathcal{O}_1, \\
\mathcal{O}_1 \mathcal{O}_2 &= 1.
\end{aligned}
\tag{218}
$$

From this, we determine the 3 vacua to be

$$v_j = \frac{1 + \omega^j \mathcal{O}_1 + \omega^{2j} \mathcal{O}_2}{3}, \tag{219}$$

for $j \in \{0, 1, 2\}$. We can move between the vacua by applying $\mathbb{Z}_3$ action, hence the $\mathbb{Z}_3$ subsymmetry is spontaneously broken in all 3 vacua, thus this is the $\mathbb{Z}_3$ SSB phase. However, note that the $\mathbb{Z}_2$ subgroup generated by $b$ permutes $v_1, v_2$, and leaves $v_0$ invariant. Consequently, this $\mathbb{Z}_2$ subgroup is spontaneously broken in $v_1, v_2$, but spontaneously unbroken in $v_0$. The vacua $v_1$ and $v_2$ preserve other $\mathbb{Z}_2$ subgroups of $S_3$ whose generators are $ab$ and $a^2 b$. We can identify the $S_3$ generators as the lines

$$
\begin{aligned}
D_1^{(a)} &\cong \mathbf{1}_{01} \oplus \mathbf{1}_{12} \oplus \mathbf{1}_{20}, \\
D_1^{(b)} &\cong \mathbf{1}_{00} \oplus \mathbf{1}_{12} \oplus \mathbf{1}_{21}.
\end{aligned}
\tag{220}
$$

From the Lagrangian algebra $\mathcal{A}_{\text{Neu}(\mathbb{Z}_2)}$, we see that there are both untwisted and twisted sector order parameters. The IR images of the untwisted sector order parameters are the operators $\mathcal{O}_1, \mathcal{O}_2$ discussed above. A twisted sector (or string) order parameter is a multiplet of local operators carrying generalized charge $\boldsymbol{Q}_{[b],+}$. As we discussed earlier, such a multiplet includes three local operators $\mathcal{O}_{a^i b}$ living respectively at the ends of topological lines $D_1^{(a^i b)}$ for $i \in \{0, 1, 2\}$. Let us consider the local operator $\mathcal{O}_b$ which lives at the end of $D_1^{(b)}$. Note that $D_1^{(b)}$ comprises of irreducible lines $\mathbf{1}_{00}$, $\mathbf{1}_{12}$ and $\mathbf{1}_{21}$. Out of these irreducible lines, only $\mathbf{1}_{00}$ can end, and has precisely one end given by the local operator $v_0$, as $\mathbf{1}_{00}$ is just the identity line in the 0-th vacuum. Thus, we must have

$$\mathcal{O}_b = v_0 \,, \tag{221}$$

where a possible scalar coefficient on the right-hand side has been removed by rescaling $\mathcal{O}_b$. By exactly similar arguments, we must have

$$\mathcal{O}_{a^i b} = v_i \,. \tag{222}$$

It is easy to see that the action of $S_3$ is respected. The $\mathbb{Z}_3$ subgroup should permute the operators $\mathcal{O}_{a^i b}$, and indeed it permutes the vacua $v_i$. The $\mathbb{Z}_2$ subgroup generated by $b$ should act trivially on $\mathcal{O}_b$, while exchanging $\mathcal{O}_{ab}$ and $\mathcal{O}_{a^2 b}$, which is indeed how it acts on vacua $v_i$.

Note that the three vacua are physically indistinguishable: in each vacuum a $\mathbb{Z}_2$ subgroup of $S_3$ is left spontaneously unbroken, and all three $\mathbb{Z}_2$ subgroups in $S_3$ are equivalent (i.e. related by automorphisms of $S_3$).

### 4.6.3  $\mathbb{Z}_2$ SSB phase

Now consider the physical boundary to be

$$\mathfrak{B}^{\text{phys}} = \mathcal{A}_{\text{Neu}(\mathbb{Z}_3)} \,. \tag{223}$$

The resulting gapped phase has 2 vacua, due to the presence of 2 untwisted sector local operators arising as the following compactifications of bulk lines

$$
\begin{array}{c}
\mathcal{A}_{\text{Dir}} \qquad \mathcal{A}_{\text{Neu}(\mathbb{Z}_3)} \\
\boldsymbol{Q}_{[\text{id}],P} \\
\boldsymbol{Q}_{[\text{id}],1}
\end{array}
\tag{224}
$$

Let us denote the non-trivial local operator descending from $\boldsymbol{Q}_{[\text{id}],P}$ as $\mathcal{O}_P$. Using the same arguments we used to derive (200), we again find

$$\mathcal{O}_P^2 = 1 \,, \tag{225}$$

and so the two vacua are

$$v_0 = \frac{1 + \mathcal{O}_P}{2} \,, \qquad v_1 = \frac{1 - \mathcal{O}_P}{2} \,. \tag{226}$$

The $\mathbb{Z}_3$ sub-symmetry acts trivially on $\mathcal{O}_P$, while the $\mathbb{Z}_2$ acts by a sign

$$\mathcal{O}_P \rightarrow -\mathcal{O}_P \,. \tag{227}$$

Thus we can identify the symmetry lines as

$$
\begin{aligned}
D_1^{(a)} &\cong \mathbf{1}_{00} \oplus \mathbf{1}_{11} = \mathbf{1} \,, \\
D_1^{(b)} &\cong \mathbf{1}_{01} \oplus \mathbf{1}_{10} \,.
\end{aligned}
\tag{228}
$$

Thus all three $\mathbb{Z}_2$ subgroups of $S_3$ are spontaneously broken in both vacua, while the $\mathbb{Z}_3$ subgroup is unbroken in both vacua. In other words, both vacua are physically indistinguishable.

For this phase, we have untwisted and twisted sector order parameters. An untwisted sector order parameter carries generalized charge $\boldsymbol{Q}_{[\text{id}],P}$ and its IR image is the operator $\mathcal{O}_P$ discussed above. A twisted sector order parameter carries generalized charge $\boldsymbol{Q}_{[a],1}$ and its IR image is a multiplet of two local operators $\mathcal{O}_a, \mathcal{O}_{a^2}$ living respectively at the ends of $D_1^{(a)}, D_1^{(a^2)}$. As both these lines are identity lines, we can simply choose

$$\mathcal{O}_a = \mathcal{O}_{a^2} = 1, \tag{229}$$

where the identity operator 1 is now regarded as a twisted sector operator for the trivially acting $\mathbb{Z}_3$ symmetry.

### 4.6.4 Trivial phase

Finally, consider the physical boundary to be

$$\mathfrak{B}^{\text{phys}} = \mathcal{A}_{\text{Neu}}. \tag{230}$$

The only possible line configuration that can end on both boundaries is

$$\begin{array}{cc} \mathcal{A}_{\text{Dir}} & \mathcal{A}_{\text{Neu}} \\[4pt] \boldsymbol{Q}_{[\text{id}],1} & \end{array} \tag{231}$$

which constructs the identity local operator. The resulting phase has a single vacuum. All symmetry lines can be identified with the identity line

$$D_1^{(a)} \cong D_1^{(b)} \cong \mathbf{1}. \tag{232}$$

All of $S_3$ symmetry is spontaneously unbroken.

The order parameters are twisted sector local operators that are uncharged under centralizers. Thus, we have a completely trivial phase in the IR. We can identify the IR image of the order parameters with the identity local operator regarded as the end of the symmetry lines.

## 5 (1+1)d $\text{Rep}(S_3)$-symmetric gapped phases

From this section onward, we will apply our general analysis to study (1+1)d gapped phases with non-invertible symmetries. Such symmetries could be divided into two types:

- **Group-Theoretical or Non-Intrinsic**: These are non-invertible symmetries that are obtained from invertible symmetries by gauging them. Mathematically, the associated fusion category $\mathcal{S}$ is Morita equivalent to $\text{Vec}_G^\omega$ for some choice of $(G, \omega)$. Examples are fusion categories $\text{Rep}(G)$ of representations of a finite non-abelian group $G$, which arise after gauging a non-anomalous $G$ symmetry. The objects of $\text{Rep}(G)$ are topological Wilson line defects for the gauged $G$ symmetry.

- **Non-Group-Theoretical or Intrinsic**: These are non-invertible symmetries that cannot be obtained by gauging invertible symmetries. The associated fusion category $\mathcal{S}$ is not Morita equivalent to $\text{Vec}_G^\omega$ for any choice of $(G, \omega)$. Examples are the Tambara-Yamagami fusion categories $\text{TY}(\mathbb{Z}_n)$ that we will discuss in detail in the following sections.

The gapped phases for group-theoretical symmetries can all be obtained by gauging gapped phases with (possibly anomalous) group symmetries. The SymTFT for such a non-invertible symmetry $\mathcal{S}$ is the same as for the corresponding group symmetry

$$\mathfrak{Z}(\mathcal{S}) \cong \mathfrak{Z}(\mathsf{Vec}_G^\omega) \,. \tag{233}$$

However, the corresponding symmetry boundaries of the SymTFT are different. Thus, even though the set of generalized charges for the non-invertible symmetry $\mathcal{S}$ are the same as for the corresponding group symmetry

$$\mathcal{Z}(\mathcal{S}) \cong \mathcal{Z}(\mathsf{Vec}_G^\omega) \,, \tag{234}$$

the precise structure of the multiplets of local operators carrying these generalized charges is in general different.

In this section, we discuss in detail the gapped phases for the simplest group-theoretical non-invertible symmetry

$$\mathcal{S} = \mathsf{Rep}(S_3) \,, \tag{235}$$

arising from the simplest non-abelian group $S_3$. In the next sections, we discuss in detail the gapped phases for non-group-theoretical or intrinsic non-invertible symmetries of the form $\mathcal{S} = \mathsf{TY}(\mathbb{Z}_n)$. The Lagrangian algebras to be considered in this section are the same of course as in (193).

## 5.1 Symmetry $\mathsf{Rep}(S_3)$

### 5.1.1 Line operators

Let us begin by discussing the structure of the $\mathsf{Rep}(S_3)$ symmetry. It involves two non-trivial simple objects

$$\Big| \qquad \text{and} \qquad \Big| \tag{236}$$
$$P \qquad\qquad\qquad E$$

corresponding respectively to the sign and the standard 2d irreducible representations of $S_3$. $P$ generates an invertible $\mathbb{Z}_2$ subsymmetry

$$\Big|\ \Big| \quad = \quad \Big| \tag{237}$$
$$P\ \ P \qquad\qquad \mathbf{1}$$

while $E$ generates a non-invertible symmetry

$$\Big|\ \Big| \quad = \quad \Big| \quad + \quad \Big| \quad + \quad \Big| \tag{238}$$
$$E\ \ E \qquad\qquad \mathbf{1} \qquad P \qquad E$$

while the composition of the two symmetries is

$$\Big|\ \Big| \quad = \quad \Big|\ \Big| \quad = \quad \Big| \tag{239}$$
$$P\ \ E \qquad\qquad E\ \ P \qquad\qquad E$$

These fusion rules follow from the tensor product decomposition rules for the corresponding representations.

### 5.1.2 Junctions of line operators

Given the above fusion rules, we have the following junctions between the line operators (up to rotations)

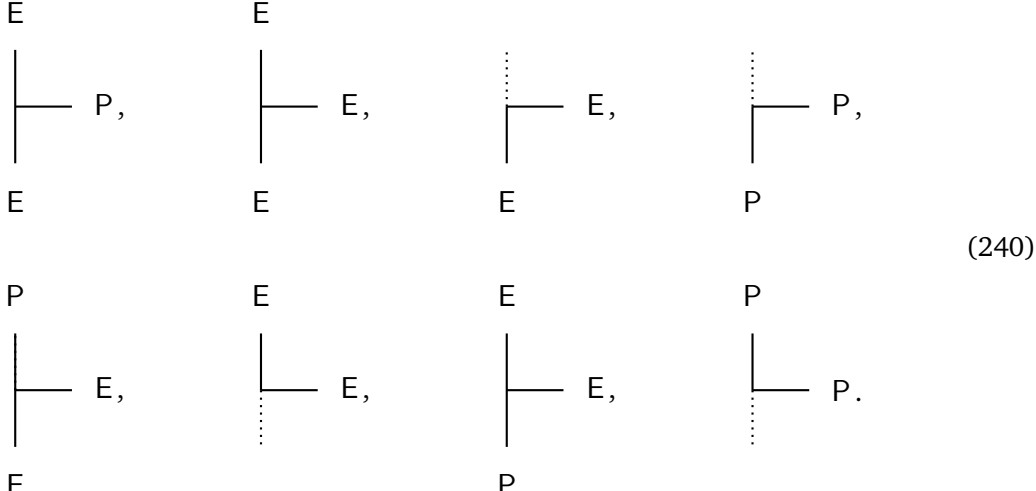

$$\tag{240}$$

Some of the fusion rules for these junctions that we will use are collected in appendix A. We also define quadrivalent junctions by composing these trivalent junctions as

$$\tag{241}$$

## 5.2 Generalized charges

Now let us describe the generalized charges that can be carried by order parameters for gapped phases with $\mathrm{Rep}(S_3)$ symmetry. Note that these are only a subset of all generalized charges for $\mathrm{Rep}(S_3)$ symmetry, corresponding to the subset of simple bulk lines of the SymTFT,

$$\mathfrak{Z}(\mathrm{Rep}(S_3)) = \mathfrak{Z}(\mathrm{Vec}_{S_3}), \tag{242}$$

that can appear in Lagrangian algebras in the Drinfeld center

$$\mathcal{Z}(\mathrm{Rep}(S_3)) = \mathcal{Z}(\mathrm{Vec}_{S_3}). \tag{243}$$

These bulk lines and the Lagrangian algebras have been discussed in section 4.6.

### 5.2.1 $Q_{[\mathrm{id}],P}$ multiplet

This multiplet consists of a single local operator $\mathcal{O}^P$ in the twisted sector for $P \in \mathrm{Rep}(S_3)$

$$P \quad\underline{\hspace{2cm}}\bullet \atop \mathcal{O}^P \tag{244}$$

The action of $\mathrm{Rep}(S_3)$ is

$$\tag{245}$$

Derivations of this and similar actions below are provided in appendix A.

### 5.2.2 $Q_{[a],1}$ multiplet

This multiplet consists of two local operators $\mathcal{O}^a_\pm$, where $\mathcal{O}^a_+$ is in the untwisted sector and $\mathcal{O}^a_-$ is in the twisted sector for $P \in \mathrm{Rep}(S_3)$

$$\bullet \atop \mathcal{O}^a_+ \quad , \qquad P \quad\underline{\hspace{2cm}}\bullet \atop \mathcal{O}^a_- \quad . \tag{246}$$

The action of $P$ is

$$\tag{247}$$

and the action of $E$ is (here $\omega = e^{2\pi i/3}$)

$$\tag{248}$$

From the above actions, we can determine the linking actions of P and E on $\mathcal{O}_+^a$ to be

$$\tag{249}$$

and

$$\tag{250}$$

where the second term on the right-hand side vanishes because there are no topological local operators in $\text{Rep}(S_3)$ converting the line P into the identity line.

### 5.2.3 $Q_{[\text{id}],E}$ multiplet

This multiplet consists of two local operators $\mathcal{O}_\pm^E$, where $\mathcal{O}_+^E$ is in the twisted sector for E and $\mathcal{O}_-^E$ converts E into P

$$\text{E} \longrightarrow \bullet_{\mathcal{O}_+^E} \, , \qquad \text{E} \longrightarrow \bullet_{\mathcal{O}_-^E} \text{P} . \tag{251}$$

The action of P is

$$\tag{252}$$

The action of E is

$$\tag{253}$$

and

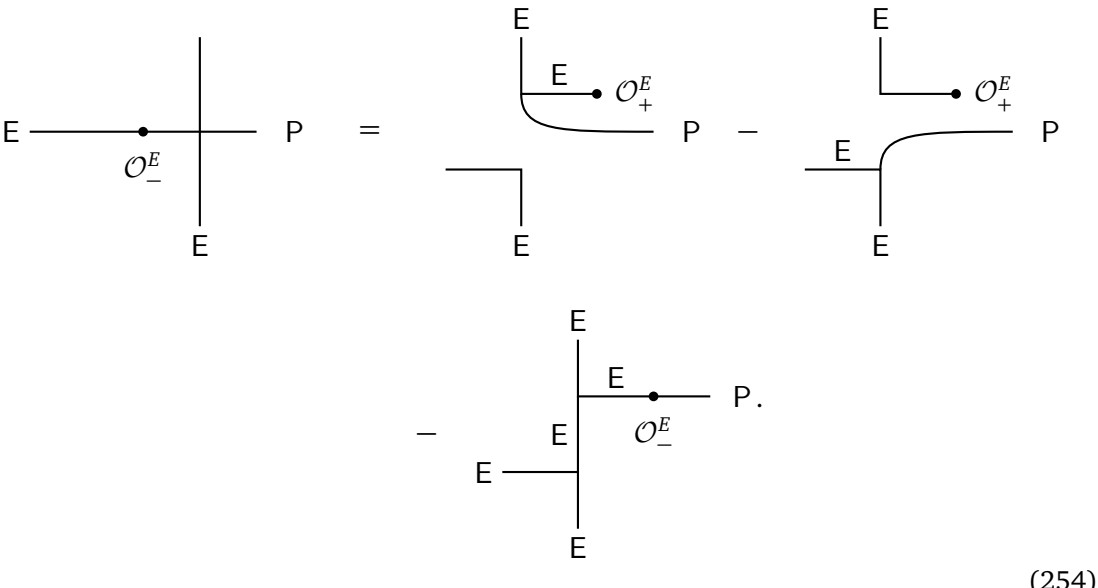

$$(254)$$

### 5.2.4  $Q_{[b],+}$ multiplet

This multiplet consists of three local operators $\mathcal{O}^b, \mathcal{O}^b_\pm$, where $\mathcal{O}^b$ is in the untwisted sector, $\mathcal{O}^b_+$ is in the twisted sector for E, and $\mathcal{O}^b_-$ converts E into P

$$\begin{array}{ccc} \bullet \\ \mathcal{O}^b \end{array} , \qquad E \longrightarrow \begin{array}{c} \bullet \\ \mathcal{O}^b_+ \end{array} , \qquad E \longrightarrow \begin{array}{c} \bullet \\ \mathcal{O}^b_- \end{array} P. \tag{255}$$

The action of P is

$$\begin{array}{ccc} \bullet \\ \mathcal{O}^b \Big| \\ P \end{array} = - \begin{array}{c} \Big| \; \bullet \\ \mathcal{O}^b \\ P \end{array} ,$$

$$E \longrightarrow \begin{array}{c} \bullet \\ \mathcal{O}^b_+ \Big| \\ P \end{array} = E \longrightarrow \begin{array}{c} \bullet \\ \mathcal{O}^b_+ \\ P \end{array} , \tag{256}$$

$$E \longrightarrow \begin{array}{c} \bullet \\ \mathcal{O}^b_- \Big| \end{array} P = E \longrightarrow \begin{array}{c} \bullet \\ \mathcal{O}^b_- \\ P \end{array} P.$$

The action of E on $\mathcal{O}^b$ is

$$\begin{array}{ccc} \bullet \\ \mathcal{O}^b \Big| \\ E \end{array} = \begin{array}{c} \Big| \; \dfrac{E}{\;\;} \bullet \\ \mathcal{O}^b_+ \\ E \end{array} , \tag{257}$$

on $\mathcal{O}_+^b$ is

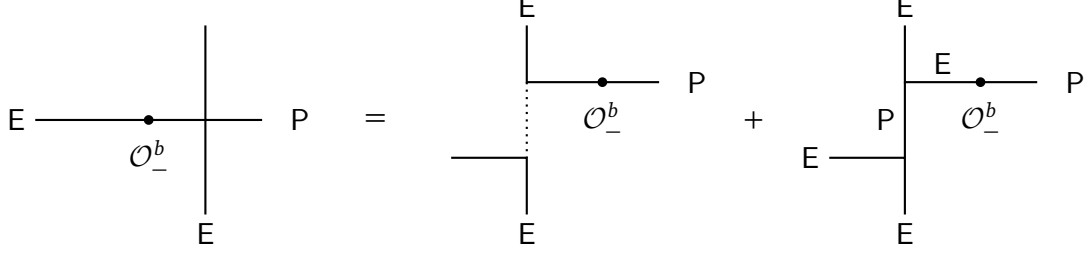

$$ \tag{258} $$

and on $\mathcal{O}_-^b$ is

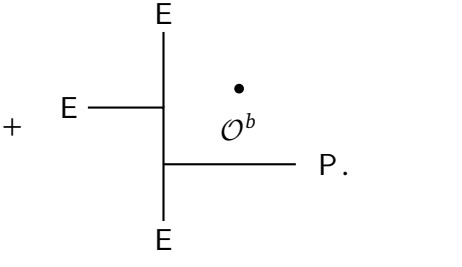

$$ \tag{259} $$

From the above, we can determine the linking actions of P and E on $\mathcal{O}^b$ to be

$$ \tag{260} $$

## 5.3 Gapped phases

Let us now systematically study all the irreducible (1+1)d gapped phases with $\text{Rep}(S_3)$ symmetry. In this subsection, we will use the SymTFT construction of these phases in order to understand their structure. An alternative approach is developed in appendix C, where $\text{Rep}(S_3)$-symmetric phases are obtained from $S_3$-symmetric phases by gauging the $S_3$ symmetry.

As noted above, the SymTFT (242) was already discussed in section 4.6. The symmetry boundary for $\text{Rep}(S_3)$ is

$$ B_{\text{Rep}(S_3)}^{\text{sym}} = \mathcal{A}_{\text{Neu}}, \tag{261} $$

in the list (193). Choosing various physical boundaries gives rise to the various $\text{Rep}(S_3)$-symmetric phases, again as in (193). Note that we could have equivalently chosen the $\text{Rep}(S_3)$ symmetry boundary to be $\mathcal{A}_{\text{Neu}(\mathbb{Z}_2)}$.

### 5.3.1 Trivial phase

First, consider choosing

$$\mathfrak{B}^{\text{phys}} = \mathcal{A}_{\text{Dir}}. \tag{262}$$

In this instance, we obtain a single untwisted local operator coming from the bulk line $\boldsymbol{Q}_{[\text{id}],1}$, which is the only line allowed to completely end on both boundaries.

$$
\begin{array}{cc}
\mathcal{A}_{\text{Neu}} & \mathcal{A}_{\text{Dir}} \\
\mid \quad \boldsymbol{Q}_{[\text{id}],1} \quad \mid & \\
\end{array}
\tag{263}
$$

Thus, the resulting phase has a single vacuum.

The linking actions of P and E on the identity local operator are

$$
\begin{array}{ccc}
\text{P} \bigcirc & = & 1, \\
& & \\
\text{E} \bigcirc & = & 2.
\end{array}
\tag{264}
$$

using which we can identify these lines as

$$
\begin{aligned}
D_1^{(\text{P})} &\cong \boldsymbol{1}, \\
D_1^{(\text{E})} &\cong \boldsymbol{1} \oplus \boldsymbol{1},
\end{aligned}
\tag{265}
$$

where $\boldsymbol{1}$ is the identity line operator. We refer to this phase as the trivial phase as it corresponds to the trivial functor $\varphi_{\text{triv}}^{(1)}$ discussed around (51)

There are two types of order parameters that coexist for this phase. Both of these are of string-type. These are multiplets carrying generalized charges $\boldsymbol{Q}_{[\text{id}],P}$ and $\boldsymbol{Q}_{[\text{id}],E}$. In such a multiplet, the IR image of the UV operators $\mathcal{O}^P$ and $\mathcal{O}_{\pm}^E$ can all be identified with the identity local operators regarded to be lying at the end of lines $D_1^{(P)}$ and $D_1^{(E)}$.

### 5.3.2 $\mathbb{Z}_2$ SSB phase

Let us now consider

$$\mathfrak{B}^{\text{phys}} = \mathcal{A}_{\text{Neu}(\mathbb{Z}_2)}. \tag{266}$$

The compactifications of bulk lines are

$$
\begin{array}{cc}
\mathcal{A}_{\text{Neu}} & \mathcal{A}_{\text{Neu}(\mathbb{Z}_2)} \\
\mid \quad \boldsymbol{Q}_{[\text{id}],1} \quad \mid & \\
\mid \quad \boldsymbol{Q}_{[b],+} \quad \mid & \\
\end{array}
\tag{267}
$$

We thus have 2 untwisted sector local operators, and hence the resulting phase has 2 vacua. Following the terminology introduced above, we refer to the untwisted sector local operator descending from the bulk line $\boldsymbol{Q}_{[b],+}$ as $\mathcal{O}^b$.

In order to precisely identify the two vacua, we need to determine the algebra formed by the untwisted local operators. In the current case, this amounts to determining $\mathcal{O}^b \mathcal{O}^b$ as a linear combination of $\mathcal{O}^b$ and the identity local operator $1$

$$\mathcal{O}^b \mathcal{O}^b = \alpha + \beta \mathcal{O}^b, \qquad (\alpha, \beta) \in \mathbb{C}^2 - \{(0,0)\}. \tag{268}$$

Recall that $\mathcal{O}^b$ is charged non-trivially under the $\mathbb{Z}_2$ subsymmetry of $\mathsf{Rep}(S_3)$ generated by $\mathsf{P}$. Consequently, we can set $\beta = 0$, leaving us with

$$\mathcal{O}^b \mathcal{O}^b = \alpha, \qquad \alpha \in \mathbb{C} - \{0\}. \tag{269}$$

Now rescaling $\mathcal{O}^b$, we can set the above relation to

$$\mathcal{O}^b \mathcal{O}^b = 1. \tag{270}$$

Using this, we can express the two vacua as

$$\begin{aligned}
v_0 &= \frac{1 + \mathcal{O}^b}{2}, \\
v_1 &= \frac{1 - \mathcal{O}^b}{2}.
\end{aligned} \tag{271}$$

The linking action of $D_1^{(\mathsf{P})}$ is

$$\begin{aligned}
v_0 &\to v_1, \\
v_1 &\to v_0,
\end{aligned} \tag{272}$$

and hence we can identify $D_1^{(\mathsf{P})}$ as

$$D_1^{(\mathsf{P})} \cong \mathbf{1}_{01} \oplus \mathbf{1}_{10}. \tag{273}$$

Note that there is no relative Euler term between the two vacua as the linking actions of $\mathbf{1}_{01}$ and $\mathbf{1}_{10}$ are

$$\begin{aligned}
\mathbf{1}_{01}: \quad v_0 &\to v_1, \\
\mathbf{1}_{10}: \quad v_1 &\to v_0,
\end{aligned} \tag{274}$$

without any extra factors. Since this $\mathbb{Z}_2$ subsymmetry is spontaneously broken in both vacua, we refer to this phase as the "$\mathbb{Z}_2$ **SSB phase**".

Using the linking action of $D_1^{(\mathsf{E})}$ on $\mathcal{O}^b$ described in (260), we determine the linking action of $D_1^{(\mathsf{E})}$ on both vacua to be

$$v_i \to 1 = v_0 + v_1, \qquad i \in \{0,1\}, \tag{275}$$

and hence we can identify $D_1^{(\mathsf{E})}$ as

$$D_1^{(\mathsf{E})} \cong \mathbf{1}_{00} \oplus \mathbf{1}_{01} \oplus \mathbf{1}_{10} \oplus \mathbf{1}_{11} \cong D_1^{(1)} \oplus D_1^{(\mathsf{P})}. \tag{276}$$

Note that this implies that the non-invertible symmetry $\mathsf{E}$ is spontaneously broken in both vacua: beginning with the presence of $v_0$ and acting by $\mathsf{E}$ we obtain $v_0 + v_1$, thus inferring the presence of another vacuum $v_1$. However, the action of $\mathsf{E}$ does not distinguish between the two vacua. Thus, the two vacua are physically indistinguishable even though we have spontaneous breaking of a non-invertible symmetry. This is tied to the fact that there are no relative Euler terms between the two vacua.

From the Lagrangian algebra $\mathcal{A}_{\mathrm{Neu}(\mathbb{Z}_2)}$, we see that there are two types of order parameters associated to this phase:

- The first type of order parameters have generalized charge $\boldsymbol{Q}_{[\text{id}],E}$. Such an order parameter can be considered to be a purely string type order parameter, as it consists of a multiplet of two local operators, both of which are in non-trivial twisted sectors of the Rep($S_3$) symmetry. The IR image of such an order parameter is a multiplet of two local operators in the IR TQFT: one of them, $\mathcal{O}_+^E$, is the identity local operator, regarded as the endpoint of line $D_1^{(E)}$

$$\mathcal{O}_+^E = 1 \,, \tag{277}$$

  while the other, $\mathcal{O}_-^E$, is the identity local operator $1_P$ on $D_1^{(P)}$, regarded as an operator converting the line $D_1^{(E)}$ to the line $D_1^{(P)}$

$$\mathcal{O}_-^E = 1_P \,. \tag{278}$$

- The second type of order parameters have generalized charge $\boldsymbol{Q}_{[b],+}$. Such an order parameter is a mixture of conventional and string type order parameters. It consists of a multiplet of three local operators, one of which is in the untwisted sector, while the other two are in non-trivial twisted sectors of the Rep($S_3$) symmetry. The IR image of such an order parameter is a multiplet of three local operators in the IR TQFT: one of them is the untwisted sector operator $\mathcal{O}^b$ discussed above, another one, $\mathcal{O}_+^b$, is the operator $\mathcal{O}^b$ regarded as the endpoint of line $D_1^{(E)}$

$$\mathcal{O}_+^b = \mathcal{O}^b \,, \tag{279}$$

  while the third one, $\mathcal{O}_-^b$, is the operator $\mathcal{O}^b \otimes 1_P$ living on $D_1^{(P)}$, regarded as an operator converting the line $D_1^{(E)}$ to the line $D_1^{(P)}$

$$\mathcal{O}_-^b = \mathcal{O}^b \otimes 1_P \,. \tag{280}$$

### 5.3.3 Rep($S_3$)/$\mathbb{Z}_2$ SSB phase

Let us now choose

$$\mathfrak{B}^{\text{phys}} = \mathcal{A}_{\text{Neu}(\mathbb{Z}_3)} \,. \tag{281}$$

From the coefficients of various bulk lines in the Lagrangian algebras $\mathcal{A}_{\text{Neu}(\mathbb{Z}_3)}$ and $\mathcal{A}_{\text{Neu}}$, we see that $\boldsymbol{Q}_{[a],1}$ has two ends along $\mathcal{A}_{\text{Neu}(\mathbb{Z}_3)}$, but a single end along $\mathcal{A}_{\text{Neu}}$. We have thus two interval compactifications of the line $\boldsymbol{Q}_{[a],1}$

$$
\begin{array}{cc}
\mathcal{A}_{\text{Neu}} & \mathcal{A}_{\text{Neu}(\mathbb{Z}_3)} \\
\end{array}
\tag{282}
$$

In total, we have 3 untwisted sector local operators, and hence the resulting phase has 3 vacua. We refer to the untwisted sector local operators descending from the bulk line $\boldsymbol{Q}_{[a],1}$ as $\mathcal{O}_{+,1}^a$ and $\mathcal{O}_{+,2}^a$.

Let us now determine the algebra formed by these untwisted local operators. For this purpose, it turns out to be very useful to track the action of the $\mathbb{Z}_3$ subgroup of the $S_3$ symmetry localized on the physical boundary $\mathfrak{B}^{\text{phys}}$. This $\mathbb{Z}_3$ acts as

$$
\begin{aligned}
\mathcal{O}_{+,1}^a &\to \omega \, \mathcal{O}_{+,1}^a \,, \\
\mathcal{O}_{+,2}^a &\to \omega^2 \, \mathcal{O}_{+,2}^a \,,
\end{aligned}
\tag{283}
$$

forcing the algebra to take the form

$$
\begin{aligned}
\mathcal{O}^a_{+,1}\mathcal{O}^a_{+,1} &= \alpha\,\mathcal{O}^a_{+,2}\,,\\
\mathcal{O}^a_{+,2}\mathcal{O}^a_{+,2} &= \beta\,\mathcal{O}^a_{+,1}\,,\\
\mathcal{O}^a_{+,1}\mathcal{O}^a_{+,2} &= \gamma\,,
\end{aligned}
\tag{284}
$$

for some $\alpha,\beta,\gamma \in \mathbb{C}^\times$. Rescaling $\mathcal{O}^a_{+,1}$ and $\mathcal{O}^a_{+,2}$, we can set the above algebra into the form

$$
\begin{aligned}
\mathcal{O}^a_{+,1}\mathcal{O}^a_{+,1} &= \mathcal{O}^a_{+,2}\,,\\
\mathcal{O}^a_{+,2}\mathcal{O}^a_{+,2} &= \mathcal{O}^a_{+,1}\,,\\
\mathcal{O}^a_{+,1}\mathcal{O}^a_{+,2} &= \gamma\,.
\end{aligned}
\tag{285}
$$

The associative nature of the algebra further sets $\gamma = 1$, leading to the final form of the algebra

$$
\begin{aligned}
\mathcal{O}^a_{+,1}\mathcal{O}^a_{+,1} &= \mathcal{O}^a_{+,2}\,,\\
\mathcal{O}^a_{+,2}\mathcal{O}^a_{+,2} &= \mathcal{O}^a_{+,1}\,,\\
\mathcal{O}^a_{+,1}\mathcal{O}^a_{+,2} &= 1\,.
\end{aligned}
\tag{286}
$$

This determines the three vacua to be

$$
v_i = \frac{1 + \omega^i\,\mathcal{O}^a_{+,1} + \omega^{2i}\,\mathcal{O}^a_{+,2}}{3}\,, \qquad i \in \{0,1,2\}\,,
\tag{287}
$$

where $\omega$ is a third root of unity.

Given that the linking action of $D_1^{(P)}$ on $\mathcal{O}^a_{+,1}$ and $\mathcal{O}^a_{+,2}$ is trivial (see equation (249)), we learn that the symmetry P leaves each vacuum invariant, and hence we can simply identify the line $D_1^{(P)}$ with the identity line $\mathbf{1}$

$$
D_1^{(P)} \cong \mathbf{1}_{00} \oplus \mathbf{1}_{11} \oplus \mathbf{1}_{22} \cong \mathbf{1}\,.
\tag{288}
$$

Since the $\mathbb{Z}_2$ subsymmetry of $\mathrm{Rep}(S_3)$ is spontaneously unbroken in all vacua, but we still have multiple vacua, we refer to this phase as the **$\mathrm{Rep}(S_3)/\mathbb{Z}_2$ SSB phase**.

On the other hand, the linking action of $D_1^{(E)}$ (see (250)) is

$$
\begin{aligned}
1 &\to 2\,,\\
\mathcal{O}^a_{+,1} &\to -\mathcal{O}^a_{+,1}\,,\\
\mathcal{O}^a_{+,2} &\to -\mathcal{O}^a_{+,2}\,,
\end{aligned}
\tag{289}
$$

which implies its linking action on the vacua is

$$
\begin{aligned}
v_0 &\to v_1 + v_2\,,\\
v_1 &\to v_2 + v_0\,,\\
v_2 &\to v_0 + v_1\,.
\end{aligned}
\tag{290}
$$

We can thus identify the line $D_1^{(E)}$ as

$$
D_1^{(E)} \cong \mathbf{1}_{01} + \mathbf{1}_{02} + \mathbf{1}_{12} + \mathbf{1}_{10} + \mathbf{1}_{20} + \mathbf{1}_{21}\,,
\tag{291}
$$

such that there are no relative Euler terms between the three vacua, i.e. the linking action of $\mathbf{1}_{ij}$ is

$$
v_i \to v_j\,,
\tag{292}
$$

without any extra factors. This suggests that we again are in a situation where we have spontaneous breaking of a non-invertible symmetry (in this case the symmetry E), but all vacua are still physically indistinguishable. Indeed, E acts by sending a vacuum to a sum of the other two vacua, and thus the action of E does not distinguish between the three vacua.

From the Lagrangian algebra $\mathcal{A}_{\text{Neu}(\mathbb{Z}_3)}$, we see that there are two types of order parameters for this phase: one type having generalized charge $\boldsymbol{Q}_{[\text{id}],P}$ which is purely a string type order parameter, and the other type having generalized charge $\boldsymbol{Q}_{[a],1}$ which is a mixture of conventional and string type order parameters. In the IR, we have a single multiplet with generalized charge $\boldsymbol{Q}_{[\text{id}],P}$, but two distinct multiplets with generalized charge $\boldsymbol{Q}_{[a],1}$. Thus, an order parameter with charge $\boldsymbol{Q}_{[\text{id}],P}$ has a unique IR image, but an order parameter with charge $\boldsymbol{Q}_{[a],1}$ realizes one of the two possible IR images, which may be distinct for different UV order parameters having the same charge $\boldsymbol{Q}_{[a],1}$. The IR multiplets with these charges are discussed below:

- The IR multiplet with charge $\boldsymbol{Q}_{[\text{id}],P}$ comprises of a single local operator

$$\mathcal{O}^P = 1, \tag{293}$$

where the identity local operator 1 is regarded as an operator living at the endpoint of a line operator $D_1^{(P)} \cong \boldsymbol{1}$.

- One of the IR multiplets with charge $\boldsymbol{Q}_{[a],1}$ comprises of two local operators $(\mathcal{O}_{+,1}^a, \mathcal{O}_{-,1}^a)$, where the operator $\mathcal{O}_{+,1}^a$ was discussed above and

$$\mathcal{O}_{-,1}^a = \mathcal{O}_{+,1}^a, \tag{294}$$

where $\mathcal{O}_{+,1}^a$ appearing on the RHS is regarded as an operator living at the endpoint of a line operator $D_1^{(P)} \cong \boldsymbol{1}$.

- The other IR multiplet with charge $\boldsymbol{Q}_{[a],1}$ comprises of two local operators $(\mathcal{O}_{+,2}^a, \mathcal{O}_{-,2}^a)$, where the operator $\mathcal{O}_{+,2}^a$ was discussed above and

$$\mathcal{O}_{-,2}^a = -\mathcal{O}_{+,2}^a, \tag{295}$$

where $-\mathcal{O}_{+,2}^a$ appearing on the RHS is regarded as an operator living at the endpoint of a line operator $D_1^{(P)} \cong \boldsymbol{1}$.

The equations (248) are satisfied by both $(\mathcal{O}_{+,1}^a, \mathcal{O}_{-,1}^a)$ and $(\mathcal{O}_{+,2}^a, \mathcal{O}_{-,2}^a)$. This can be seen by using the trivalent junctions

$$
\begin{array}{c}
\text{E} \\
\vdash\!\!-\!\!-\ \text{P} \ = \ \bigg|\ -\ \bigg|\ +\ \bigg|\ -\ \bigg|\ +\ \bigg|\ -\ \bigg| \\
\text{E} \qquad \mathbf{1}_{01} \quad \mathbf{1}_{21} \quad \mathbf{1}_{12} \quad \mathbf{1}_{02} \quad \mathbf{1}_{20} \quad \mathbf{1}_{10}
\end{array}
\tag{296}
$$

and

$$
\begin{array}{c}
\text{E} \\
\text{P} \ -\!\!-\!\!-\!\dashv \ = \ \bigg|\ -\ \bigg|\ +\ \bigg|\ -\ \bigg|\ +\ \bigg|\ -\ \bigg| \\
\text{E} \qquad \mathbf{1}_{10} \quad \mathbf{1}_{12} \quad \mathbf{1}_{21} \quad \mathbf{1}_{20} \quad \mathbf{1}_{02} \quad \mathbf{1}_{01}
\end{array}
\tag{297}
$$

Combining the above two, the top quadrivalent junction in (241) is

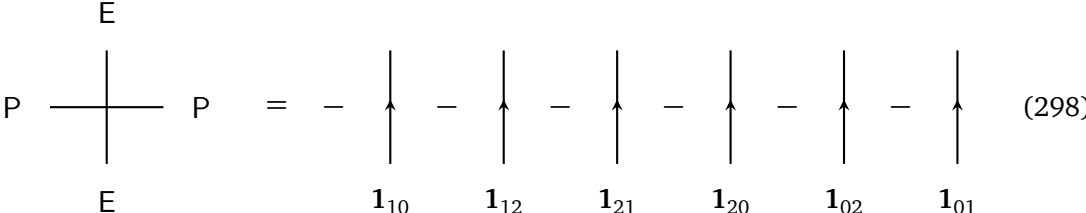

$$\tag{298}$$

The reader can now easily verify that the equations (248) are satisfied.

### 5.3.4  Rep($S_3$) SSB phase

Let us now consider

$$\mathfrak{B}^{\text{phys}} = \mathcal{A}_{\text{Neu}}. \tag{299}$$

The resulting phase has 3 untwisted local operators, and hence 3 vacua

$$\tag{300}$$

We refer to the untwisted local operators arising from $Q_{[a],1}$ and $Q_{[b],+}$ respectively as $\mathcal{O}^a_+$ and $\mathcal{O}^b$.

Let us now determine the algebra formed by these untwisted local operators. Since the fusion

$$Q_{[b],+} \otimes Q_{[a],1}, \tag{301}$$

of the bulk line operators does not contain the identity bulk line operator or the $Q_{[a],1}$ bulk line operator, we must have

$$\mathcal{O}^a_+ \mathcal{O}^b = \mathcal{O}^b, \tag{302}$$

where we have rescaled $\mathcal{O}^a_+$ to ensure that there is no non-trivial coefficient on the RHS of the above equation.

Similarly, since the fusion

$$Q_{[a],1} \otimes Q_{[a],1}, \tag{303}$$

does not contain $Q_{[b],+}$, we must have

$$\mathcal{O}^a_+ \mathcal{O}^a_+ = \alpha + (1-\alpha)\mathcal{O}^a_+, \qquad \alpha \in \mathbb{C}, \tag{304}$$

where the relative weight between the two coefficients on the RHS has been set by imposing associativity with (302).

We can fix $\alpha$ by studying the action of E on (304). Acting by E on the LHS of (304), we obtain

$$\tag{305}$$

where $\mathcal{O}_-^a$ is the local operator in the P-twisted sector lying in the same irreducible multiplet as $\mathcal{O}_+^a$. We are going to focus on the middle term involving the product of $\mathcal{O}_-^a$ and $\mathcal{O}_+^a$, which must be proportional to $\mathcal{O}_-^a$. The other two terms involving products $\mathcal{O}_+^a \mathcal{O}_+^a$ and $\mathcal{O}_-^a \mathcal{O}_-^a$ do not give rise to $\mathcal{O}_-^a$. Now, acting by E on the RHS of (304), we obtain

$$
\alpha \quad \Big| \quad - \quad \tfrac{1}{2}(1-\alpha) \quad \Big|_{\mathcal{O}_+^a} \bullet \quad + \quad \left(\omega + \tfrac{1}{2}\right)(1-\alpha) \quad \Big|^{\ P}_{\mathcal{O}_-^a}\bullet \ .
$$

$$\text{E} \qquad\qquad \text{E} \qquad\qquad\qquad \text{E} \tag{306}$$

Matching the $\mathcal{O}_-^a$ contributions of (305) and (306), we obtain

$$
\mathcal{O}_-^a \mathcal{O}_+^a = -(1-\alpha)\mathcal{O}_-^a \, . \tag{307}
$$

Imposing associativity with (304) fixes

$$
\alpha = 2 \, , \qquad \text{or} \qquad \alpha = \frac{1}{2} \, . \tag{308}
$$

For the final product relation, note that the fusion

$$
\boldsymbol{Q}_{[b],+} \otimes \boldsymbol{Q}_{[b],+} \, , \tag{309}
$$

does not contain $\boldsymbol{Q}_{[b],+}$, and so we can express

$$
\mathcal{O}^b \mathcal{O}^b = \beta + \gamma \mathcal{O}_+^a \, , \qquad (\beta, \gamma) \in \mathbb{C}^2 - \{(0,0)\} \, . \tag{310}
$$

Imposing associativity with (302), we obtain $\beta = \alpha\gamma$, and further rescaling $\mathcal{O}^b$ we obtain

$$
\mathcal{O}^b \mathcal{O}^b = \alpha + \mathcal{O}_+^a \, . \tag{311}
$$

We now need to analyze the two cases (308). It turns out that the case $\alpha = 2$ is not consistent, as will be explained later. For now, we focus on the case

$$
\alpha = \frac{1}{2} \, , \tag{312}
$$

which is consistent. In this case, the operator algebra is

$$
\begin{aligned}
\mathcal{O}_+^a \mathcal{O}^b &= \mathcal{O}^b \, , \\
\mathcal{O}_+^a \mathcal{O}_+^a &= \frac{1}{2}(1 + \mathcal{O}_+^a) \, , \\
\mathcal{O}^b \mathcal{O}^b &= \frac{1}{2} + \mathcal{O}_+^a \, .
\end{aligned} \tag{313}
$$

We can compute the vacua to be

$$
\begin{aligned}
v_0 &= \frac{2}{3}\left(1 - \mathcal{O}_+^a\right) , \\
v_1 &= \frac{1}{6}\left(1 + 2\mathcal{O}_+^a + \sqrt{6}\mathcal{O}^b\right) , \\
v_2 &= \frac{1}{6}\left(1 + 2\mathcal{O}_+^a - \sqrt{6}\mathcal{O}^b\right) .
\end{aligned} \tag{314}
$$

The linking action of $D_1^{(\text{P})}$ is

$$
\begin{aligned}
\mathcal{O}_+^a &\to \mathcal{O}_+^a \, , \\
\mathcal{O}^b &\to -\mathcal{O}^b \, ,
\end{aligned} \tag{315}
$$

and hence it acts on the vacua as

$$v_0 \to v_0 \,,$$
$$v_1 \to v_2 \,, \qquad (316)$$
$$v_2 \to v_1 \,.$$

Thus the $\mathrm{Rep}(S_3)$ phase under discussion decomposes as a sum of a $\mathbb{Z}_2$ SSB phase (formed by vacua $v_1$ and $v_2$) and a $\mathbb{Z}_2$ non-SSB phase (formed by vacuum $v_0$) when we restrict our attention only to the $\mathbb{Z}_2$ subsymmetry of the full $\mathrm{Rep}(S_3)$ symmetry. We can identify the line $D_1^{(\mathrm{P})}$ generating this $\mathbb{Z}_2$ subsymmetry as

$$D_1^{(\mathrm{P})} \cong \mathbf{1}_{00} \oplus \mathbf{1}_{12} \oplus \mathbf{1}_{21} \,. \qquad (317)$$

Note that there is no relative Euler term between vacua $v_1$ and $v_2$ as the linking action of $\mathbf{1}_{12}$ is

$$v_1 \to v_2 \,, \qquad (318)$$

without any extra factors. However, there could be relative Euler terms between vacua $v_0$ and $v_1$ or $v_2$. We will soon see that there are indeed such non-trivial relative Euler terms present in this phase. Thus, the vacuum $v_0$ can be physically distinguished from the vacua $v_1$ and $v_2$, while the vacua $v_1$ and $v_2$ are physically indistinguishable. This is also apparent already from the action of the unique $\mathbb{Z}_2$ subsymmetry of $\mathrm{Rep}(S_3)$ on these vacua: it treats $v_0$ differently from $v_1, v_2$.

The linking action of $D_1^{(\mathrm{E})}$ is

$$1 \to 2 \,,$$
$$\mathcal{O}_+^a \to -\mathcal{O}_+^a \,, \qquad (319)$$
$$\mathcal{O}^b \to 0 \,,$$

implying the action on the vacua

$$v_0 \to \frac{2}{3}\left(2 + \mathcal{O}_+^a\right) = v_0 + 2(v_1 + v_2) \,,$$
$$v_1 \to \frac{1}{3}\left(1 - \mathcal{O}_+^a\right) = \frac{1}{2}v_0 \,, \qquad (320)$$
$$v_2 \to \frac{1}{3}\left(1 - \mathcal{O}_+^a\right) = \frac{1}{2}v_0 \,.$$

The presence of fractions on the RHS above is an indication of the presence of relative Euler terms. We can identify

$$D_1^{(\mathrm{E})} \cong \mathbf{1}_{00} \oplus \mathbf{1}_{01} \oplus \mathbf{1}_{02} \oplus \mathbf{1}_{10} \oplus \mathbf{1}_{20} \,, \qquad (321)$$

with non-trivial relative Euler counterterms encoded in the linking actions

$$
\begin{aligned}
\mathbf{1}_{01} : & \quad v_0 \to 2v_1 \,, \\
\mathbf{1}_{02} : & \quad v_0 \to 2v_2 \,, \\
\mathbf{1}_{10} : & \quad v_1 \to \frac{1}{2}v_0 \,, \\
\mathbf{1}_{20} : & \quad v_2 \to \frac{1}{2}v_0 \,.
\end{aligned}
\qquad (322)
$$

We refer to this phase as the **$\mathrm{Rep}(S_3)$ SSB phase**. In particular, the E symmetry is spontaneously broken in all three vacua. Finally, we have encountered an example where spontaneous breaking of non-invertible symmetry leads to physically distinguishable vacua!

Let us now tie the remaining loose end by considering the case $\alpha = 2$ and showing that it is inconsistent. In this case, the algebra of untwisted local operators is

$$
\begin{aligned}
\mathcal{O}_+^a \mathcal{O}^b &= \mathcal{O}^b \,, \\
\mathcal{O}_+^a \mathcal{O}_+^a &= 2 - \mathcal{O}_+^a \,, \\
\mathcal{O}^b \mathcal{O}^b &= 2 + \mathcal{O}_+^a \,,
\end{aligned}
\tag{323}
$$

from which we can compute the vacua to be

$$
\begin{aligned}
v_0 &= \frac{1}{3} \left( 1 - \mathcal{O}_+^a \right) \,, \\
v_1 &= \frac{1}{6} \left( 2 + \mathcal{O}_+^a + \sqrt{3} \mathcal{O}^b \right) \,, \\
v_2 &= \frac{1}{6} \left( 2 + \mathcal{O}_+^a - \sqrt{3} \mathcal{O}^b \right) \,.
\end{aligned}
\tag{324}
$$

The linking action of the line E on the vacua is

$$
\begin{aligned}
v_0 &\to v_1 + v_2 \,, \\
v_1 &\to v_0 + \frac{1}{2}(v_1 + v_2) \,, \\
v_2 &\to v_0 + \frac{1}{2}(v_1 + v_2) \,.
\end{aligned}
\tag{325}
$$

Since the action on $v_1$ produces a fractional value of $v_1$, it is not possible to represent the line E by any line operator in the phase, even if we allow for the presence of non-trivial relative Euler counterterms. Thus the resulting phase does not admit a consistent action of $\mathrm{Rep}(S_3)$ symmetry, leading to a contradiction.

The order parameters for this phase carry generalized charges $\boldsymbol{Q}_{[a],1}$ and $\boldsymbol{Q}_{[b],+}$. An order parameter carrying charge $\boldsymbol{Q}_{[a],1}$ is a mixture of standard and string type order parameters, while an order parameter carrying charge $\boldsymbol{Q}_{[b],+}$ is purely of string type. There are unique possible IR images for both kinds of order parameters. The IR image for charge $\boldsymbol{Q}_{[a],1}$ comprises of an untwisted sector local operator $\mathcal{O}_+^a$ discussed above and a P-twisted sector local operator

$$
\mathcal{O}_-^a = \frac{3v_0}{2(2\omega + 1)} \,,
\tag{326}
$$

regarded as an operator living at the end of $D_1^{(\mathrm{P})}$. The reader can check that $(\mathcal{O}_+^a, \mathcal{O}_-^a)$ transform under $\mathrm{Rep}(S_3)$ as a multiplet with generalized charge $\boldsymbol{Q}_{[a],1}$, by using the junctions



$$
(327)
$$

Similarly, with more effort, an interested reader can find the IR multiplet carrying charge $\boldsymbol{Q}_{[b],+}$. One of the three local operators comprising this multiplet is the operator $\mathcal{O}^b$ discussed above.

# 6  (1+1)d Ising-symmetric gapped phases

From this section onward, we turn our attention to (1+1)d gapped phases for non-group-theoretical non-invertible symmetries. We study a class of such symmetries described by Tambara-Yamagami (TY) fusion categories [136] (see [83] for a physics review). A general TY category is

$$\mathsf{TY}(A, \chi, \tau),\tag{328}$$

where $A$ is an abelian group, $\chi$ is a symmetric non-degenerate bicharacter on $A$ and $\tau$ is a sign. In this work, we will only consider TY categories

$$\mathsf{TY}(\mathbb{Z}_N) := \mathsf{TY}(\mathbb{Z}_N, \chi_{\mathrm{id}}, +),\tag{329}$$

where $A = \mathbb{Z}_N$, the sign $\tau$ is chosen to be trivial, and the bicharacter $\chi = \chi_{\mathrm{id}}$ is specified by

$$\chi_{\mathrm{id}} = \left(e^{2\pi i/N}, e^{2\pi i/N}\right) = e^{2\pi i/N}.\tag{330}$$

We begin in this section by considering the simplest case

$$\mathcal{S} = \mathsf{TY}(\mathbb{Z}_2) \equiv \mathsf{Ising},\tag{331}$$

that we also refer to as the Ising symmetry as it arises in the Ising CFT in 2d. The gapped phases in the Ising case have appeared before in [88] albeit using different methods, whereas the general $\mathsf{TY}(\mathbb{Z}_N)$ case to our knowledge has not been discussed before.

## 6.1  Symmetry

Ising symmetry involves two non-trivial topological line operators

$$\Big|\qquad\text{and}\qquad\Big|\tag{332}$$
$$\mathsf{P}\qquad\qquad\qquad\mathsf{S}$$

with fusion rules

$$\mathsf{P} \otimes \mathsf{P} = \mathbf{1}, \qquad \mathsf{P} \otimes \mathsf{S} = \mathsf{S} = \mathsf{S} \otimes \mathsf{P}, \qquad \mathsf{S} \otimes \mathsf{S} = \mathbf{1} \oplus \mathsf{P}.\tag{333}$$

Thus $\mathsf{P}$ is a $\mathbb{Z}_2$ subsymmetry, and $\mathsf{S}$ generates a non-invertible symmetry,

## 6.2  Generalized charges

The full set of generalized charges for Ising symmetry was discussed in section 4.4.5 of [58], including the structures of the irreducible multiplets carrying these charges and the action of the Ising symmetry on these multiplets. The SymTFT is the 3d TQFT carrying modular fusion category,

$$\mathcal{Z}(\mathsf{Ising}) = \mathsf{Ising} \boxtimes \overline{\mathsf{Ising}},\tag{334}$$

of topological line defects. We label the simple lines as

$$\boldsymbol{Q}_{\mathrm{id},\pm}, \qquad \boldsymbol{Q}_{\mathsf{P},\pm}, \qquad \boldsymbol{Q}_{\mathsf{S},\pm 1/16}, \qquad \boldsymbol{Q}_{\mathsf{S},\pm 7/16}, \qquad \boldsymbol{Q}_{\mathrm{id}+\mathsf{P}},\tag{335}$$

with $Q_{\text{id},+}$ being the identity line. The fusion rules of these lines are

$$
\begin{aligned}
Q_{\text{id},-} \otimes Q_{\text{id},-} &\cong Q_{\text{id},+} , \\
Q_{\text{id},-} \otimes Q_{\text{P},\pm} &\cong Q_{\text{P},\mp} , \\
Q_{\text{id},-} \otimes Q_{\text{S},m/16} &\cong Q_{\text{S},-m/16} , && m \in \{\pm 1, \pm 7\} , \\
Q_{\text{P},s} \otimes Q_{\text{P},s'} &\cong Q_{\text{id},ss'} , && s,s' \in \{+,-\} , \\
Q_{\text{P},s} \otimes Q_{\text{S},m/16} &\cong Q_{\text{S},sm/16} , && s \in \{+,-\} , \quad m \in \{1,-7\} , \\
Q_{\text{P},s} \otimes Q_{\text{S},m/16} &\cong Q_{\text{S},-sm/16} , && s \in \{+,-\} , \quad m \in \{-1,7\} , \\
Q_{\text{S},m/16} \otimes Q_{\text{S},n/16} &\cong Q_{\text{id}+\text{P}} , && m \in \{1,-7\} , \quad n \in \{-1,7\} , \\
Q_{\text{S},m/16} \otimes Q_{\text{S},n/16} &\cong Q_{\text{id},(-1)^{(n-m)/8}} \oplus Q_{\text{P},(-1)^{(n-m)/8}} , && m,n \in \{1,-7\} , \\
Q_{\text{S},m/16} \otimes Q_{\text{S},n/16} &\cong Q_{\text{id},(-1)^{(n-m)/8}} \oplus Q_{\text{P},-(-1)^{(n-m)/8}} , && m,n \in \{-1,7\} , \\
Q_{\text{id},\pm} \otimes Q_{\text{id}+\text{P}} &\cong Q_{\text{id}+\text{P}} , \\
Q_{\text{P},\pm} \otimes Q_{\text{id}+\text{P}} &\cong Q_{\text{id}+\text{P}} , \\
Q_{\text{S},m/16} \otimes Q_{\text{id}+\text{P}} &\cong Q_{\text{S},-1/16} \oplus Q_{\text{S},7/16} , && m \in \{1,-7\} , \\
Q_{\text{S},m/16} \otimes Q_{\text{id}+\text{P}} &\cong Q_{\text{S},1/16} \oplus Q_{\text{S},-7/16} , && m \in \{-1,7\} , \\
Q_{\text{id}+\text{P}} \otimes Q_{\text{id}+\text{P}} &\cong Q_{\text{id},+} \oplus Q_{\text{id},-} \oplus Q_{\text{P},+} \oplus Q_{\text{P},-} .
\end{aligned}
\tag{336}
$$

In this subsection, we review the structure of a subset of the generalized charges that can be carried by order parameters for gapped phases. These correspond to simple bulk lines labeled

$$
Q_{\text{id},-} \qquad \text{and} \qquad Q_{\text{id}+\text{P}} .
\tag{337}
$$

### 6.2.1 $Q_{\text{id},-}$ multiplet

This multiplet consists of a single untwisted sector local operator $\mathcal{O}^-$ on which the action of Ising is

$$
\tag{338}
$$

### 6.2.2 $Q_{\text{id}+\text{P}}$ multiplet

This multiplet consists of two local operators $\mathcal{O}$ and $\mathcal{O}^{\text{P}}$, where $\mathcal{O}$ is in the untwisted sector and $\mathcal{O}^{\text{P}}$ is in the twisted sector for $\text{P} \in$ Ising

$$
\bullet \quad \text{and} \quad \text{P} \longrightarrow \bullet
$$
$$
\mathcal{O} \qquad\qquad\qquad \mathcal{O}^{\text{P}} .
\tag{339}
$$

The action of P is

$$\tag{340}$$

and the action of S is

$$\tag{341}$$

## 6.3  Gapped phases

The SymTFT $\mathfrak{Z}(\mathsf{Ising})$ admits only a single irreducible topological boundary condition (up to isomorphisms). Arguments for this would be provided in the next section where we discuss irreducible topological boundary conditions for general $\mathfrak{Z}(\mathsf{TY}(\mathbb{Z}_N))$.

The Lagrangian algebra corresponding to this unique topological boundary condition takes the form

$$\mathcal{A}_{\mathrm{Dir}} = \mathbf{Q}_{\mathrm{id},+} \oplus \mathbf{Q}_{\mathrm{id},-} \oplus \mathbf{Q}_{\mathrm{id}+\mathrm{P}} \, . \tag{342}$$

Note that this boundary must serve as the symmetry boundary for Ising symmetry

$$\mathsf{B}^{\mathrm{sym}}_{\mathsf{Ising}} = \mathcal{A}_{\mathrm{Dir}} \, . \tag{343}$$

### 6.3.1  Ising **SSB phase**

Since we have only one possible irreducible topological boundary of the SymTFT $\mathfrak{Z}(\mathsf{Ising})$, there is a single irreducible (1+1)d gapped phase with Ising symmetry, corresponding to choosing the physical boundary to be

$$\mathfrak{B}^{\mathrm{phys}} = \mathcal{A}_{\mathrm{Dir}} \, . \tag{344}$$

We have the following compactifications of bulk lines

$$\tag{345}$$

That is, the resulting 2d TQFT has 3 untwisted sector local operators, and hence **three vacua**. We label the untwisted local operator descending from $\mathbf{Q}_{\mathrm{id},-}$ as $\mathcal{O}^-$, and the untwisted local operator descending from $\mathbf{Q}_{\mathrm{id}+\mathrm{P}}$ as $\mathcal{O}$.

Let us now determine the algebra formed by these untwisted local operators. Using the bulk fusion rule,

$$\boldsymbol{Q}_{\text{id},-} \otimes \boldsymbol{Q}_{\text{id},-} \cong \boldsymbol{Q}_{\text{id},+} \,, \tag{346}$$

we can fix

$$\mathcal{O}^- \mathcal{O}^- = 1 \,, \tag{347}$$

where we have used a rescaling of $\mathcal{O}^-$ to fix the coefficient on the right-hand side. Using the bulk fusion rule

$$\boldsymbol{Q}_{\text{id},-} \otimes \boldsymbol{Q}_{\text{id}+\text{P}} \cong \boldsymbol{Q}_{\text{id}+\text{P}} \,, \tag{348}$$

we can fix

$$\mathcal{O}^- \mathcal{O} = \alpha \mathcal{O} \,. \tag{349}$$

Associativity with (347) imposes $\alpha^2 = 1$, and we can further fix $\alpha = 1$ by rescaling $\mathcal{O}^-$ by a sign, leading to

$$\mathcal{O}^- \mathcal{O} = \mathcal{O} \,. \tag{350}$$

Finally, bulk fusion

$$\boldsymbol{Q}_{\text{id}+\text{P}} \otimes \boldsymbol{Q}_{\text{id}+\text{P}} \tag{351}$$

contains both the identity bulk line and $\boldsymbol{Q}_{\text{id},-}$, but not $\boldsymbol{Q}_{\text{id}+\text{P}}$, implying a product rule of the form

$$\mathcal{O}\mathcal{O} = \beta + \gamma \mathcal{O}^- \,. \tag{352}$$

Imposing associativity with (347) fixes $\beta = \gamma$, and we can further fix $\beta = 1$ by rescaling $\mathcal{O}$, leading to the final form for the algebra of the untwisted local operators

$$\begin{aligned} \mathcal{O}^- \mathcal{O}^- &= 1 \,, \\ \mathcal{O}^- \mathcal{O} &= \mathcal{O} \,, \\ \mathcal{O}\mathcal{O} &= 1 + \mathcal{O}^- \,. \end{aligned} \tag{353}$$

Using this algebra, one can determine the three vacua to be

$$\begin{aligned} v_0 &= \frac{1}{2}(1 - \mathcal{O}^-) \,, \\ v_1 &= \frac{1}{4}(1 + \mathcal{O}^- + \sqrt{2}\mathcal{O}) \,, \\ v_2 &= \frac{1}{4}(1 + \mathcal{O}^- - \sqrt{2}\mathcal{O}) \,. \end{aligned} \tag{354}$$

The linking action of $D_1^{(\text{P})}$ is

$$\begin{aligned} \mathcal{O}^- &\to \mathcal{O}^- \,, \\ \mathcal{O} &\to -\mathcal{O} \,, \end{aligned} \tag{355}$$

and hence acts on the vacua as

$$\begin{aligned} v_0 &\to v_0 \,, \\ v_1 &\to v_2 \,, \\ v_2 &\to v_1 \,, \end{aligned} \tag{356}$$

leading to the identification

$$D_1^{(\text{P})} \cong \mathbf{1}_{00} \oplus \mathbf{1}_{12} \oplus \mathbf{1}_{21} \,. \tag{357}$$

Note that analogously to the $\text{Rep}(S_3)$ SSB phase discussed in the previous section, the Ising phase also decomposes as a sum of a $\mathbb{Z}_2$ SSB phase (formed by vacua $v_1$ and $v_2$) and a $\mathbb{Z}_2$ trivial phase (formed by vacuum $v_0$), when we restrict our attention to the $\mathbb{Z}_2$ subsymmetry of

the full Ising symmetry. Similarly to the Rep($S_3$) case, we refer to the phase under discussion as Ising **SSB phase**. Even though the discussions for the Rep($S_3$) SSB and the Ising SSB phases are the same so far, we will see shortly that the relative Euler terms are different for the two phases.

The linking action of $D_1^{(S)}$ is

$$
\begin{aligned}
1 &\to \sqrt{2}\,, \\
\mathcal{O}^- &\to -\sqrt{2}\mathcal{O}^-\,, \\
\mathcal{O} &\to 0\,,
\end{aligned}
\tag{358}
$$

implying the action on the vacua

$$
\begin{aligned}
v_0 &\to \frac{1}{\sqrt{2}}(1+\mathcal{O}^-) = \sqrt{2}(v_1+v_2)\,, \\
v_1, v_2 &\to \frac{1}{2\sqrt{2}}(1-\mathcal{O}^-) = \frac{1}{\sqrt{2}}v_0\,.
\end{aligned}
\tag{359}
$$

We can thus identify

$$
D_1^{(S)} \cong \mathbf{1}_{01} \oplus \mathbf{1}_{02} \oplus \mathbf{1}_{10} \oplus \mathbf{1}_{20}\,,
\tag{360}
$$

with non-trivial relative Euler counterterms encoded in the linking actions

$$
\begin{aligned}
\mathbf{1}_{01}: \quad & v_0 \to \sqrt{2}v_1\,, \\
\mathbf{1}_{02}: \quad & v_0 \to \sqrt{2}v_2\,, \\
\mathbf{1}_{10}: \quad & v_1 \to \frac{1}{\sqrt{2}}v_0\,, \\
\mathbf{1}_{20}: \quad & v_2 \to \frac{1}{\sqrt{2}}v_0\,.
\end{aligned}
\tag{361}
$$

Note that these relative Euler terms are different from that for the Rep($S_3$) SSB phase.

Just like for Rep($S_3$) SSB phase, in the Ising SSB phase we have physically distinguishable vacua: the vacuum $v_0$ has different physical properties compared to the vacua $v_1, v_2$. This can again be seen from the fact that the unique $\mathbb{Z}_2$ subsymmetry of Ising acts differently on $v_0$ and $v_1, v_2$, and also reflects in the presence of relative Euler terms between these vacua.

The order parameters for the Ising SSB phase have generalized charges $\boldsymbol{Q}_{\mathrm{id},-}$ and $\boldsymbol{Q}_{\mathrm{id}+\mathrm{P}}$. An order parameter of charge $\boldsymbol{Q}_{\mathrm{id},-}$ is of conventional type, but instead of carrying charge under an invertible symmetry, it carries a charge under the non-invertible symmetry S as in (338). On the other hand, an order parameter of charge $\boldsymbol{Q}_{\mathrm{id}+\mathrm{P}}$ is a mixture of conventional and string types, see (339). The IR multiplet carrying charge $\boldsymbol{Q}_{\mathrm{id},-}$ comprises only of the untwisted sector operator $\mathcal{O}^-$ discussed above. On the other hand, the IR multiplet carrying charge $\boldsymbol{Q}_{\mathrm{id}+\mathrm{P}}$ comprises of the untwisted sector operator $\mathcal{O}$ discussed above and a P-twisted sector operator,

$$
\mathcal{O}^{\mathrm{P}} = \sqrt{2}v_0\,,
\tag{362}
$$

viewed as living at the end of $D_1^{(\mathrm{P})}$. The reader can see that this satisfies the action (341)

by using the junctions

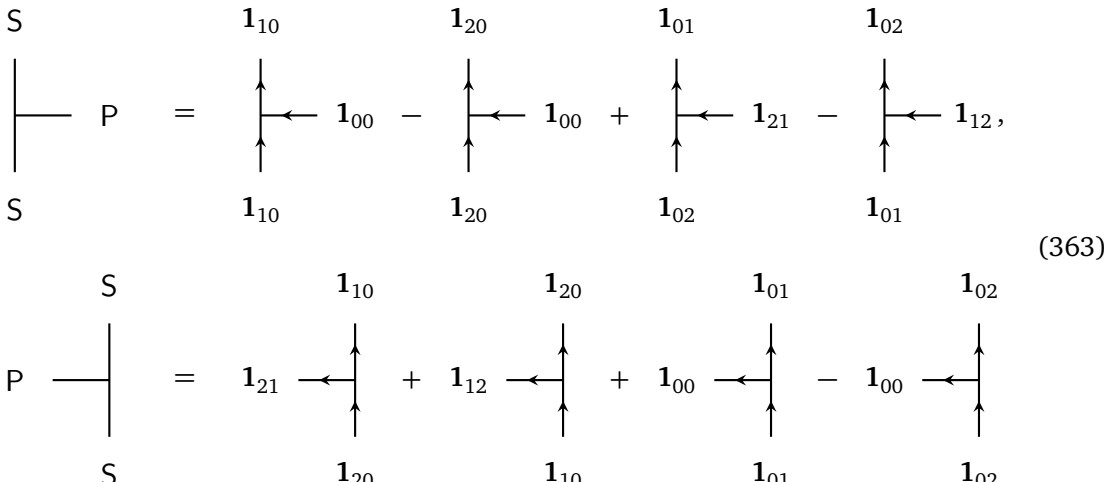

$$(363)$$

# 7 (1+1)d $\mathsf{TY}(\mathbb{Z}_N)$-symmetric gapped phases

In this section, we extend the results of the previous section to understand gapped phases for (1+1)d systems with Tambara-Yamagami $\mathsf{TY}(\mathbb{Z}_N)$ symmetries (329).

## 7.1 $\mathsf{TY}(\mathbb{Z}_N)$ symmetry and its generalized charges

The $\mathsf{TY}(\mathbb{Z}_N)$ symmetry involves a $\mathbb{Z}_N$ subsymmetry involving non-identity invertible lines

$$A^i, \qquad i \in \{1, \cdots, N-1\}, \tag{364}$$

and a non-invertible symmetry line $\mathcal{S}$. The fusion rules are

$$A^N \cong \mathbf{1}, \qquad A \otimes S \cong S \otimes A \cong S, \qquad S \otimes S = \bigoplus_{i=1}^{N} A^i. \tag{365}$$

The SymTFT $\mathfrak{Z}(\mathsf{TY}(\mathbb{Z}_N))$ can be obtained by gauging $\mathbb{Z}_2$ electric-magnetic duality symmetry of 3d $\mathbb{Z}_N$ DW gauge theory [23]. The topological lines of this theory are identified with the Drinfeld center $\mathcal{Z}(\mathsf{TY}(\mathbb{Z}_N))$, which was computed in [137]. Let us describe the topological line defects of $\mathfrak{Z}(\mathsf{TY}(\mathbb{Z}_N))$, which are also the generalized charges for $\mathsf{TY}(\mathbb{Z}_N)$ symmetry.

We can obtain these by explicitly performing the $\mathbb{Z}_2$ gauging on the topological lines of the Dijkgraaf Witten theory, which form the Drinfeld center $\mathcal{Z}(\mathsf{Vec}_{\mathbb{Z}_N})$ described in section 4.2. The lines of DW theory can be denoted as

$$L_{e,m}, \qquad e, m \in \mathbb{Z}_N = \{0, 1, \cdots, N-1\}, \tag{366}$$

where $m$ denotes the $\mathbb{Z}_N$ vortex sector and $e$ describes the $\mathbb{Z}_N$ representation carried by it. The $\mathbb{Z}_2$ symmetry to be gauged exchanges the two labels

$$e \longleftrightarrow m. \tag{367}$$

After gauging, the lines of the Drinfeld center $\mathcal{Z}(\mathsf{TY}(\mathbb{Z}_N))$ are

- $Q_{e,+}$: These are the invariant lines $L_{e,e}$;

- $Q_{0,-}$: These is the generator of the dual $\mathbb{Z}_2$ 1-form symmetry, or in other words the Wilson line for the newly introduced $\mathbb{Z}_2$ gauge group;

- $Q_{e,-}$: These are obtained by stacking $L_{e,e}$ with $Q_{\mathrm{id},-}$;

- $Q_{e,m} \equiv L_{e,m} + L_{m,e}$: These invariant combinations with $e \neq m$, which are quantum dimension 2 lines. Note that $Q_{e,m} = Q_{m,e}$, so we can impose $e > m$ to avoid overcounting;

- $Q_{\Sigma_e,\pm}$: Before gauging, there are topological lines in the twisted sector for $\mathbb{Z}_2$ 0-form symmetry. Sometimes these are also referred to as lines in the *flux sector*. After gauging, these become genuine lines and are denoted $Q_{\Sigma_e,+}$. In addition we can stack these with the dual line $Q_{0,-}$ which yields the lines $Q_{\Sigma_e,-}$.

The fusion of these lines are, see e.g. [23]

$$
\begin{aligned}
Q_{e_1,\epsilon} \otimes Q_{e_2,\epsilon'} &= Q_{e_1+e_2,\epsilon\epsilon'}\,, \\
Q_{e_1,\epsilon} \otimes Q_{e_2,m_2} &= Q_{e_1+e_2,e_1+m_2}\,,
\end{aligned}
\tag{368}
$$

and

$$
Q_{e_1,m_1} \otimes Q_{e_2,m_2}
\tag{369}
$$
$$
= \begin{cases}
Q_{e_1+e_2,+} \oplus Q_{e_1+e_2,-} \oplus Q_{m_1+e_2,+} \oplus Q_{m_1+e_2,-}\,, & e_1+e_2 = m_1+m_2\,, \quad m_1+e_2 = e_1+m_2\,, \\
Q_{e_1+e_2,+} \oplus Q_{e_1+e_2,-} \oplus Q_{m_1+e_2,e_1+m_2}\,, & e_1+e_2 = m_1+m_2\,, \quad m_1+e_2 \neq e_1+m_2\,, \\
Q_{e_1+e_2,m_1+m_2} \oplus Q_{m_1+e_2,+} \oplus Q_{m_1+e_2,-}\,, & e_1+e_2 \neq m_1+m_2\,, \quad m_1+e_2 = e_1+m_2\,, \\
Q_{e_1+e_2,m_1+m_2} \oplus Q_{m_1+e_2,e_1+m_2}\,, & e_1+e_2 \neq m_1+m_2\,, \quad m_1+e_2 \neq e_1+m_2\,.
\end{cases}
$$

In what follows, we discuss the subset of the generalized charges that can be carried by order parameters for gapped phases, which in this case are simple bulk lines $Q_{0,-}$, $Q_{e,m}$, and $Q_{e,\pm}$ for $e, m \in \mathbb{Z}_N$, $e > m$.

### 7.1.1 $Q_{0,-}$ multiplet

This multiplet consists of a single untwisted sector local operator $\mathcal{O}^-$ on which the action of $\mathrm{TY}(\mathbb{Z}_N)$ is

$$
\begin{array}{ccc}
\underset{\mathcal{O}^-}{\bullet} \; \Big|_{A^k} & = & \Big|_{A^k} \; \underset{\mathcal{O}^-}{\bullet}
\end{array}
\tag{370}
$$

$$
\begin{array}{ccc}
\underset{\mathcal{O}^-}{\bullet} \; \Big|_{S} & = & - \; \Big|_{S} \; \underset{\mathcal{O}^-}{\bullet}
\end{array}
$$

where $k = 1, 2, \ldots, N-1$.

### 7.1.2 $Q_{e,\pm}$ multiplets

Each $Q_{e,\pm}$ multiplet for $e = 1, 2, \ldots, N-1$ (and choosing $+$ or $-$) consists of a single local operator $\mathcal{O}_e^{\pm}$ in the twisted sector for $A^e \in \mathrm{TY}(\mathbb{Z}_N)$

$$
A^e \; \rule{2cm}{0.4pt} \; \underset{\mathcal{O}_e^{\pm}}{\bullet}
\tag{371}
$$

The action of $\mathsf{TY}(\mathbb{Z}_N)$ on these multiplets is

$$
\mathsf{A}^e \quad\bullet_{\mathcal{O}_e^\pm}\quad\bigg| \;=\; e^{-2\pi iek/N}\;\mathsf{A}^e \quad\Big|\!\!-\!\!\bullet_{\mathcal{O}_e^\pm}
$$
$$
\mathsf{A}^k \qquad\qquad\qquad \mathsf{A}^k
$$

(372)

$$
\mathsf{A}^e \quad\bullet_{\mathcal{O}_e^\pm}\quad\bigg| \;=\; \pm e^{\frac{2\pi ie^2}{2N}}\;\mathsf{A}^e \quad\Big|\!\!-\!\!\bullet_{\mathcal{O}_e^\pm}
$$
$$
\mathsf{S} \qquad\qquad\qquad \mathsf{S}
$$

where $k = 1, 2, \ldots, N-1$.

### 7.1.3 $Q_{e,0}$ multiplets

The $Q_{e,0}$ multiplet with $e = 1, 2, \ldots, N-1$ consists of an untwisted local operator $\mathcal{O}_e$ and a local operator $\mathcal{O}_e^A$ in the twisted sector for $\mathsf{A}^e \in \mathsf{TY}(\mathbb{Z}_N)$

$$
\bullet_{\mathcal{O}_e} \quad \text{and} \quad \mathsf{A}^e \;-\!\!\!-\!\!\bullet_{\mathcal{O}_e^A}
$$

(373)

The action of $\mathsf{A}^k$ for $k = 1, 2, \ldots, N-1$ is

$$
\bullet_{\mathcal{O}_e}\;\Big| \;=\; e^{-2\pi iek/N}\;\Big|\;\bullet_{\mathcal{O}_e}
$$
$$
\mathsf{A}^k \qquad\qquad\qquad \mathsf{A}^k
$$

(374)

$$
\mathsf{A}^e \;-\!\!\!-\!\!\bullet_{\mathcal{O}_e^A}\;\Big| \;=\; \mathsf{A}^e \;-\!\!\Big|\!\!-\!\!\bullet_{\mathcal{O}_e^A}
$$
$$
\mathsf{A}^k \qquad\qquad\qquad \mathsf{A}^k
$$

and the action of $\mathsf{S}$ is

$$
\bullet_{\mathcal{O}_e}\;\Big| \;=\; \Big|\!\!-\!\!\!\!\!-^{\mathsf{A}^e}\!\!\bullet_{\mathcal{O}_e^A}
$$
$$
\mathsf{S} \qquad\qquad \mathsf{S}
$$

(375)

$$
\mathsf{A}^e \;-\!\!\!-\!\!\bullet_{\mathcal{O}_e^A}\;\Big| \;=\; \mathsf{A}^e \;-\!\!\Big|\quad\bullet_{\mathcal{O}_e}
$$
$$
\mathsf{S} \qquad\qquad\qquad \mathsf{S}
$$

### 7.1.4 $Q_{e,m}$ multiplets

The $Q_{e,m}$ multiplet with $e, m = 1, 2, \ldots, N-1$ and $e > m$ consists of two local operators $\mathcal{O}_{e,m}$ and $\mathcal{O}_{m,e}$ in the twisted sectors for $A^e$ and $A^m$ respectively,

$$
\mathsf{A}^e \;-\!\!\!-\!\!\bullet_{\mathcal{O}_{e,m}} \quad \text{and} \quad \mathsf{A}^m \;-\!\!\!-\!\!\bullet_{\mathcal{O}_{m,e}}
$$

(376)

The action of $\mathsf{A}^k$ for $k = 1, 2, \ldots, N-1$ is

$$
\begin{array}{ccc}
\mathsf{A}^e \; \xrightarrow{\;\;\;\bullet\;} \Big|_{\substack{\mathcal{O}_{e,m}}} & = & e^{-2\pi i mk/N} \;\; \mathsf{A}^e \; \xrightarrow{\;\;+\;\bullet\;}_{\mathcal{O}_{e,m}} \\
\qquad\quad \mathsf{A}^k & & \qquad\quad \mathsf{A}^k \\[2em]
\mathsf{A}^m \; \xrightarrow{\;\;\;\bullet\;} \Big|_{\substack{\mathcal{O}_{m,e}}} & = & e^{-2\pi i ek/N} \;\; \mathsf{A}^m \; \xrightarrow{\;\;+\;\bullet\;}_{\mathcal{O}_{m,e}} \\
\qquad\quad \mathsf{A}^k & & \qquad\quad \mathsf{A}^k
\end{array}
\tag{377}
$$

and the action of $\mathsf{S}$ is

$$
\begin{array}{ccc}
\mathsf{A}^e \; \xrightarrow{\;\;\;\bullet\;} \Big|_{\substack{\mathcal{O}_{e,m}}} & = & \mathsf{A}^e \; \xrightarrow{\quad}\Big|^{\mathsf{A}^m}_{\;\bullet\,\mathcal{O}_{m,e}} \\
\qquad\quad \mathsf{S} & & \qquad\quad \mathsf{S} \\[2em]
\mathsf{A}^m \; \xrightarrow{\;\;\;\bullet\;} \Big|_{\substack{\mathcal{O}_{m,e}}} & = & \mathsf{A}^m \; \xrightarrow{\quad}\Big|^{\mathsf{A}^e}_{\;\bullet\,\mathcal{O}_{e,m}} \\
\qquad\quad \mathsf{S} & & \qquad\quad \mathsf{S}
\end{array}
\tag{378}
$$

## 7.2 Lagrangian algebras and order parameters

Topological boundary conditions of the SymTFT $\mathfrak{Z}(\mathsf{TY}(\mathbb{Z}_N))$ correspond to Lagrangian algebras in the Drinfeld center $\mathcal{Z}(\mathsf{TY}(\mathbb{Z}_N))$. These can be used as physical boundary conditions in the sandwich construction to construct $\mathsf{TY}(\mathbb{Z}_N)$ symmetric gapped phases. Moreover, the bulk lines participating in these Lagrangian algebras describe the generalized charges of order parameters for these gapped phases.

For determining the Lagrangian algebras we need to determine first of all the bosonic lines, i.e. those that have spin $+1$, and then make an ansatz

$$
\mathcal{A} = \bigoplus_{a \text{ bosonic}} n_a \mathbf{Q}_a \,,
\tag{379}
$$

for non-negative integers $n_a$, solving the dimension condition (66) and the inequality in (68). The spins are

$$
\theta(\mathbf{Q}_{e,\pm}) = e^{-\frac{2\pi i e^2}{N}} \,, \qquad \theta(\mathbf{Q}_{e,m}) = e^{-\frac{2\pi i em}{N}} \,.
\tag{380}
$$

The spin of the lines $\Sigma_{e,\epsilon}$ can be determined as well and unless $N = q^2$ is never equal to 1. In particular, the $\Sigma$ lines only contribute to Lagrangian algebras for $N = q^2$.[22] In what follows, we will discuss general Lagrangian algebras that do not involve $\Sigma_{e,\epsilon}$ lines. Thus, we discuss most general Lagrangian algebras for the case $N \neq q^2$, but miss Lagrangian algebras involving $\Sigma_{e,\epsilon}$ lines when $N = q^2$.

**Lagrangians for the DW theory.** Before discussing the TY Lagrangians, it will be useful to recapitulate the gapped boundary conditions for the DW theory for $\mathbb{Z}_N$, and fix some notation.

---

[22]We thank A. Antinucci and C. Copetti for discussions on this.

There are two canonical $N$-independent boundary conditions for the $\mathbb{Z}_N$ DW theory: Dirichlet (Dir), where the lines $L_{e,0}$ end, and Neumann (Neu), where the lines $L_{0,m}$ end. We will denote these by

$$
\begin{aligned}
\mathcal{A}_{\text{Dir}} &= \sum_{e=0}^{N-1} L_{e,0}, \\
\mathcal{A}_{\text{Neu}} &= \sum_{m=0}^{N-1} L_{0,m}.
\end{aligned}
\tag{381}
$$

For $N$ not prime, and $p|N$, there are additional mixed boundary conditions $\text{Neu}(\mathbb{Z}_p)$. We denote these by

$$
\mathcal{A}_{\text{Neu}(\mathbb{Z}_p)} = \sum_{(e,m)} L_{e,m},
\tag{382}
$$

where the sum is over

$$
L_{e,m}, \qquad e \in \{0, p, 2p, \cdots, (q-1)p\}, \quad m \in \{0, q, 2q, \cdots, (p-1)q\},
\tag{383}
$$

i.e. the lines in $\mathbb{Z}_N/\mathbb{Z}_p \cong \mathbb{Z}_q$ as well as their dual lines in $\widehat{Z}_p$ end.

$\mathcal{A}_{\textbf{Dir,Neu}}$ **for all** $\mathsf{TY}(\mathbb{Z}_N)$. For any $N$ the following is a Lagrangian algebra

$$
\mathcal{A}_{\text{Dir,Neu}}^{(N)} = \boldsymbol{Q}_{0,+} + \boldsymbol{Q}_{0,-} + \sum_{e=1}^{N-1} \boldsymbol{Q}_{e,0}.
\tag{384}
$$

The subscript Dir, Neu indicates that the resulting boundary is a combination of the Dirichlet and Neumann boundaries of the DW theory. The combination is forced as these two boundaries are exchanged by the gauged $\mathbb{Z}_2$ 0-form symmetry. To see the inclusion of both Dirichlet and Neumann from the Lagrangian algebra, note that the fact that $\boldsymbol{Q}_{e,0}$ can end on the boundary means that both $L_{e,0}$ and $L_{0,e}$ must end. The Dir boundary of DW has the property that $L_{e,0}$ end on it, and the Neu boundary of DW has the property that $L_{0,e}$ end on it.

It is easy to see that the Lagrangian algebra $\mathcal{A}_{\text{Dir,Neu}}$ satisfies all the necessary conditions. First note that the spin of the lines $\boldsymbol{Q}_{0,\pm}$ and $\boldsymbol{Q}_{e,0}$ is always +1. To see that the condition (66) holds, note that

$$
\begin{aligned}
\dim(\mathcal{A}^{(N)}) &= \dim(\boldsymbol{Q}_{0,+}) + \dim(\boldsymbol{Q}_{0,-}) + \sum_{e=1}^{N-1} \dim(\boldsymbol{Q}_{e,0}) \\
&= 1 + 1 + 2(N-1) \\
&= 2N.
\end{aligned}
\tag{385}
$$

On the other hand, we have

$$
\begin{aligned}
\dim^2(\mathsf{TY}(\mathbb{Z}_N)) &= \sum_{i=1}^{N} \dim^2(\mathsf{A}^i) + \dim^2(\mathsf{S}) \\
&= N + N \\
&= 2N.
\end{aligned}
\tag{386}
$$

The inequalities (68) imply

$$
n_{\text{id},\pm} n_{e,0} \le n_{e,0},
\tag{387}
$$

and

$$
e + k \ne N: \quad n_{e,0} n_{k,0} \le n_{e+k,0}, \qquad e + k = N: \quad n_{e,0} n_{k,0} \le n_{\text{id},+} + n_{\text{id},-}.
\tag{388}
$$

One can indeed check that the above solution satisfies all these conditions.

**Lagrangian algebras for $N \leq 8$.** We provide some examples for low $N$, where we determined the complete set of Lagrangian algebras by solving the boson, dimension, and inequality conditions. We only mention the Lagrangian algebras other than $\mathcal{A}_{\text{Dir,Neu}}$ in (384)

$$\text{TY}(\mathbb{Z}_4): \quad \mathcal{A}_{\text{Neu}(\mathbb{Z}_2, \mathbb{Z}_2)} = \boldsymbol{Q}_{0,+} + \boldsymbol{Q}_{0,-} + \boldsymbol{Q}_{2,+} + \boldsymbol{Q}_{2,-} + 2\boldsymbol{Q}_{2,0}, \tag{389}$$

$$\mathcal{A}_{\text{Neu}(\mathbb{Z}_2)} = \boldsymbol{Q}_{0,+} + \boldsymbol{Q}_{2,-} + \boldsymbol{Q}_{2,0} + \boldsymbol{Q}_{\Sigma_{1,+}} + \boldsymbol{Q}_{\Sigma_{3,-}},$$

$$\text{TY}(\mathbb{Z}_6): \quad \mathcal{A}_{\text{Neu}(\mathbb{Z}_2, \mathbb{Z}_3)} = \boldsymbol{Q}_{0,+} + \boldsymbol{Q}_{0,-} + \boldsymbol{Q}_{2,0} + \boldsymbol{Q}_{3,0} + \boldsymbol{Q}_{4,0} + \boldsymbol{Q}_{3,2} + \boldsymbol{Q}_{4,3},$$

$$\text{TY}(\mathbb{Z}_8): \quad \mathcal{A}_{\text{Neu}(\mathbb{Z}_2, \mathbb{Z}_4)} = \boldsymbol{Q}_{0,+} + \boldsymbol{Q}_{0,-} + \boldsymbol{Q}_{4,+} + \boldsymbol{Q}_{4,-} + \boldsymbol{Q}_{2,0} + 2\boldsymbol{Q}_{4,0} + \boldsymbol{Q}_{6,0} + \boldsymbol{Q}_{4,2} + \boldsymbol{Q}_{6,4}.$$

In the above, we are using the conventions of [23], using which one can check that for $N = 4$ the topological lines $\boldsymbol{Q}_{\Sigma_{1,+}}$ and $\boldsymbol{Q}_{\Sigma_{3,-}}$ are bosons and can participate in a Lagrangian algebra. For other values of $N \in [2, 8]$, the only Lagrangian algebra is $\mathcal{A}_{\text{Dir,Neu}}$. The labeling of the Lagrangian algebras is chosen to reflect the boundary conditions of the DW theory that combine to form the boundary condition of the SymTFT $\mathfrak{Z}(\text{TY}(\mathbb{Z}_N))$. The boundary condition of $\mathfrak{Z}(\text{TY}(\mathbb{Z}_N))$ corresponding to $\mathcal{A}_{\text{Neu}(\mathbb{Z}_p, \mathbb{Z}_q)}$ includes a combination (direct sum) of $\mathbb{Z}_p$ and $\mathbb{Z}_q$ Neumann boundary conditions of the $\mathbb{Z}_N$ DW theory. For $p = q$, this means we have two copies of the $\mathbb{Z}_m$ Neumann boundary condition. The boundary condition of $\mathfrak{Z}(\text{TY}(\mathbb{Z}_{q^2}))$ corresponding to $\mathcal{A}_{\text{Neu}(\mathbb{Z}_q)}$ includes only the $\mathbb{Z}_q$ Neumann boundary condition of the $\mathbb{Z}_{q^2}$ DW theory.

For example, consider the Lagrangian algebra $\mathcal{A}_{\text{Neu}(\mathbb{Z}_2, \mathbb{Z}_2)}$ of $\mathfrak{Z}(\text{TY}(\mathbb{Z}_4))$ described above. The $\mathbb{Z}_2$ Neumann boundary condition of $\mathbb{Z}_4$ DW theory has the property that $L_{2,0}$, $L_{0,2}$ and $L_{2,2}$ can end on it. This means that $\boldsymbol{Q}_{2,+}$ can end on it, and $\boldsymbol{Q}_{2,0}$ has two ends along it. The number of ends is reflected in the coefficients of these bulk lines in $\mathcal{A}_{\text{Neu}(\mathbb{Z}_2)}$. Finally, $\boldsymbol{Q}_{0,-}$ can end as well because the gauged $\mathbb{Z}_2$ exchanges the two $\mathbb{Z}_2$ Neumann boundary conditions, and the difference of the identity operators along the two boundaries has to be attached to $\boldsymbol{Q}_{0,-}$. This implies $\boldsymbol{Q}_{2,-}$ must end as well.

**Lagrangian algebras for general $N$.** In fact, we can describe the structure of all Lagrangian algebras for arbitrary $N$ – this is a complete set for $N$ not a perfect square. As we pointed out before, in the case of a perfect square there will be additional algebras involving the $\boldsymbol{Q}_\Sigma$. For this purpose, let us note that the $\mathbb{Z}_2$ electric-magnetic duality symmetry of the $\mathbb{Z}_N$ Dijkgraaf-Witten theory acts as

$$L_{e,m} \longleftrightarrow L_{m,e}, \tag{390}$$

which exchanges topological boundary conditions of the DW theory as

$$\mathcal{A}_{\text{Neu}(\mathbb{Z}_p)} \longleftrightarrow \mathcal{A}_{\text{Neu}(\mathbb{Z}_q)}, \tag{391}$$

such that

$$pq = N. \tag{392}$$

The irreducible boundary conditions for the SymTFT $\mathfrak{Z}(\text{TY}(\mathbb{Z}_N))$ are then, using the notation in (382),

$$\mathcal{A}_{\text{Neu}(\mathbb{Z}_p, \mathbb{Z}_q)} \equiv \mathcal{A}_{\text{Neu}(\mathbb{Z}_p)} \oplus \mathcal{A}_{\text{Neu}(\mathbb{Z}_q)}, \qquad p \leq q. \tag{393}$$

Thus $\boldsymbol{Q}_{e,m}$ appears in $\mathcal{A}_{\text{Neu}(\mathbb{Z}_p, \mathbb{Z}_q)}$ if and only if either $L_{e,m}$ or $L_{m,e}$ appears in $\mathcal{A}_{\text{Neu}(\mathbb{Z}_p)}$. If this is the case, and only one of $L_{e,m}$ or $L_{m,e}$ appears in $\mathcal{A}_{\text{Neu}(\mathbb{Z}_p)}$, then the coefficient of $\boldsymbol{Q}_{e,m}$ in $\mathcal{A}_{\text{Neu}(\mathbb{Z}_p, \mathbb{Z}_q)}$ is one. If on the other hand, both $L_{e,m}$ and $L_{m,e}$ appear in $\mathcal{A}_{\text{Neu}(\mathbb{Z}_p)}$, then the coefficient of $\boldsymbol{Q}_{e,m}$ in $\mathcal{A}_{\text{Neu}(\mathbb{Z}_p, \mathbb{Z}_q)}$ is two. Similarly, if $L_{e,e}$ appears in $\mathcal{A}_{\text{Neu}(\mathbb{Z}_p)}$, then both $\boldsymbol{Q}_{e,\pm}$ appear in $\mathcal{A}_{\text{Neu}(\mathbb{Z}_p, \mathbb{Z}_q)}$.

**Matching with classification of module categories.** Before closing this section, let us note that from the analysis of [138], one can deduce that the indecomposable module categories for $\mathsf{TY}(\mathbb{Z}_N)$ are also parameterized by factorizations of $N$

$$pq = N\,, \tag{394}$$

when $N$ is not a perfect square, thus having the same classification as for the Lagrangian algebras in the Drinfeld center $\mathcal{Z}(\mathsf{TY}(\mathbb{Z}_N))$ discussed above. This has to be the case as both indecomposable module categories and Lagrangian algebras parameterize different topological boundary conditions of the SymTFT $\mathfrak{Z}(\mathsf{TY}(\mathbb{Z}_N))$. The correspondence takes the following form. Pick a Lagrangian algebra $\mathcal{A}$ and let $\mathfrak{B}^{\text{top}}_{\mathcal{A}}$ be the topological boundary of $\mathfrak{Z}(\mathsf{TY}(\mathbb{Z}_N))$ obtained by condensing $\mathcal{A}$. Then, the corresponding indecomposable module category $\mathcal{M}_{\mathcal{A}}$ describes topological line defects living at the interface between the symmetry boundary $\mathsf{B}^{\text{sym}}_{\mathsf{TY}(\mathbb{Z}_N)}$ and the topological boundary $\mathfrak{B}^{\text{top}}_{\mathcal{A}}$. See [58] for more details.

## 7.3 Gapped phases

The symmetry boundary is fixed to be

$$\mathsf{B}^{\text{sym}}_{\mathsf{TY}(\mathbb{Z}_N)} = \mathcal{A}_{\text{Dir,Neu}}\,. \tag{395}$$

Using the decomposition as boundaries for $\mathbb{Z}_N$ DW theory

$$\mathcal{A}_{\text{Dir,Neu}} \equiv \mathcal{A}_{\text{Dir}} \oplus \mathcal{A}_{\text{Neu}}\,, \tag{396}$$

we can recognize the line $\mathsf{A} \in \mathsf{TY}(\mathbb{Z}_N)$ along the boundary $\mathcal{A}_{\text{Dir,Neu}}$ as

$$\mathsf{A} \equiv \mathsf{M} \oplus \mathsf{E}\,, \tag{397}$$

where $\mathsf{M}$ is the line generating $\mathbb{Z}_N$ symmetry localized along $\mathcal{A}_{\text{Dir}}$ and $\mathsf{E}$ is the line generating $\mathbb{Z}_N$ symmetry localized along $\mathcal{A}_{\text{Neu}}$.[23] On the other hand, the line $\mathsf{S}$ arises from the end of $\mathbb{Z}_2$ electric magnetic duality defect, which interchanges $\mathcal{A}_{\text{Dir}}$ and $\mathcal{A}_{\text{Neu}}$.

### 7.3.1 $\mathsf{TY}(\mathbb{Z}_N)$ SSB $= (\mathbb{Z}_1, \mathbb{Z}_N)$ SSB phase

Now let us choose the physical boundary to be the same as symmetry boundary

$$\mathsf{B}^{\text{phys}} = \mathsf{B}^{\text{sym}}_{\mathsf{TY}(\mathbb{Z}_N)} = \mathcal{A}_{\text{Dir,Neu}}\,, \tag{398}$$

and analyze the resulting $\mathsf{TY}(\mathbb{Z}_N)$-symmetric gapped phase.

**Derivation directly from the SymTFT $\mathfrak{Z}(\mathsf{TY}(\mathbb{Z}_N))$.** From the SymTFT perspective, we find the following lines can end

$$
\begin{array}{cc}
\mathcal{A}_{\text{Dir,Neu}} & \mathcal{A}_{\text{Dir,Neu}} \\
\end{array}
$$

$$
\boxed{
\begin{array}{c}
\boldsymbol{Q}_{0,\pm} \\
\boldsymbol{Q}_{e,0}
\end{array}
} \tag{399}
$$

---

[23]The line $\mathsf{M}$ comes from projecting bulk line $L_{0,1}$ onto the boundary $\mathcal{A}_{\text{Dir}}$, and the line $\mathsf{E}$ comes from projecting bulk line $L_{1,0}$ onto the boundary $\mathcal{A}_{\text{Neu}}$.

and therefore this phase has $N+1$ vacua which we denote $v_0, v_1, \cdots, v_N$. Let us denote the untwisted sector local operator arising from $Q_{0,-}$ as $\mathcal{O}_-$ and the untwisted sector local operators arising from $Q_{e,0}$ as $\mathcal{O}_e$. From the fusion of bulk lines, we can straightforwardly derive the product rules for these operators to be

$$
\begin{aligned}
\mathcal{O}_-^2 &= 1, \\
\mathcal{O}_- \mathcal{O}_e &= \mathcal{O}_e, & e &\in \{1, 2, \cdots, N-1\}, \\
\mathcal{O}_{e_1} \mathcal{O}_{e_2} &= \sqrt{2}\, \mathcal{O}_{e_1 + e_2 \bmod N}, & &\text{if } e_1 + e_2 \neq N, \\
\mathcal{O}_e \mathcal{O}_{N-e} &= 1 + \mathcal{O}_-.
\end{aligned}
\tag{400}
$$

From this, we determine the vacua to be

$$
\begin{aligned}
v_0 &= \frac{1}{2}(1 - \mathcal{O}_-), \\
v_i &= \frac{1}{2N}\left(1 + \mathcal{O}_- + \sqrt{2}\sum_{e=1}^{N-1} \omega_N^{ie} \mathcal{O}_e\right), & i &\in \{1, 2, \cdots, N\},
\end{aligned}
\tag{401}
$$

where $\omega_N$ is a primitive $N$-th root of unity.

The linking action of $D_1^{(A)}$ can be deduced from the DW theory, and is given on untwisted local operators as

$$
\begin{aligned}
\mathcal{O}_- &\to \mathcal{O}_-, \\
\mathcal{O}_e &\to \omega_N^e \mathcal{O}_e, & e &\in \{1, 2, \cdots, N-1\},
\end{aligned}
\tag{402}
$$

which implies its linking action on vacua is

$$
\begin{aligned}
v_0 &\to v_0, \\
v_i &\to v_{i+1}, & i &\in \{1, 2, \cdots, N-1\}, \\
v_N &\to v_1.
\end{aligned}
\tag{403}
$$

Thus, from the perspective of $\mathbb{Z}_N$ subsymmetry of $\mathsf{TY}(\mathbb{Z}_N)$ generated by A, the vacuum $v_0$ gives rise to a trivial phase, while the other vacua $\{v_1, v_2, \cdots, v_N\}$ give rise to a $\mathbb{Z}_N$ SSB phase. In other words, in terms of the $\mathbb{Z}_N$ subsymmetry, the $\mathsf{TY}(\mathbb{Z}_N)$-symmetric phase under study decomposes as

$$
\text{Trivial Phase} \oplus \mathbb{Z}_N \text{ SSB Phase.} \tag{404}
$$

The line operator $D_1^{(A)}$ generating the A-symmetry is

$$
D_1^{(A)} = \mathbf{1}_{00} \oplus \bigoplus_{i=1}^{N-1} \mathbf{1}_{i\,i+1} \oplus \mathbf{1}_{N1}. \tag{405}
$$

The S-symmetry exchanges the two sub-phases in (404). To see this, note that the linking action of $D_1^{(S)}$ on untwisted local operators is

$$
\begin{aligned}
1 &\to \sqrt{N}, \\
\mathcal{O}_- &\to -\sqrt{N}\,\mathcal{O}_-, \\
\mathcal{O}_e &\to 0, & e &\in \{1, 2, \cdots, N-1\},
\end{aligned}
\tag{406}
$$

which implies its linking action on vacua is

$$
\begin{aligned}
v_0 &\to \sqrt{N} \sum_{i=1}^{N} v_i, \\
v_i &\to \frac{1}{\sqrt{N}} v_0, & i &\in \{1, 2, \cdots, N\}.
\end{aligned}
\tag{407}
$$

Thus, the line operator $D_1^{(\mathsf{S})}$ generating the $\mathsf{S}$-symmetry is

$$D_1^{(\mathsf{S})} = \bigoplus_{i=1}^{N} \mathbf{1}_{0i} \oplus \bigoplus_{i=1}^{N} \mathbf{1}_{i0}, \tag{408}$$

along with non-trivial relative Euler terms encoded in the following linking actions

$$
\begin{aligned}
\mathbf{1}_{0i}: \quad & v_0 \to \sqrt{N}\, v_i, \\
\mathbf{1}_{i0}: \quad & v_i \to \frac{1}{\sqrt{N}} v_0,
\end{aligned}
\tag{409}
$$

for $1 \le i \le N$. It is easy to check that the lines $D_1^{(\mathsf{A})}$ and $D_1^{(\mathsf{S})}$ satisfy the fusions (365).

We denote this phase as the $\mathsf{TY}(\mathbb{Z}_N)$ **SSB phase**, or equivalently as $(\mathbb{Z}_1, \mathbb{Z}_N)$ **SSB phase**, using the structure of the two sub-phases with respect to the $\mathbb{Z}_N$ subsymmetry. Note that the vacuum $v_0$ can be physically distinguished from the other vacua, while vacua $\{v_1, v_2, \cdots, v_N\}$ are physically indistinguishable.

**Derivation from the $\mathbb{Z}_N$ DW theory.** From the point of view of the $\mathbb{Z}_N$ DW theory, we are considering a collection of four interval compactifications

$$(\mathcal{A}_{\mathrm{Dir}}, \mathcal{A}_{\mathrm{Dir}}), \qquad (\mathcal{A}_{\mathrm{Dir}}, \mathcal{A}_{\mathrm{Neu}}), \qquad (\mathcal{A}_{\mathrm{Neu}}, \mathcal{A}_{\mathrm{Dir}}), \qquad (\mathcal{A}_{\mathrm{Neu}}, \mathcal{A}_{\mathrm{Neu}}). \tag{410}$$

The first and fourth ones are related by the action of $\mathbb{Z}_2$ electric-magnetic duality symmetry, and so are gauge equivalent. Similarly, the second and the third ones are gauge equivalent. We can thus focus our attention only on the first two compactifications, and express the $\mathfrak{Z}(\mathsf{TY}(\mathbb{Z}_N))$ compactification as

$$(\mathcal{A}_{\mathrm{Dir,Neu}}, \mathcal{A}_{\mathrm{Dir,Neu}}) \equiv (\mathcal{A}_{\mathrm{Dir}}, \mathcal{A}_{\mathrm{Neu}}) \oplus (\mathcal{A}_{\mathrm{Dir}}, \mathcal{A}_{\mathrm{Dir}}). \tag{411}$$

The $\mathbb{Z}_N$ subsymmetry of $\mathsf{TY}(\mathbb{Z}_N)$ living on the left $\mathcal{A}_{\mathrm{Dir,Neu}}$ boundary is identified with the $\mathbb{Z}_N$ symmetry living on the left $\mathcal{A}_{\mathrm{Dir}}$ boundary on the RHS of the above equation. Thus, from the RHS we read that the resulting $\mathsf{TY}(\mathbb{Z}_N)$-symmetric gapped phase decomposes as (404) phase in terms of the $\mathbb{Z}_N$ subsymmetry. This matches what we discussed above.

The $\mathsf{S}$ line changes the left boundary condition for the DW theory as

$$\mathcal{A}_{\mathrm{Dir}} \to \mathcal{A}_{\mathrm{Neu}}, \tag{412}$$

thus acting on the sub-phases as

$$
\begin{aligned}
(\mathcal{A}_{\mathrm{Dir}}, \mathcal{A}_{\mathrm{Neu}}) &\to (\mathcal{A}_{\mathrm{Neu}}, \mathcal{A}_{\mathrm{Neu}}), \\
(\mathcal{A}_{\mathrm{Dir}}, \mathcal{A}_{\mathrm{Dir}}) &\to (\mathcal{A}_{\mathrm{Neu}}, \mathcal{A}_{\mathrm{Dir}}).
\end{aligned}
\tag{413}
$$

Combining it with the $\mathbb{Z}_2$ gauge transformation discussed above, we can describe the action of $\mathsf{S}$ as a permutation of two sub-phases

$$(\mathcal{A}_{\mathrm{Dir}}, \mathcal{A}_{\mathrm{Neu}}) \longleftrightarrow (\mathcal{A}_{\mathrm{Dir}}, \mathcal{A}_{\mathrm{Dir}}). \tag{414}$$

This also matches what we discussed above.

**Order parameters.** The order parameters of the $\mathsf{TY}(\mathbb{Z}_1, \mathbb{Z}_N)$ phase have generalized charges $\boldsymbol{Q}_{0,-}$ and $\boldsymbol{Q}_{e,0}$. The order parameter of charge $\boldsymbol{Q}_{0,-}$ is a conventional untwisted local operator, but uncharged under the invertible $\mathbb{Z}_N$ subsymmetry and charged under the non-invertible symmetry $\mathsf{S}$ as in (370). The other order parameters, of charge $\boldsymbol{Q}_{e,0}$, are instead a mixture of conventional order parameters and string type, see 7.1.3. The IR multiplet carrying charge $\boldsymbol{Q}_{0,-}$ has only the untwisted sector operator $\mathcal{O}^-$. On the other hand, the IR multiplet carrying charge $\boldsymbol{Q}_{e,0}$ comprises of an untwisted local operator $\mathcal{O}_e$ and an $A^e$-twisted sector operator

$$\mathcal{O}_e^A = \sqrt{2}v_0, \tag{415}$$

viewed as living at the end of $D_1^{(A^e)}$. This can be seen using the junction

$$
\begin{array}{cccccc}
\mathsf{S} & \mathbf{1}_{10} & \mathbf{1}_{20} & \mathbf{1}_{02} & \mathbf{1}_{03} \\
\Big| & \uparrow \mathbf{1}_{00} & \uparrow \mathbf{1}_{00} & \uparrow \mathbf{1}_{12} & \uparrow \mathbf{1}_{23} \\
\Big| \leftarrow & \mathsf{A} = \omega^{N-1} \leftarrow & + \omega^{N-2} \leftarrow & + \ldots + \leftarrow & + \omega \leftarrow & + \ldots \\
\Big| & \uparrow & \uparrow & \uparrow & \uparrow \\
\mathsf{S} & \mathbf{1}_{10} & \mathbf{1}_{20} & \mathbf{1}_{01} & \mathbf{1}_{02}
\end{array}
\tag{416}
$$

### 7.3.2 $(\mathbb{Z}_p, \mathbb{Z}_q)$ SSB phases

Let us now choose a general physical boundary

$$\mathfrak{B}^{\text{phys}} = \mathcal{A}_{\text{Neu}(\mathbb{Z}_p), \text{Neu}(\mathbb{Z}_q)}, \tag{417}$$

and study the resulting $\mathsf{TY}(\mathbb{Z}_N)$-symmetric gapped phase.

**Derivation directly from the SymTFT $\mathfrak{Z}(\mathsf{TY}(\mathbb{Z}_N))$.** This gives rise to a total of $(p+q)$ untwisted sector local operators. Let us label the operators arising from ending $\boldsymbol{Q}_{ep,0}$ along $\mathcal{A}_{\text{Neu}(\mathbb{Z}_p)} \subset \mathcal{A}_{\text{Neu}(\mathbb{Z}_p), \text{Neu}(\mathbb{Z}_q)}$ as

$$\mathcal{O}_e^{(2)}, \qquad e \in \{1, 2, \cdots, q-1\}, \tag{418}$$

and the operators arising from ending $\boldsymbol{Q}_{mq,0}$ along $\mathcal{A}_{\text{Neu}(\mathbb{Z}_q)} \subset \mathcal{A}_{\text{Neu}(\mathbb{Z}_p), \text{Neu}(\mathbb{Z}_q)}$ as

$$\mathcal{O}_m^{(1)}, \qquad m \in \{1, 2, \cdots, p-1\}. \tag{419}$$

The product of these operators can be deduced to be

$$
\begin{aligned}
\mathcal{O}_-^2 &= 1, \\
\mathcal{O}_- \mathcal{O}_e^{(2)} &= \mathcal{O}_e^{(2)}, \\
\mathcal{O}_- \mathcal{O}_m^{(1)} &= -\mathcal{O}_m^{(1)}, \\
\mathcal{O}_{e_1}^{(2)} \mathcal{O}_{e_2}^{(2)} &= \sqrt{2}\mathcal{O}_{e_1+e_2 \bmod q}^{(2)}, && \text{if } e_1 + e_2 \neq q, \\
\mathcal{O}_{m_1}^{(1)} \mathcal{O}_{m_2}^{(1)} &= \sqrt{2}\mathcal{O}_{m_1+m_2 \bmod p}^{(1)}, && \text{if } m_1 + m_2 \neq p, \\
\mathcal{O}_e^{(2)} \mathcal{O}_{q-e}^{(2)} &= 1 + \mathcal{O}_-, \\
\mathcal{O}_m^{(1)} \mathcal{O}_{p-m}^{(1)} &= 1 - \mathcal{O}_-, \\
\mathcal{O}_m^{(1)} \mathcal{O}_e^{(2)} &= 0.
\end{aligned}
\tag{420}
$$

From this, we can compute the vacua to be

$$
\begin{aligned}
v_{i-1} &= \frac{1}{2p}\left(1 - \mathcal{O}_- + \sqrt{2}\sum_{m=1}^{p-1}\omega_p^{im}\mathcal{O}_m^{(1)}\right), && i \in \{1,\cdots,p\}, \\
v_{p+i-1} &= \frac{1}{2q}\left(1 + \mathcal{O}_- + \sqrt{2}\sum_{e=1}^{q-1}\omega_q^{ie}\mathcal{O}_e^{(2)}\right), && i \in \{1,\cdots,q\},
\end{aligned}
\tag{421}
$$

where $\omega_p$ and $\omega_q$ are primitive $p$-th and $q$-th roots of unity respectively.

The linking action of $D_1^{(A)}$ on untwisted local operators is

$$
\begin{aligned}
\mathcal{O}_- &\to \mathcal{O}_-, \\
\mathcal{O}_e^{(2)} &\to \omega_q^e \mathcal{O}_e, \\
\mathcal{O}_m^{(1)} &\to \omega_p^m \mathcal{O}_m,
\end{aligned}
\tag{422}
$$

which implies its linking action on vacua is

$$
\begin{aligned}
v_i &\to v_{i+1}, && i \in \{0,1,\cdots,p-2\}, \\
v_{p-1} &\to v_0, \\
v_{p-1+i} &\to v_{p+i}, && i \in \{1,\cdots,q-1\}, \\
v_{p+q-1} &\to v_p.
\end{aligned}
\tag{423}
$$

Thus, from the perspective of $\mathbb{Z}_N$ subsymmetry of $\mathsf{TY}(\mathbb{Z}_N)$ generated by A, the vacua $\{v_0, v_1, \cdots, v_{p-1}\}$ give rise to a $\mathbb{Z}_p$ SSB phase, while the other vacua $\{v_p, v_{p+1}, \cdots, v_{p+q-1}\}$ give rise to a $\mathbb{Z}_q$ SSB phase. In other words, in terms of the $\mathbb{Z}_N$ subsymmetry, the $\mathsf{TY}(\mathbb{Z}_N)$-symmetric phase under study decomposes as

$$
\mathbb{Z}_p \text{ SSB Phase} \ \oplus \ \mathbb{Z}_q \text{ SSB Phase}.
\tag{424}
$$

The line operator $D_1^{(A)}$ generating the A-symmetry is

$$
D_1^{(A)} = \bigoplus_{i=0}^{p-2} \mathbf{1}_{i\,i+1} \oplus \mathbf{1}_{p-1\,0} \bigoplus_{i=1}^{q-1} \mathbf{1}_{p-1+i\,p+i} \oplus \mathbf{1}_{p+q-1\,p}.
\tag{425}
$$

The S-symmetry exchanges the two sub-phases in (424). To see this, note that the linking action of $D_1^{(S)}$ on untwisted local operators is

$$
\begin{aligned}
1 &\to \sqrt{pq}, \\
\mathcal{O}_- &\to -\sqrt{pq}\,\mathcal{O}_-, \\
\mathcal{O}_e^{(2)} &\to 0, \\
\mathcal{O}_m^{(1)} &\to 0,
\end{aligned}
\tag{426}
$$

which implies its linking action on vacua is

$$
\begin{aligned}
v_i &\to \sqrt{\frac{q}{p}}\,\sum_{j=1}^{q} v_{p-1+j}, && i \in \{0,1,\cdots,p-1\}, \\
v_{p-1+i} &\to \sqrt{\frac{p}{q}}\,\sum_{j=0}^{p-1} v_j, && i \in \{1,\cdots,q\}.
\end{aligned}
\tag{427}
$$

Thus, the line operator $D_1^{(S)}$ generating the S-symmetry is

$$D_1^{(S)} = \bigoplus_{(i,j)=(0,1)}^{(p-1,q)} \mathbf{1}_{i\,p-j+1} \oplus \bigoplus_{(i,j)=(1,0)}^{(q,p-1)} \mathbf{1}_{p-1+i\,j}\,, \tag{428}$$

along with non-trivial relative Euler terms encoded in the following linking actions

$$
\begin{aligned}
\mathbf{1}_{ip-1+j} : v_i &\to \sqrt{\frac{q}{p}} v_{p-1+j}\,, && i \in \{0,1,\cdots,p-1\},\ j \in \{1,\cdots,q\}\,, \\
\mathbf{1}_{p-1+ij} : v_{p-1+i} &\to \sqrt{\frac{p}{q}} v_j\,, && i \in \{1,\cdots,q\},\ j \in \{0,1,\cdots,p-1\}\,.
\end{aligned}
\tag{429}
$$

It is easy to check that the lines $D_1^{(A)}$ and $D_1^{(S)}$ satisfy the fusions (365).

We denote this phase as $(\mathbb{Z}_p, \mathbb{Z}_q)$ **SSB phase** using the structure of the two sub-phases (424) with respect to the $\mathbb{Z}_N$ subsymmetry. Note that the vacua $\{v_0, v_1, \cdots, v_{p-1}\}$ are physically indistinguishable and the vacua $\{v_p, v_{p+1}, \cdots, v_{p+q-1}\}$ are physically indistinguishable, but any two vacua $v_i$ and $v_j$ for $0 \le i \le p-1$ and $p \le j \le p+q-1$ are physically distinguishable.

**Derivation from the $\mathbb{Z}_N$ DW theory.** From the point of view of the $\mathbb{Z}_N$ DW theory, we are considering a collection of four interval compactifications

$$(\mathcal{A}_{\text{Dir}}, \mathcal{A}_{\text{Neu}(\mathbb{Z}_p)})\,, \qquad (\mathcal{A}_{\text{Dir}}, \mathcal{A}_{\text{Neu}(\mathbb{Z}_q)})\,, \qquad (\mathcal{A}_{\text{Neu}}, \mathcal{A}_{\text{Neu}(\mathbb{Z}_p)})\,, \qquad (\mathcal{A}_{\text{Neu}}, \mathcal{A}_{\text{Neu}(\mathbb{Z}_q)})\,. \tag{430}$$

The first and fourth ones are related by the action of $\mathbb{Z}_2$ electric-magnetic duality symmetry, and so are gauge equivalent. Similarly, the second and the third ones are gauge equivalent. We can thus focus our attention only on the first two compactifications, and express the $\mathfrak{Z}(\mathsf{TY}(\mathbb{Z}_N))$ compactification as

$$(\mathcal{A}_{\text{Dir,Neu}}, \mathcal{A}_{\text{Neu}(\mathbb{Z}_p),\text{Neu}(\mathbb{Z}_q)}) \equiv (\mathcal{A}_{\text{Dir}}, \mathcal{A}_{\text{Neu}(\mathbb{Z}_p)}) \oplus (\mathcal{A}_{\text{Dir}}, \mathcal{A}_{\text{Neu}(\mathbb{Z}_q)})\,. \tag{431}$$

The $\mathbb{Z}_N$ subsymmetry of $\mathsf{TY}(\mathbb{Z}_N)$ living on the left $\mathcal{A}_{\text{Dir,Neu}}$ boundary is identified with the $\mathbb{Z}_N$ symmetry living on the left $\mathcal{A}_{\text{Dir}}$ boundary on the RHS of the above equation. Thus, from the RHS we read that the resulting $\mathsf{TY}(\mathbb{Z}_N)$-symmetric gapped phase decomposes as (424) phase in terms of the $\mathbb{Z}_N$ subsymmetry. This matches what we discussed above.

The S line changes the left boundary condition for the DW theory as

$$\mathcal{A}_{\text{Dir}} \to \mathcal{A}_{\text{Neu}}\,, \tag{432}$$

thus acting on the sub-phases as

$$
\begin{aligned}
(\mathcal{A}_{\text{Dir}}, \mathcal{A}_{\text{Neu}(\mathbb{Z}_p)}) &\to (\mathcal{A}_{\text{Neu}}, \mathcal{A}_{\text{Neu}(\mathbb{Z}_p)})\,, \\
(\mathcal{A}_{\text{Dir}}, \mathcal{A}_{\text{Neu}(\mathbb{Z}_q)}) &\to (\mathcal{A}_{\text{Neu}}, \mathcal{A}_{\text{Neu}(\mathbb{Z}_q)})\,.
\end{aligned}
\tag{433}
$$

Combining it with the $\mathbb{Z}_2$ gauge transformation discussed above, we can describe the action of S as a permutation of two sub-phases

$$(\mathcal{A}_{\text{Dir}}, \mathcal{A}_{\text{Neu}(\mathbb{Z}_p)}) \longleftrightarrow (\mathcal{A}_{\text{Dir}}, \mathcal{A}_{\text{Neu}(\mathbb{Z}_q)})\,. \tag{434}$$

This also matches what we discussed above.

**Order parameters.** The order parameters of these phases are captured in the respective Lagrangian algebras. There are various possibilities:

- If $\boldsymbol{Q}_{i,j}$ appears in $\mathcal{A}_{\mathrm{Neu}(\mathbb{Z}_p, \mathbb{Z}_q)}$ and $L_{i,j}$ appears in the algebra $\mathcal{A}_{\mathrm{Neu}(\mathbb{Z}_p)}$ of the DW theory, then we have a multiplet in the IR theory carrying generalized charge $\boldsymbol{Q}_{i,j}$, which involves a local operator

$$\mathcal{O}_{i,j}^{(e)} = \sqrt{2}\left(\sum_{k=1}^{q} \omega_q^{ik} v_{p-1+k}\right), \tag{435}$$

viewed as living at the end of $D_1^{(\mathsf{A}^j)}$ and a local operator

$$\mathcal{O}_{i,j}^{(e,t)} = \sqrt{2}\left(\sum_{k=1}^{p} \omega_p^{jk} v_{k-1}\right), \tag{436}$$

viewed as living at the end of $D_1^{(\mathsf{A}^i)}$.

- If $\boldsymbol{Q}_{i,j}$ appears in $\mathcal{A}_{\mathrm{Neu}(\mathbb{Z}_p, \mathbb{Z}_q)}$ and $L_{i,j}$ appears in the algebra $\mathcal{A}_{\mathrm{Neu}(\mathbb{Z}_q)}$ of the DW theory, then we have a multiplet in the IR theory carrying generalized charge $\boldsymbol{Q}_{i,j}$, which involves a local operator

$$\mathcal{O}_{i,j}^{(m,t)} = \sqrt{2}\left(\sum_{k=1}^{q} \omega_q^{jk} v_{p-1+k}\right), \tag{437}$$

viewed as living at the end of $D_1^{(\mathsf{A}^i)}$ and a local operator

$$\mathcal{O}_{i,j}^{(m)} = \sqrt{2}\left(\sum_{k=1}^{p} \omega_p^{ik} v_{k-1}\right), \tag{438}$$

viewed as living at the end of $D_1^{(\mathsf{A}^j)}$.
Note that if $L_{i,j}$ appears in both $\mathcal{A}_{\mathrm{Neu}(\mathbb{Z}_p)}$ and $\mathcal{A}_{\mathrm{Neu}(\mathbb{Z}_q)}$, then we have two possible IR multiplets with generalized charge $\boldsymbol{Q}_{i,j}$ described above.

- If $\boldsymbol{Q}_{i,+}$ appears in $\mathcal{A}_{\mathrm{Neu}(\mathbb{Z}_p, \mathbb{Z}_q)}$, then we have a multiplet in the IR theory carrying generalized charge $\boldsymbol{Q}_{i,+}$, which involves a local operator

$$\mathcal{O}_{i,+} = \sqrt{2}\left(\sum_{k=1}^{q} \omega_q^{ik} v_{p-1+k} + \sum_{k=1}^{p} \omega_p^{ik} v_{k-1}\right), \tag{439}$$

viewed as living at the end of $D_1^{(\mathsf{A}^i)}$.

- If $\boldsymbol{Q}_{i,-}$ appears in $\mathcal{A}_{\mathrm{Neu}(\mathbb{Z}_p, \mathbb{Z}_q)}$, then we have a multiplet in the IR theory carrying generalized charge $\boldsymbol{Q}_{i,-}$, which involves a local operator

$$\mathcal{O}_{i,-} = \sqrt{2}\left(\sum_{k=1}^{q} \omega_q^{ik} v_{p-1+k} - \sum_{k=1}^{p} \omega_p^{ik} v_{k-1}\right), \tag{440}$$

viewed as living at the end of $D_1^{(\mathsf{A}^i)}$.

### 7.3.3 Example: $\mathsf{TY}(\mathbb{Z}_4)$

We now discuss in detail the case $N = 4$, illustrating the additional structure that appears when $N$ is a perfect square. The Lagrangian algebras in addition to the $\mathcal{A}_{\mathrm{Dir,Neu}}$ are listed in (389). To see that the algebra $\mathcal{A}_{\mathrm{Duality}}$ is consistent, note first that the spin of the lines is 1.[24]

---

[24]We use the formula in appendix D of [23], although caution in identifying these lines with $\boldsymbol{Q}_{\Sigma_{e,\epsilon}}$ needs to be taken.

**$(\mathbb{Z}_2, \mathbb{Z}_2)$ SSB phase.** As an example consider $(\mathbb{Z}_2, \mathbb{Z}_2)$ SSB phase for $\mathsf{TY}(\mathbb{Z}_4)$. The vacua are

$$
\begin{aligned}
v_0 &= \frac{1}{4}\left(1 - \mathcal{O}_- \sqrt{2}\mathcal{O}^{(1)}\right), \\
v_1 &= \frac{1}{4}\left(1 - \mathcal{O}_+ \sqrt{2}\mathcal{O}^{(1)}\right), \\
v_2 &= \frac{1}{4}\left(1 + \mathcal{O}_- \sqrt{2}\mathcal{O}^{(2)}\right), \\
v_3 &= \frac{1}{4}\left(1 + \mathcal{O}_+ \sqrt{2}\mathcal{O}^{(2)}\right).
\end{aligned}
\tag{441}
$$

We have two multiplets in the IR carrying generalized charge $\boldsymbol{Q}_{2,0}$. One multiplet consists of an untwisted sector local operator $\mathcal{O}_{2,0}^{(e)} = \mathcal{O}^{(2)}$ and an operator

$$
\mathcal{O}_{2,0}^{(e,t)} = \sqrt{2}(v_0 + v_1),
\tag{442}
$$

viewed as an operator living at the end of $D_1^{(A^2)}$. The other multiplet consists of an untwisted sector local operator $\mathcal{O}_{2,0}^{(m)} = \mathcal{O}^{(1)}$ and an operator

$$
\mathcal{O}_{2,0}^{(m,t)} = \sqrt{2}(v_2 + v_3),
\tag{443}
$$

viewed as an operator living at the end of $D_1^{(A^2)}$. We also have a multiplet in the IR carrying generalized charge $\boldsymbol{Q}_{2,+}$ realized by the operator

$$
\mathcal{O}_{2,+} = \sqrt{2}(v_0 - v_1 + v_2 - v_3),
\tag{444}
$$

viewed as living at the end of $D_1^{(A^2)}$, and a multiplet in the IR carrying generalized charge $\boldsymbol{Q}_{2,-}$ realized by the operator

$$
\mathcal{O}_{2,-} = \sqrt{2}(v_0 - v_1 - v_2 + v_3),
\tag{445}
$$

viewed as living at the end of $D_1^{(A^2)}$. Finally, we have a multiplet in the IR carrying generalized charge $\boldsymbol{Q}_0^-$ realized by the untwisted sector operator $\mathcal{O}_{0,-} = \mathcal{O}_-$.

**$\mathbb{Z}_2$ SSB phase.** For $N = 4$, which is a perfect square, we have a further Lagrangian algebra involving the twist defects $\boldsymbol{Q}_{\Sigma_{1,+}}$ and $\boldsymbol{Q}_{\Sigma_{3,-}}$

$$
\mathcal{A}_{\text{Duality}(\mathbb{Z}_2)} = \boldsymbol{Q}_{0,+} + \boldsymbol{Q}_{2,-} + \boldsymbol{Q}_{2,0} + \boldsymbol{Q}_{\Sigma_{1,+}} + \boldsymbol{Q}_{\Sigma_{3,-}}.
\tag{446}
$$

We now determine the gapped phase obtained by selecting this as physical boundary

$$
\mathfrak{B}^{\text{phys}} = \mathcal{A}_{\text{Duality}(\mathbb{Z}_2)}.
\tag{447}
$$

From the SymTFT perspective, we find that the only lines that can end on both boundaries are $\boldsymbol{Q}_0^+$ and $\boldsymbol{Q}_{2,0}$. Let us denote the non-trivial untwisted sector local operator arising from $\boldsymbol{Q}_{2,0}$ as $\mathcal{O}_2$. Since $\mathcal{O}_2^2 = 1$, we can straightforwardly determine the two resulting vacua to be

$$
v_0 = \frac{1 + \mathcal{O}_2}{2}, \qquad v_1 = \frac{1 - \mathcal{O}_2}{2}.
\tag{448}
$$

The linking action of $D_1^{(A)}$ on $\mathcal{O}_2$ is

$$
\mathcal{O}_2 \to -\mathcal{O}_2,
\tag{449}
$$

from which we can derive its linking action on the vacua

$$\nu_0 \longleftrightarrow \nu_1 \,. \tag{450}$$

Notice that $D_1^{(A^2)}$ instead acts as the identity on the vacua. Therefore, from the perspective of the $\mathbb{Z}_4$ subsymmetry of $\mathsf{TY}(\mathbb{Z}_4)$, we identify this as a $\mathbb{Z}_2$ **SSB phase**. The line operator $D_1^{(A)}$ can be expressed as

$$D_1^{(A)} = \mathbf{1}_{01} \oplus \mathbf{1}_{10} \,. \tag{451}$$

The linking action of $D_1^{(S)}$ on the untwisted operators is

$$
\begin{aligned}
1 &\to 2\,, \\
\mathcal{O}_2 &\to 0\,,
\end{aligned}
\tag{452}
$$

from which we can derive its linking action on the vacua

$$
\begin{aligned}
\nu_0 &\to 1 = \nu_0 + \nu_1\,, \\
\nu_1 &\to 1 = \nu_0 + \nu_1\,.
\end{aligned}
\tag{453}
$$

Therefore we can express $D_1^{(S)}$ as

$$D_1^{(S)} = \mathbf{1}_{00} \oplus \mathbf{1}_{01} \oplus \mathbf{1}_{10} \oplus \mathbf{1}_{11} \,. \tag{454}$$

Notice that despite the fact that we call this a $\mathbb{Z}_2$ SSB phase, also $\mathsf{S}$ is spontaneously broken in this phase, since by acting with it on one vacuum we can reach the other one. Nevertheless, $\nu_0$ and $\nu_1$ remain indistinguishable, and correspondingly we do not have non-trivial Euler terms. Therefore this is an instance where spontaneous breaking of non-invertible symmetry does not lead to physically distinguishablee vacua, in sharp contrast with what happens in the $(\mathbb{Z}_1, \mathbb{Z}_4)$ SSB phase and the $(\mathbb{Z}_2, \mathbb{Z}_2)$ SSB phase.

This phase has three types of order parameters. The first one has generalized charge $\mathbf{Q}_{2,0}$ and is a mixture of untwisted and string-like order parameters. The IR image of such an order parameter contains the untwisted operator $\mathcal{O}_2$ discussed above, and $O_2^A$, which is $\mathcal{O}_2$ regarded as the endpoint of $D_1^{(A^2)} \simeq D_1^{(id)}$. The second type of order parameters has generalized charge $\mathbf{Q}_{2,-}$ and it is purely of string-type. The IR image of such an order parameter is the identity local operator $\mathcal{O}_2^-$ regarded as the endpoint of $D_1^{(A^2)} \simeq D_1^{(id)}$. Finally, the third kind of order parameters is also purely of string-type and has generalized charges $\mathbf{Q}_{\Sigma_{1,+}}$ and $\mathbf{Q}_{\Sigma_{3,-}}$, containing local operators at the end of $D_1^{(S)}$.

# Acknowledgments

We thank Andrea Antinucci, Federico Bonetti, Mathew Bullimore, Nils Carqueville, Jing-Yuan Chen, Christian Copetti, Giovanni Galati, Jonathan Heckman, Wenjie Ji, Justin Kaidi, Anatoly Konechny, Giovanni Rizi, Apoorv Tiwari, Gerard Watts, Jingxiang Wu and Yunqin Zheng for discussions. LB thanks the "Triangle Seminar" in London, the "FPUK 2024" meeting in Durham, and the "Categorical Aspects of Symmetry Program" at Nordita (funded by the Simons Collaboration on Global Categorical Symmetries) for stimulating interactions.

**Funding information**  This work is supported in part by the European Union's Horizon 2020 Framework through the ERC grants 682608 (LB, SSN) and 787185 (LB). LB is also funded as a Royal Society University Research Fellow through grant URF\R1\231467. The work of SSN is supported by the UKRI Frontier Research Grant, underwriting the ERC Advanced Grant "Generalized Symmetries in Quantum Field Theory and Quantum Gravity" and the Simons Foundation Collaboration on "Special Holonomy in Geometry, Analysis, and Physics", Award ID: 724073, Schäfer-Nameki.

# A  Details for the $\mathrm{Rep}(S_3)$ and Ising **generalized charges**

In this appendix, we use various consistency conditions to derive the symmetry action of a symmetry $\mathcal{S} \in \{\mathrm{Rep}(S_3), \mathrm{Ising}\}$ on the generalized charges transforming in multiplets of $\mathcal{S}$. These derivations have been omitted in the main body of the text because they are quite bulky, but we include them here to showcase how the charges can be fixed using an intrinsic 2d perspective. The section is divided into two parts: the first dealing with the $\mathrm{Rep}(S_3)$ charges from Section 5 and the second with the Ising charges from Section 6.

## A.1  $\mathrm{Rep}(S_3)$ charges

$\boldsymbol{Q_{[\mathbf{id}],\mathrm{P}}}$ **multiplet.**  Here we derive the $\mathrm{Rep}(S_3)$ action on the multiplet with generalized charge $\boldsymbol{Q_{[\mathbf{id}],P}}$. This multiplet consists of a single local operator $\mathcal{O}^P$ in the twisted sector for $\mathrm{P} \in \mathrm{Rep}(S_3)$

$$\mathrm{P} \longrightarrow \bullet \atop \mathcal{O}^P \tag{A.1}$$

The action of $\mathrm{Rep}(S_3)$ turns out to be

$$\tag{A.2}$$

Let us derive this below.

By looking at possible quadrivalent junctions in this case as in (241), we conclude that only possible actions of the symmetry generators P and E of $\text{Rep}(S_3)$ on $\mathcal{O}^P$ can be of the form

$$\text{(A.3)}$$

where $\alpha, \beta \in \mathbb{C} - \{0\}$. As P is the generator of (a non-anomalous) $\mathbb{Z}_2$ subsymmetry of $\text{Rep}(S_3)$, one must find that $\alpha^2 = 1$. To further pin down the value of $\alpha$, one can look at the action of P followed by E, which produces the following consistency condition

$$\text{(A.4)}$$

One then finds $\alpha\beta = \beta$, while $\alpha^2 = 1$, hence $\alpha = 1$ is the solution (since $\beta \neq 0$).

Now in order to prove the second part of the claim, it is useful to derive the following identity (where we denote a P line as a dashed line and an identity line **1** as a dotted line for easy readability)

$$(A.5)$$

To derive the above identity, we used insertions of the completeness relation

$$(A.6)$$

and F-moves at various places. In particular, the non-trivial $-$ sign in the last diagram on the RHS follows from $(F^{E}_{EPE})_{EE} = -1$. The F-symbols for $\mathrm{Rep}(S_3)$ are discussed e.g. in [139]. An alternative method relies on noticing that we can also obtain $\mathrm{Rep}(S_3)$ by gauging $S_3$. We discuss this approach in appendix C.

To pin down $\beta$ as well, we consider acting with E twice and use the identity we have just derived



$$\tag{A.7}$$

Hence we find $\beta^2 = 1$ and $\beta = -\beta^2$ which is consistent with $\beta = -1$ and the claim is proved.

**$Q_{[a],1}$ multiplet.** The action of the $\text{Rep}(S_3)$ lines on the local operators $\mathcal{O}_\pm^a$ can be derived using consistency with the fusion properties of symmetry lines and their junctions, as well as the F-symbols. We will now show explicitly how to determine the coefficients $-1/2$ and $(\omega^2 + 1/2)$ in the case of the E action on the $a$-multiplet, as it is quite instructive and paradigmatic of the other cases. As it is known, the typical action of a non-invertible symmetry line, such as E, involves a map between twisted and untwisted local operators in the same multiplet. Therefore, the most generic E action we can have is

$$\tag{A.8}$$

for some coefficients $\alpha, \beta, \gamma, \delta \in \mathbb{C}$ that we need to determine. In order to do this, we consider the action of two E lines on the untwisted operator $\mathcal{O}_+^a$. We can first consider acting with the

two E lines sequentially, which produces the following result:

$$
\begin{aligned}
\bullet \;\; \Big|\Big| \; &= \; \Big( \alpha \; \Big| \; \bullet + \beta \; \overline{\phantom{P}}^{\,P}\!\!\!\bullet \Big) \Big| \\
&\qquad\qquad\quad E\;E \qquad\qquad E \qquad\quad E \qquad\qquad E
\end{aligned}
$$

$$
= \; \alpha \; \Big| \Big( \alpha \; \Big| \bullet + \beta \; \overline{\phantom{P}}^{\,P}\!\!\bullet \Big) + \beta \Big( \gamma \; \overline{\phantom{P}}^{\,P}\!\! \; \bullet + \delta \; \overline{\phantom{P}}^{\,P}\,\overline{\phantom{P}}^{\,P}\!\!\bullet \Big)
$$

(A.9)

$$
= \; \alpha^2 \Big( \; \big|+\big|+\big| \; \Big) \; \bullet \; + \alpha\beta \Big( \; \underset{P}{\rule{0pt}{1em}\sqsubset}\!\bullet + \overline{\phantom{P}}^{\,P}\!\bullet - \underset{P}{\rule{0pt}{1em}\longrightarrow}\!\bullet \Big)
$$

$$
+ \; \beta\gamma \Big( \; \big|+\big|-\big| \; \Big) \; \bullet \; + \beta\delta \Big( \; \underset{P}{\sqsubset}\!\bullet + \overline{\phantom{P}}^{\,P}\!\bullet + \underset{P}{\longrightarrow}\!\bullet \Big) \,.
$$

In the above, we used the following identities satisfied by the Rep($S_3$) symmetry lines

$$
\overline{\phantom{P}}^{\,P}\big| = \big|+\big|-\big| \;, \qquad \big|\overline{\phantom{P}}^{\,P} = \sqsubset^{P} + \overline{\phantom{P}}^{\,P} - \big|^{P} \;,
$$

(A.10)

$$
\overline{\phantom{P}}\overline{\phantom{P}}^{\,P} = \sqsubset^{P} + \overline{\phantom{P}}^{\,P} + \big|^{P}
$$

These identities can be derived using the F-symbols of the Rep($S_3$) fusion category, in a similar fashion to (A.5), or with the methods employed in appendix C.

Alternatively, we could first fuse the two E lines and then act on $\mathcal{O}_+^a$. This gives the following:

$$
\bullet \Big( \; \big|+\big|+\big| \; \Big) = \bullet + \big| \; \bullet + \alpha \; \big| \; \bullet + \beta \; \overline{\phantom{P}}^{\,P}\!\!\bullet
$$

(A.11)

The two procedures should give the same result, which gives us the identities

$$
\alpha^2 + \beta\gamma = 1\,, \qquad \alpha^2 - \beta\gamma = \alpha\,, \qquad \alpha\beta + \beta\delta = 0\,, \qquad -\alpha\beta + \beta\delta = \beta\,. \tag{A.12}
$$

From this, we derive

$$
\alpha = -\delta = -\frac{1}{2}\,, \qquad \beta\gamma = \frac{3}{4}\,. \tag{A.13}
$$

To pin down $\beta$ and $\gamma$ individually, note that we can rescale

$$
\begin{aligned}
\mathcal{O}_+^a &\to r\mathcal{O}_+^a\,, \\
\mathcal{O}_-^a &\to s\mathcal{O}_-^a\,,
\end{aligned} \tag{A.14}
$$

which does not change the above conditions we found, but can be used to fix $\beta$ and $\gamma$ to any allowed value satisfying the condition $\beta\gamma = 3/4$. We can furthermore impose rotation

invariance on the above actions, using the fact that the bulk line $Q_{[a],1}$ has spin 1, which implies

$$\begin{array}{c}\text{P} \\ \vdash\!\!\!-\!\!\bullet \\ \text{E}\end{array} = -\beta \bullet \left|\begin{array}{c} \\ \text{E}\end{array}\right. -\alpha \bullet \begin{array}{c}\text{P} \\ -\!\!\!-\!\!\dashv \\ \text{E}\end{array} \tag{A.15}$$

Combining the above with the standard action (A.8), we find

$$\bullet\left|\begin{array}{c} \\ \text{E}\end{array}\right. = \alpha^2 \bullet \left|\begin{array}{c} \\ \text{E}\end{array}\right. + \alpha\beta \begin{array}{c}\text{P} \\ \bullet\!\!-\!\!\dashv \\ \text{E}\end{array} -\beta^2 \bullet\left|\begin{array}{c} \\ \text{E}\end{array}\right. - \alpha\beta \begin{array}{c}\text{P} \\ \bullet\!\!-\!\!\dashv \\ \text{E}\end{array} \tag{A.16}$$

from which we finally find

$$\alpha^2 - \beta^2 = 1 \implies \beta^2 = -\frac{3}{4} \implies \beta = \pm\left(\omega + \frac{1}{2}\right). \tag{A.17}$$

We can further remove the sign from $\beta$ by rescaling $\mathcal{O}_\pm^a$.

$Q_{[b],+}$ **multiplet.** Here we consider the $\mathrm{Rep}(S_3)$ action on the multiplet $Q_{[b],+}$. In particular, as a consistency check, we show that the actions we employed in the main text (256), (257) and (258) are consistent with the composition of symmetries. In particular, we focus on $\mathcal{O}^b$ and $\mathcal{O}_+^b$ and we show explicitly how to determine the following fusion coefficients

$$\begin{array}{c}\bullet \\ \mathcal{O}^b \\ \text{P}\end{array} \left|\begin{array}{c} \\ \\ \end{array}\right. = \alpha \left|\begin{array}{c}\bullet \\ \mathcal{O}^b, \\ \text{P}\end{array}\right. \qquad \begin{array}{c}\bullet \\ \mathcal{O}^b \\ \text{E}\end{array}\left|\begin{array}{c} \\ \\ \end{array}\right. = \beta \begin{array}{c}\text{E} \\ \vdash\!\!-\!\!\bullet \ \mathcal{O}_+^b, \\ \text{E}\end{array}$$

$$\begin{array}{c}\text{E} \\ -\!\!\bullet \\ \mathcal{O}^b \\ \text{E}\end{array}\left|\begin{array}{c} \\ \\ \end{array}\right. = \gamma \begin{array}{c}\text{E} \\ \neg\!\cdots\!\bullet \ \mathcal{O}_+^b \\ \text{E}\end{array} + \delta \begin{array}{c}\text{E} \ \ \text{P} \\ \neg\!\!-\!\!-\!\bullet \ \mathcal{O}_+^b \\ \text{E}\end{array} + \epsilon \begin{array}{c}\text{E} \\ -\!\!\dashv \\ \text{E}\end{array} \bullet \ \mathcal{O}^b. \tag{A.18}$$

Again, as a consistency condition to fix the coefficients we use the fact that we can either bring the symmetry lines past the local operator and fuse them, or first fuse them and then bring them past the local operator. We then first use the fusion $\mathrm{P} \otimes E = E$. Fusing the lines on the LHS of $\mathcal{O}^b$ gives

$$\bullet\left|\begin{array}{cc} & \\ \text{P} & \text{E}\end{array}\right. = \alpha \left|\begin{array}{cc}\bullet \\ \text{P} \ \ \text{E}\end{array}\right. = \alpha\beta\left|\begin{array}{c}\text{E} \\ \vdash\!\!-\!\!\bullet \\ \text{P} \ \ \text{E}\end{array}\right. = -\alpha\beta\begin{array}{c}\text{E} \\ \vdash\!\!-\!\!\bullet \\ \text{E}\end{array} \tag{A.19}$$

where we used the following identity between the $\mathrm{Rep}(S_3)$ symmetry lines

$$\left|\begin{array}{c}\text{E} \\ \vdash\!\!\!- \\ \text{P} \ \ \text{E}\end{array}\right. = - \begin{array}{c}\text{E} \\ \vdash\!\!\!- \\ \text{E}\end{array} \tag{A.20}$$

Fusing the lines on the RHS simply gives the E action. Equating the two we get

$$-\alpha\beta = \beta \implies \alpha = -1. \tag{A.21}$$

Note this is also consistent with the fusion $P \otimes P = \mathbf{1}$, which fixes $\alpha^2 = 1$. We remark that this consistency condition also discards the possibility of having E mapping $\mathcal{O}^b$ to both the untwisted $\mathcal{O}^b$ and the twisted $\mathcal{O}^b_+$ local operators (see the top right of (A.18)). Indeed, suppose that the E action contained also an untwisted piece with generic coefficient $\gamma$. Then using (A.19), we would obtain

$$-\alpha\beta = \beta\,, \qquad \alpha\gamma = \gamma\,. \tag{A.22}$$

Since $\alpha \neq 0$, this is consistent only if either $\beta$ or $\gamma$ are 0. In a similar fashion, the conditions derived from $E \otimes E$ fusion, which we employ next, do not allow for the possibility of having an 'E-channel' in the action of $E$ on $\mathcal{O}^b_+$ (see the bottom of (A.18)).

Now we use the fusion $E \otimes E = \mathbf{1} \oplus P \oplus E$. This gives

$$\bullet \;\; \Big|\;\Big| = \beta \;\; \Big|\!-\!\Big|\bullet = \beta \left( \gamma \;\; \boxed{E}\!\Big|_E\bullet \;+\; \delta \;\; \boxed{E\;\;P}\Big|_E\bullet \;+\; \epsilon \;\; \boxed{E}\Big|\bullet \right) \tag{A.23}$$

Fusing first the E lines produces instead

$$\bullet \left( \Big|+\Big|+\Big| \right) = \bullet - \Big|\;\bullet \;+\; \beta \Big|\!-\!\boxed{E}\bullet \tag{A.24}$$

The first two terms on the RHS above are matched by the contribution of last term in (A.23), which fixes

$$\beta\epsilon = 1\,. \tag{A.25}$$

The last term on the RHS above is instead matched by the sum of the first two terms in (A.23), which fixes

$$\gamma = \delta\,, \qquad \beta\gamma = \beta \Longrightarrow \gamma = 1\,. \tag{A.26}$$

We are left with only an overall coefficient $\beta$ to fix, which we can set to $\beta = 1$ by rescaling the local operator $\mathcal{O}^b$.

## A.2 Ising charges

We are first going to determine some useful relations involving the Ising symmetry lines that we are going to need to determine the charges. The first one is

$$\boxed{P}\;\Big| = \Big| - \Big| \tag{A.27}$$

To determine this relation, we again use insertions of the completeness relation (A.6) and F-moves. Namely, using (A.6) twice (A.27) becomes



$$\tag{A.28}$$

where the $-$ sign in front of the second diagram comes from the non-trivial F-symbol $F^P_{SPS} = -1$. This implies (A.27).

We will also need the relation

$$
\left| \quad \begin{array}{c} P \\ \rule{0pt}{1em} \\ \end{array} \right| = - \left| \begin{array}{c} P \\ \rule{0pt}{1em} \\ \end{array} \right|
\qquad\qquad\qquad \text{(A.29)}
$$
$$
\quad S \quad\quad S \qquad\qquad S
$$

which again can be easily derive inserting (A.6) in the above diagram and using an F-move, with $F^P_{SPS} = -1$.

**$Q_{id,-}$-multiplet:**  This multiplet consists of a single local operator $\mathcal{O}$. Determining the Ising action is straightforward. The only possibility is

$$
\begin{array}{c} \bullet \\ \mathcal{O}^- \end{array} \Big| \;\; = \;\; \alpha \, \Big| \; \begin{array}{c} \bullet \\ \mathcal{O}^- \end{array} ,
$$
$$
\qquad\qquad P \qquad\qquad\qquad P
$$

$$
\text{(A.30)}
$$

$$
\begin{array}{c} \bullet \\ \mathcal{O}^- \end{array} \Big| \;\; = \;\; \beta \, \Big| \; \begin{array}{c} \bullet \\ \mathcal{O}^- \end{array}
$$
$$
\qquad\qquad S \qquad\qquad\qquad S
$$

for two coefficients $\alpha, \beta \in \mathbb{C}$ that we need to determine. The fusion rule $P \otimes P = \mathbf{1}$ fixes $\alpha^2 = 1$, and $P \otimes S = S$ then selects $\alpha = 1$. Then the fusion $S \otimes S = \mathbf{1} \oplus P$ gives $\beta^2 = 1$. The solution $\beta = 1$ is the trivial charge, while $\beta = -1$ gives the non-trivial charge being discussed here.

**$Q_{id+P}$-multiplet**  This multiplet consists of an untwisted local operator $\mathcal{O}_+$ and a local operator $\mathcal{O}_-$ in the twisted sector of the P line. We now determine the Ising lines action on $\mathcal{O}_+$, with $\mathcal{O}_-$ that can be worked out analogously. The most general action we can have is

$$
\begin{array}{c} \bullet \\ \mathcal{O} \end{array} \Big| \;\; = \;\; \alpha \, \Big| \; \bullet \;\; , \qquad\qquad \begin{array}{c} \bullet \\ \mathcal{O} \end{array} \Big| \;\; = \;\; \beta \begin{array}{c} P \\ \rule{0pt}{1em} \bullet \end{array} \mathcal{O}^P \; + \; \gamma \, \Big| \; \bullet \; \mathcal{O} \, .
$$
$$
\quad P \qquad\qquad P \qquad\qquad\qquad\qquad S \qquad\qquad S \qquad\qquad E
\qquad\qquad \text{(A.31)}
$$

As usual, the P fusion with itself fixes $\alpha^2 = 1$. Now we consider the $P \otimes S = S$ fusion. We obtain

$$
\begin{array}{c} \bullet \\ \mathcal{O} \end{array} \Big| \; \Big| \;\; = \;\; \alpha \, \Big| \; \begin{array}{c} \bullet \\ \mathcal{O} \end{array} \Big| \;\; = \;\; \alpha\beta \, \Big| \begin{array}{c} P \\ \rule{0pt}{1em} \bullet \\ \mathcal{O}^P \end{array} \; + \;\; \alpha\gamma \, \Big| \; \begin{array}{c} \bullet \\ \mathcal{O} \end{array} .
$$
$$
\qquad P \;\; S \qquad\qquad P \;\; S \qquad\qquad P \;\; S \qquad\qquad\qquad S
\qquad \text{(A.32)}
$$

By comparing with (A.31), we obtain

$$
\alpha\beta = -\beta \, , \qquad \alpha\gamma = \gamma \, ,
\qquad\qquad\qquad \text{(A.33)}
$$

where the first $-$ sign comes from using (A.29). This then tells us that we cannot have both a twisted and untwisted action of S on $\mathcal{O}_+$. Since we know that the non-invertible S line should map to a twisted sector, we choose $\beta \neq 0$ and $\gamma = 0$, and then the previous equation fixes

$\alpha = -1$. Finally, we consider the $S \otimes S$ fusion. For this, we also need the $S$ action on the twisted sector, which we expect to be of the form

$$
\begin{array}{cc}
\underset{\mathcal{O}^{\mathsf{P}}}{\overset{\mathsf{P}}{\longrightarrow}}\Big| & = \; \delta \quad \overset{\mathsf{P}}{\longrightarrow}\Big| \quad \bullet \\
\mathsf{E} & \mathsf{E} \quad \mathcal{O}
\end{array}
\tag{A.34}
$$

By acting with two $S$ sequentially we then obtain

$$
\begin{array}{cccc}
\bullet \\
\mathcal{O} \quad \Big|\,\Big| & = \; \beta \; \overset{\mathsf{P}}{\Big|}\bullet_{\mathcal{O}^{\mathsf{P}}}\Big| & = \; \beta\delta \; \overset{\mathsf{P}}{\Big|}\Big| \; \bullet \\
\mathsf{S}\;\mathsf{S} & \mathsf{S}\quad\mathsf{S} & \mathsf{S}\;\mathsf{S} \quad \mathcal{O}
\end{array}
\tag{A.35}
$$

Comparing with (A.31), this gives us

$$
\beta\delta = 1 \,.
\tag{A.36}
$$

Notice that this is consistent with the $\mathbf{1} \oplus \mathsf{P}$ action thanks to the identity (A.27). We are then left with an overall coefficient $\beta$ which we can set to 1 by rescaling $\mathcal{O}_+$, and this concludes the derivation of the desired action.

# B Lagrangian algebras for $\mathcal{Z}(\mathsf{Vec}_{S_3})$ and boundary conditions

As it is well known, the irreducible topological boundary conditions of the SymTFT are captured by Lagrangian algebras in the corresponding Drinfeld center. Here we aim to sketch how the choice of a specific Lagrangian algebra $\mathcal{A}$ in $\mathcal{Z}(\mathcal{S})$ determines the resulting symmetry category $\mathcal{S}_{\mathcal{A}}$ on the boundary, focusing on the non-abelian case $\mathcal{Z}(\mathsf{Vec}_{S_3})$, for which the procedure is more subtle than for the well-known abelian case. We will mainly follow [120]. For a more general discussion of anyon condensation see e.g. [140].

Physically, a Lagrangian algebra $\mathcal{A}$ represents a maximal choice of anyons of the bulk theory that can be simultaneously condensed. This describes which line operators of the SymTFT can terminate on the 2d topological boundary $\mathfrak{B}^{\mathsf{sym}}$. The remaining line operators that are not in $\mathcal{A}$ are said to be **confined**. Upon interval compactification, these will give rise to the topological line operators of the symmetry $\mathcal{S}_{\mathcal{A}}$ of the 2d QFT. Since we are dealing with a group-like case, we expect the Lagrangian algebras to be in 1-1 correspondence with possible discrete gaugings performed in the 2d boundary theory, which are given by a choice of subgroup $H \subseteq S_3$.[25] For $\mathcal{Z}(\mathsf{Vec}_{S_3})$ the possible Lagrangians are listed in (193).

To determine the boundary symmetry category $\mathcal{S}_{\mathcal{A}}$ in $\mathfrak{B}^{\mathsf{sym}}$ specified by choosing the bulk Lagrangian algebra $\mathcal{A}$, we make the following important observation: since the anyons in $\mathcal{A}$ are terminable, in the boundary symmetry category $\mathcal{S}_{\mathcal{A}}$ these are identified with the trivial line defects. Therefore we have the following identity

$$
\mathrm{Hom}_{\mathcal{S}_{\mathcal{A}}}(\boldsymbol{Q}_x, \boldsymbol{Q}_y) = \mathrm{Hom}_{\mathcal{Z}(\mathsf{Vec}_{S_3})}(\boldsymbol{Q}_x, \boldsymbol{Q}_y \otimes \mathcal{A}) \,,
\tag{B.1}
$$

where $\boldsymbol{Q}_x, \boldsymbol{Q}_y$ denote two generic simple objects of $\mathcal{Z}(\mathsf{Vec}_{S_3})$. In practice, this will imply the following:

- some of the simple objects of $\mathcal{Z}(\mathsf{Vec}_{S_3})$ will be identified in $\mathcal{S}_{\mathcal{A}}$;

- some of the simple objects of $\mathcal{Z}(\mathsf{Vec}_{S_3})$ are not going to be simple anymore in $\mathcal{S}_{\mathcal{A}}$, which means the they will undergo a *splitting* procedure.

---

[25]Notice that there is no possible discrete torsion $c \in H^2(H, U(1))$ in this case.

The fusion of simple objects in $\mathcal{S}_{\mathcal{A}}$ can be derived from the corresponding fusions in $\mathcal{Z}(\mathsf{Vec}_{S_3})$ using consistency conditions.

Notice that we always have

$$\mathrm{Hom}_{\mathcal{S}_{\mathcal{A}}}(\boldsymbol{Q}_{[\mathrm{id}],1}, \boldsymbol{Q}_{[\mathrm{id}],1}) = \mathbb{C}, \tag{B.2}$$

as $\mathcal{A}$ always contains $\boldsymbol{Q}_{[\mathrm{id}],1}$ with $n_{\mathrm{id}} = 1$. This guarantees the trivial line remains a simple object also in the boundary theory.

**Lagrangian algebra $\mathcal{A}_{\mathbf{Dir}}$.** Let us first illustrate how the above procedure works in the case of the Dirichlet Lagrangian algebra

$$\mathcal{A}_{\mathrm{Dir}} = \boldsymbol{Q}_{[\mathrm{id}],1} \oplus \boldsymbol{Q}_{[\mathrm{id}],P} \oplus 2\boldsymbol{Q}_{[\mathrm{id}],E}. \tag{B.3}$$

We claim that the corresponding 2d boundary theory has the symmetry category

$$\mathcal{S}_{\mathcal{A}_{\mathrm{Dir}}} = \mathsf{Vec}_{S_3}. \tag{B.4}$$

Therefore we expect to find from the 8 anyons of $\mathcal{Z}(\mathsf{Vec}_{S_3})$ (188) the 6 simple objects (183) of $\mathsf{Vec}_{S_3}$.

Let us now use (B.1) do determine the simple objects in $\mathcal{S}_{\mathcal{A}_{\mathrm{Dir}}}$. For example, we have

$$\begin{aligned} \mathrm{Hom}_{\mathcal{S}_{\mathcal{A}_{\mathrm{Dir}}}}(\boldsymbol{Q}_{[\mathrm{id}],1}, \boldsymbol{Q}_{[\mathrm{id}],P}) &= \mathrm{Hom}_{\mathcal{Z}(\mathsf{Vec}_{S_3})}(\boldsymbol{Q}_{[\mathrm{id}],1}, \boldsymbol{Q}_{[\mathrm{id}],P} \otimes \mathcal{A}_{\mathrm{Dir}}) \\ &= \mathrm{Hom}_{\mathcal{Z}(\mathsf{Vec}_{S_3})}(\boldsymbol{Q}_{[\mathrm{id}],1}, \boldsymbol{Q}_{[\mathrm{id}],P} \oplus \boldsymbol{Q}_{[\mathrm{id}],1} \oplus 2\boldsymbol{Q}_{[\mathrm{id}],E}) = \mathbb{C}, \end{aligned} \tag{B.5}$$

where we used that

$$\mathrm{Hom}_{\mathcal{Z}(\mathsf{Vec}_{S_3})}(\boldsymbol{Q}_{[\mathrm{id}],1}, \boldsymbol{Q}_{[\mathrm{id}],1}) = \mathbb{C}, \tag{B.6}$$

while there are no morphisms for $\boldsymbol{Q}_{[\mathrm{id}],1}$ to other simple objects. This tells us that on the boundary we have an identification

$$\boldsymbol{Q}_{[\mathrm{id}],P} \sim \boldsymbol{Q}_{[\mathrm{id}],1}. \tag{B.7}$$

We denote the corresponding trivial simple object in $\mathcal{S}_{\mathcal{A}_{\mathrm{Dir}}}$ by 1. Similarly, we have

$$\mathrm{Hom}_{\mathcal{S}_{\mathcal{A}_{\mathrm{Dir}}}}(\boldsymbol{Q}_{[\mathrm{id}],1}, \boldsymbol{Q}_{[\mathrm{id}],E}) = \mathbb{C}^2, \tag{B.8}$$

which implies $\boldsymbol{Q}_{[\mathrm{id}],E}$ is identified with two copies of the trivial line, namely $1 \oplus 1$. Now we can compute

$$\begin{aligned} \mathrm{Hom}_{\mathcal{S}_{\mathcal{A}_{\mathrm{Dir}}}}(\boldsymbol{Q}_{[b],+}, \boldsymbol{Q}_{[b],+}) &= \mathrm{Hom}_{\mathcal{Z}(\mathsf{Vec}_{S_3})}(\boldsymbol{Q}_{[b],+}, \boldsymbol{Q}_{[b],+} \otimes \mathcal{A}_{\mathrm{Dir}}) \\ &= \mathrm{Hom}_{\mathcal{Z}(\mathsf{Vec}_{S_3})}(\boldsymbol{Q}_{[b],+}, 3\boldsymbol{Q}_{[b],+} \oplus 3\boldsymbol{Q}_{[b],-}) = \mathbb{C}^3, \end{aligned} \tag{B.9}$$

and similarly

$$\mathrm{Hom}_{\mathcal{S}_{\mathcal{A}_{\mathrm{Dir}}}}(\boldsymbol{Q}_{[b],+}, \boldsymbol{Q}_{[b],-}) = \mathbb{C}^3, \qquad \mathrm{Hom}_{\mathcal{S}_{\mathcal{A}_{\mathrm{Dir}}}}(\boldsymbol{Q}_{[b],-}, \boldsymbol{Q}_{[b],-}) = \mathbb{C}^3. \tag{B.10}$$

This tells us that in $\mathcal{S}_{\mathcal{A}_{\mathrm{Dir}}}$

$$\boldsymbol{Q}_{[b],+} \sim \boldsymbol{Q}_{[b],-}, \tag{B.11}$$

and also that $\boldsymbol{Q}_{[b],+}$ is not a simple line anymore as its endomorphism vector space is $\mathbb{C}^3$. We then argue that in $\mathcal{S}_{\mathcal{A}_{\mathrm{Dir}}}$, $\boldsymbol{Q}_{[b],+}$ should split into 3 simple objects

$$b_1, b_2, b_3. \tag{B.12}$$

Completely analogously, we compute

$$\text{Hom}_{\mathcal{S}_{\mathcal{A}_{\text{Dir}}}}(\boldsymbol{Q}_{[a],1}, \boldsymbol{Q}_{[a],1}) = \text{Hom}_{\mathcal{S}_{\mathcal{A}_{\text{Dir}}}}(\boldsymbol{Q}_{[a],1}, \boldsymbol{Q}_{[a],\omega}) = \text{Hom}_{\mathcal{S}_{\mathcal{A}_{\text{Dir}}}}(\boldsymbol{Q}_{[a],1}, \boldsymbol{Q}_{[a],\omega^2})$$

$$\text{Hom}_{\mathcal{S}_{\mathcal{A}_{\text{Dir}}}}(\boldsymbol{Q}_{[a],\omega}, \boldsymbol{Q}_{[a],\omega}) = \text{Hom}_{\mathcal{S}_{\mathcal{A}_{\text{Dir}}}}(\boldsymbol{Q}_{[a],\omega}, \boldsymbol{Q}_{[a],\omega^2}) = \text{Hom}_{\mathcal{S}_{\mathcal{A}_{\text{Dir}}}}(\boldsymbol{Q}_{[a],\omega^2}, \boldsymbol{Q}_{[a],\omega^2}) = \mathbb{C}^2 .$$

(B.13)

This tells us that in $\mathcal{S}_{\mathcal{A}_{\text{Dir}}}$ we have the identifications

$$\boldsymbol{Q}_{[a],1} \sim \boldsymbol{Q}_{[a],\omega} \sim \boldsymbol{Q}_{[a],\omega^2} . \tag{B.14}$$

Moreover, we find that $\boldsymbol{Q}_{[a],1}$ is not a simple object in $\mathcal{S}_{\mathcal{A}_{\text{Dir}}}$, which means that it should split into two simple objects

$$a_1 , \quad a_2 . \tag{B.15}$$

In summary, in the boundary theory symmetry category corresponding to the Lagrangian algebra $\mathcal{L}_e$ we get 6 simple objects

$$\mathcal{C}_{\text{bdry}}^{\mathcal{L}_e} = \{1, b_1, b_2, b_3, a_1, a_2\} . \tag{B.16}$$

The fusions of these simple objects can be derived from the fusion of the corresponding anyons in $\mathcal{Z}(\text{Vec}_{S_3})$. For example, let us consider the simple objects $a_1, a_2$. The fusion

$$\boldsymbol{Q}_{[a],1} \otimes \boldsymbol{Q}_{[a],1} = \boldsymbol{Q}_{[\text{id}],1} \oplus \boldsymbol{Q}_{[\text{id}],P} \oplus \boldsymbol{Q}_{[a],1} , \tag{B.17}$$

in $\mathcal{Z}(\text{Vec}_{S_3})$ becomes in $\mathcal{S}_{\mathcal{A}_{\text{Dir}}}$

$$(a_1 \oplus a_2) \otimes (a_1 \oplus a_2) = 1 \oplus 1 \oplus a_1 \oplus a_2 . \tag{B.18}$$

Using associativity, the only consistent set of fusion rules we can derive from the above are

$$\begin{aligned} a_1 \otimes a_2 = a_2 \otimes a_1 &= 1 , \\ a_2 \otimes a_2 &= a_1 , \\ a_1 \otimes a_1 &= a_2 . \end{aligned} \tag{B.19}$$

Then we see that we can identify the three simple objects $\{1, a_1, a_2\}$ as forming the $\text{Vec}_{\mathbb{Z}_3}$ subcategory of $\text{Vec}_{S_3}$. Now let us consider the other simple objects $b_1, b_2, b_3$. For example, the fusions

$$\begin{aligned} \boldsymbol{Q}_{[a],1} \otimes \boldsymbol{Q}_{[b],+} &= \boldsymbol{Q}_{[b],+} \oplus \boldsymbol{Q}_{[b],-} , \\ \boldsymbol{Q}_{[b],+} \otimes \boldsymbol{Q}_{[b],+} &= \boldsymbol{Q}_{[\text{id}],1} \oplus \boldsymbol{Q}_{[\text{id}],E} \oplus \boldsymbol{Q}_{[a],1} \oplus \boldsymbol{Q}_{[a],\omega} \oplus \boldsymbol{Q}_{[a],\omega^2} , \end{aligned} \tag{B.20}$$

in $\mathcal{Z}(\text{Vec}_{S_3})$ become in $\mathcal{S}_{\mathcal{A}_{\text{Dir}}}$

$$(a_1 \oplus a_2) \otimes (b_1 \oplus b_2 \oplus b_3) = 2b_1 \oplus 2b_2 \oplus 2b_3 , \tag{B.21}$$

and

$$(b_1 \oplus b_2 \oplus b_3) \otimes (b_1 \oplus b_2 \oplus b_3) = 3 \, 1 \oplus 3 \, a_1 \oplus 3 \, a_2 , \tag{B.22}$$

respectively.[26] It can be checked explicitly that the only consistent set of associative fusions that can be derived from the above gives simple objects reproducing $\text{Vec}(S_3)$, with the identifications

$$1 \leftrightarrow 1 , \quad a_1 \leftrightarrow a , \quad a_2 \leftrightarrow a^2 , \quad b_1 \leftrightarrow b , \quad b_2 \leftrightarrow ab , \quad b_3 \leftrightarrow a^2 b . \tag{B.23}$$

---

[26]Notice all the other fusions in $\mathcal{Z}(\text{Vec}_{S_3})$ become trivial due to the identifications we made.

**Lagrangian algebra $\mathcal{A}_{\mathbf{Neu}(\mathbb{Z}_2)}$.** We now want to determine the boundary theory corresponding to the choice of Lagrangian algebra

$$\mathcal{A}_{\text{Neu}(\mathbb{Z}_2)} = \boldsymbol{Q}_{[\text{id}],1} \oplus \boldsymbol{Q}_{[\text{id}],E} \oplus \boldsymbol{Q}_{[b],+} \,. \tag{B.24}$$

We will argue that this boundary condition corresponds to the gauging of the $\mathbb{Z}_2 \subset S_3$ subgroup in $\text{Vec}(S_3)$.

Computing the Hom-spaces as in (B.1), we have the following non-trivial identifications

$$
\begin{aligned}
\boldsymbol{Q}_{[\text{id}],E} &\sim \boldsymbol{Q}_{[\text{id}],1} \oplus \boldsymbol{Q}_{[\text{id}],P} \,, & \boldsymbol{Q}_{[b],+} &\sim \boldsymbol{Q}_{[\text{id}],1} \oplus \boldsymbol{Q}_{[a],1} \,, \\
\boldsymbol{Q}_{[b],-} &\sim \boldsymbol{Q}_{[\text{id}],P} \oplus \boldsymbol{Q}_{[a],1} \,, & \boldsymbol{Q}_{[a],1} &\sim \boldsymbol{Q}_{[a],\omega} \sim \boldsymbol{Q}_{[a],\omega^2} \,.
\end{aligned}
\tag{B.25}
$$

The first identification comes from the fact that $\text{Hom}_{\mathcal{S}_{\mathcal{A}_{\text{Neu}(\mathbb{Z}_2)}}}(\boldsymbol{Q}_{[\text{id}],E}, \boldsymbol{Q}_{[\text{id}],E}) = \mathbb{C}^2$, meaning that $\boldsymbol{Q}_{[\text{id}],E}$ is not simple anymore, and the fact that we have morphisms from $\boldsymbol{Q}_{[\text{id}],E}$ to $\boldsymbol{Q}_{[\text{id}],1}$ and $\boldsymbol{Q}_{[\text{id}],P}$ tells us that this is the correct splitting. The discussion is similar for $\boldsymbol{Q}_{[b],+}$ and $\boldsymbol{Q}_{[b],-}$. In summary, we are left with only three simple objects in the boundary category, which we denote

$$1 \,, \quad P \,, \quad a \,. \tag{B.26}$$

Notice that contrary to the previous case, here we have no splitting of simple objects, which means that the fusions in $\mathcal{S}_{\mathcal{A}_{\text{Neu}(\mathbb{Z}_2)}}$ can be derived directly from the fusions in $\mathcal{Z}(\text{Vec}_{S_3})$. In particular, we have the non-trivial fusions

$$
\begin{aligned}
1 \otimes P &= P \,, \\
P \otimes P &= 1 \,, \\
P \otimes a &= a \,, \\
a \otimes a &= 1 \oplus P \oplus a \,.
\end{aligned}
\tag{B.27}
$$

We can naturally identify the boundary theory corresponding to $\mathcal{A}_{\text{Neu}(\mathbb{Z}_2)}$ as the theory obtained by performing a gauging of the $\mathbb{Z}_2$ subgroup of the $S_3$ global symmetry in $\text{Vec}_{S_3}$. Indeed, consider gauging the symmetry generated by $b$ in $\text{Vec}(S_3)$. Since $b$ acts by conjugation as

$$b \otimes a \otimes b = a^2 \,, \tag{B.28}$$

the only gauge invariant object that survives after gauging is

$$a = (a \oplus a^2)_{\text{Vec}_{S_3}} \,. \tag{B.29}$$

We also have a topological line $P$ generating the dual $\mathbb{Z}_2$ symmetry. The fusions of these objects can be computed easily by knowledge of the fusions in the pre-gauged category and match exactly what we found in (B.27) with our Lagrangian algebra argument.

Incidentally, notice that this symmetry category is also equivalent to $\text{Rep}(S_3)$. This can be understood in the following way. Start with a 2d theory $\mathfrak{T}$ with $S_3$ symmetry and consider gauging the $\mathbb{Z}_2 \subset S_3$ symmetry. The theory $\mathfrak{T}/\mathbb{Z}_2$ can be obtained also by first gauging the $\mathbb{Z}_3$ symmetry to go to $\mathfrak{T}/\mathbb{Z}_3$ and then gauging the full symmetry of that theory, which is $\widehat{\mathbb{Z}}_3 \rtimes \mathbb{Z}_2 \simeq S_3$, so clearly the symmetry category of $\mathfrak{T}/\mathbb{Z}_2$ must be $\text{Rep}(S_3)$. In this identification, the two-dimensional irrep of $S_3$, $E$, corresponds to the 'composite' object $a \oplus a^2$, while the sign irrep $P$ corresponds to the dual $\mathbb{Z}_2$ symmetry line.

**Lagrangian algebra $\mathcal{A}_{\mathbf{Neu}(\mathbb{Z}_3)}$.** We now consider the Lagrangian algebra

$$\mathcal{A}_{\text{Neu}(\mathbb{Z}_3)} = \boldsymbol{Q}_{[\text{id}],1} \oplus \boldsymbol{Q}_{[\text{id}],P} \oplus 2\boldsymbol{Q}_{[a],1} \,. \tag{B.30}$$

We argue that this boundary condition corresponds to the gauging of the $\mathbb{Z}_3$ subgroup in $\text{Vec}_{S_3}$. The symmetry category $\mathcal{S}_{\mathcal{A}_{\text{Neu}(\mathbb{Z}_3)}}$ is again as $\text{Vec}_{S_3}$ since $\mathbb{Z}_3$ is an abelian normal subgroup.

**Lagrangian algebra $\mathcal{A}_{\mathbf{Neu}}$.** We finally consider the Lagrangian algebra

$$\mathcal{A}_{\mathrm{Neu}} = \boldsymbol{Q}_{[\mathrm{id}],1} \oplus \boldsymbol{Q}_{[a],1} \oplus \boldsymbol{Q}_{[b],+}, \tag{B.31}$$

and argue that this boundary condition corresponds to the gauging of the full $S_3$ symmetry in $\mathrm{Vec}_{S_3}$, so that we expect

$$\mathcal{S}_{\mathcal{A}_{\mathrm{Neu}}} = \mathrm{Rep}(S_3). \tag{B.32}$$

Computing the Hom-spaces between simple objects as in (B.1), we obtain the following non-trivial identifications

$$
\begin{aligned}
\boldsymbol{Q}_{[a],\omega} &\sim \boldsymbol{Q}_{[a],\omega^2} \sim \boldsymbol{Q}_{[\mathrm{id}],E}, & \boldsymbol{Q}_{[a],1} &\sim \boldsymbol{Q}_{[\mathrm{id}],1} \oplus \boldsymbol{Q}_{[\mathrm{id}],P}, \\
\boldsymbol{Q}_{[b],+} &\sim \boldsymbol{Q}_{[\mathrm{id}],1} \oplus \boldsymbol{Q}_{[\mathrm{id}],E}, & \boldsymbol{Q}_{[b],-} &\sim \boldsymbol{Q}_{[\mathrm{id}],P} \oplus \boldsymbol{Q}_{[\mathrm{id}],E}.
\end{aligned}
\tag{B.33}
$$

In summary, we are left with 3 simple objects in the boundary symmetry category $\mathcal{S}_{\mathcal{A}_{\mathrm{Neu}}}$, which we denote

$$1, \quad P, \quad E. \tag{B.34}$$

The fusions of these simple objects can be straightforwardly derived from the respective fusions in the bulk theory. In particular, we have the non-trivial fusions

$$
\begin{aligned}
P \otimes P &= 1, \\
P \otimes E &= E, \\
E \otimes E &= 1 \oplus P \oplus E.
\end{aligned}
\tag{B.35}
$$

Hence we have recovered that the boundary symmetry category is indeed $\mathrm{Rep}(S_3)$.

# C  Rep($S_3$) via gauging $S_3$

As we have discussed in the main text, $\mathrm{Rep}(S_3)$ symmetry can be obtained by gauging $S_3$ symmetry. As a consequence, the structure of generalized charges and gapped phases for $\mathrm{Rep}(S_3)$ symmetry can be obtained from the structure of generalized charges and gapped phases for $S_3$ symmetry. We will discuss this gauging procedure in detail in this appendix. This analysis serves as a cross-check for the results for $\mathrm{Rep}(S_3)$ that we derived using the SymTFT approach in the main text.

## C.1  Rep($S_3$) symmetry from $S_3$ symmetry

Let us begin with the $S_3$ symmetry as in (183). We can perform gauging of this symmetry in two steps: first gauge the $\mathbb{Z}_3$ normal subgroup and then gauge the residual non-normal $\mathbb{Z}_2$ symmetry.

Since the first step involves gauging a normal subgroup, it is quite straightforward to implement. The dual symmetry after $\mathbb{Z}_3$ gauging is a copy of $\mathbb{Z}_3$ that we label as

$$\widehat{\mathbb{Z}}_3 = \{1, \widehat{a}, \widehat{a}^2\}, \tag{C.1}$$

which can be identified with the irreducible representations of the original $\mathbb{Z}_3$ subgroup generated by $a$. The residual $\mathbb{Z}_2$ symmetry is generated by $b$. Since conjugation by $b$ exchanges $a$ and $a^2$, it exchanges the action of $\mathbb{Z}_3$ on the representations in (C.1). Thus conjugation by $b$ also exchanges $\widehat{a}$ and $\widehat{a}^2$; and we learn that the dual symmetry $\widehat{Z}_3$ and the residual $\mathbb{Z}_2$ symmetry combine to form an $S_3$ symmetry that we label as

$$\widehat{S}_3 = \{\mathrm{id}, \widehat{a}, \widehat{a}^2, b, \widehat{a}b, \widehat{a}^2 b\}. \tag{C.2}$$

Now we gauge the residual $\mathbb{Z}_2$ symmetry generated by $b$ in $\widehat{S}_3$. The resulting symmetry is $\text{Rep}(S_3)$, with $\mathsf{P} \in \text{Rep}(S_3)$ generating the dual $\mathbb{Z}_2$ symmetry arising from the $\mathbb{Z}_2$ gauging. The non-invertible symmetry line $\mathsf{E} \in \text{Rep}(S_3)$ arises as a combination of $\widehat{a}$ and $\widehat{a}^2$ lines as follows

$$
\Bigg|_{\mathsf{E}} \; \equiv \; \Bigg\uparrow_{\widehat{a}} \; + \; \Bigg\uparrow_{\widehat{a}^2}
\tag{C.3}
$$

To see this, note that $b$ exchanges $\widehat{a}$ and $\widehat{a}^2$, so these lines are not gauge invariant anymore, but the sum of these lines remains gauge invariant and becomes a simple line. The difference of the identity local operators on lines $\widehat{a}$ and $\widehat{a}^2$ does not completely disappear from the spectrum. Instead, since it is charged non-trivially under $b$, it has to be attached to the $\mathbb{Z}_2$ Wilson line $\mathsf{P}$. Thus, it provides an end of the $\mathsf{P}$ line along the $\mathsf{E}$ line

$$
\begin{array}{c} \mathsf{E} \\[2pt] \Big|\!\!-\!\!- \; \mathsf{P} \\[2pt] \mathsf{E} \end{array} \; \equiv \; \Bigg\uparrow_{\widehat{a}} \; - \; \Bigg\uparrow_{\widehat{a}^2}
\tag{C.4}
$$

Let us define the end of $\mathsf{P}$ from the left in such a way that it is related to the end of $\mathsf{P}$ from the right by a 180°-rotation

$$
\mathsf{P} \; -\!\!-\!\!\Big|\begin{array}{c} \mathsf{E} \\[4pt] \\ \mathsf{E} \end{array} \; \equiv \; \Bigg\uparrow_{\widehat{a}^2} \; - \; \Bigg\uparrow_{\widehat{a}}
\tag{C.5}
$$

The ends of $\mathsf{E}$ on $\mathsf{E}$ can be described as follows:

$$
\begin{array}{c} \mathsf{E} \\ \Big|\!\!-\!\!- \; \mathsf{E} \\ \mathsf{E} \end{array} \; \equiv \; \begin{array}{c} \widehat{a}^2 \\ \Big\uparrow\!\!\leftarrow \; \widehat{a} \\ \widehat{a} \end{array} \; + \; \begin{array}{c} \widehat{a} \\ \Big\uparrow\!\!\leftarrow \; \widehat{a}^2 \\ \widehat{a}^2 \end{array}
$$

$$
\mathsf{E} \; -\!\!-\!\!\Big|\begin{array}{c} \mathsf{E} \\ \\ \mathsf{E} \end{array} \; \equiv \; \widehat{a}^2 \; \leftarrow\!\!\Big|\begin{array}{c} \widehat{a}^2 \\ \\ \widehat{a} \end{array} \; + \; \widehat{a} \; \leftarrow\!\!\Big|\begin{array}{c} \widehat{a} \\ \\ \widehat{a}^2 \end{array}
\tag{C.6}
$$

which are again related by a rotation. We define E corners as[27]

$$\begin{array}{ccc} & \quad\text{E} \quad & \equiv \quad \tfrac{1}{\sqrt{2}} \quad\widehat{a}\quad + \quad \tfrac{1}{\sqrt{2}}\quad \widehat{a}^2 \\ \text{E} & & \end{array} \tag{C.7}$$

Other corners are obtained by rotating this configuration. Finally, we have junctions

$$\begin{array}{c} \text{P} \\[4pt] \quad\text{E}\quad \equiv \quad \tfrac{1}{\sqrt{2}}\quad \widehat{a}^2 \quad - \quad \tfrac{1}{\sqrt{2}}\quad \widehat{a} \\[6pt] \text{E} \\[12pt] \text{E} \\[4pt] \quad\text{E}\quad \equiv \quad \tfrac{1}{\sqrt{2}}\quad \widehat{a}^2 \quad - \quad \tfrac{1}{\sqrt{2}}\quad \widehat{a} \\[6pt] \text{P} \end{array} \tag{C.8}$$

and their rotated versions. These junctions are good projectors onto various fusion channels. The reader can verify that they satisfy the identities

$$\text{E}\;\rangle\;\text{E} \;=\; 1\;,\qquad \text{E}\;\rangle\;\text{E}\;=\;\Big|\;,\qquad \text{E}\;\rangle\;\text{E}\;=\;\Big|\;,$$

$$\Big|\;\Big|\;=\;\bigcup\limits_{\cap}\;+\;\text{P}\;\bigg\}\;+\;\text{E}\;\bigg\} \tag{C.9}$$

---

[27]Note that we do not need to define corners for $\widehat{a}$ and $\widehat{a}^2$ lines for the analysis here. In case it is helpful, the reader may freely interchange smooth curved $\widehat{a}$ and $\widehat{a}^2$ lines, with $\widehat{a}$ and $\widehat{a}^2$ lines having a corner. However, note that one cannot freely interchange an E with a corner, with a smooth curved E line, because of the presence of the extra $1/\sqrt{2}$ factor present in the definition of the corner presented below. Such interchanges introduce factors of $\sqrt{2}$ and $1/\sqrt{2}$.

### C.2   Rep($S_3$) charges from $S_3$ charges

Let us now use the gauging perspective to derive the action of Rep($S_3$) on various multiplets discussed in section 5.2.

#### C.2.1   $Q_{[\mathrm{id}],P}$ multiplet

In terms of the original $S_3$ symmetry, this multiplet consists of an untwisted local operator $\mathcal{O}^P$ having a non-trivial charge under $b$ and trivial charge under $a$. Thus this operator does not talk at all to the $\mathbb{Z}_3$ subgroup of $S_3$ symmetry. After gauging $\mathbb{Z}_3$, under the $\widehat{S}_3$ symmetry it has exactly the same properties as above.

Now gauge the residual $\mathbb{Z}_2$ symmetry generated by $b$. Since $\mathcal{O}^P$ is charged under $b$, it goes into the twisted sector for the dual $\mathbb{Z}_2$ symmetry generated by P

$$
\text{P} \quad\longrightarrow\quad \bullet\atop{\mathcal{O}^P}
\tag{C.10}
$$

Since before gauging it was not in the twisted sector for $b$, it is uncharged under P

$$
\text{P} \longrightarrow \underset{\mathcal{O}^P}{\bullet} \Big|_{\text{P}} \quad = \quad \text{P} \underset{\mathcal{O}^P}{\vdash\!\!\!\!\bullet} \Big|_{\text{P}}
\tag{C.11}
$$

Since it was uncharged under $\widehat{a}$ and $\widehat{a}^2$, it is uncharged under E. This fact descends to a relation of the form

$$
\text{P} \longrightarrow \underset{\mathcal{O}^P}{\bullet} \Big|_{\text{E}} \quad = \quad - \quad \text{P} \underset{\mathcal{O}^P}{\vdash\!\!\!\!\bullet} \Big|_{\text{E}}
\tag{C.12}
$$

where the sign on the RHS arises because we choose to define the quadrivalent junction between E and P in terms of left-handed and right-handed trivalent junctions as in (241), and according to (C.4) and (C.5) we have a relative sign between the two trivalent junctions.

#### C.2.2   $Q_{[a],1}$ multiplet

In terms of the original $S_3$ symmetry, this multiplet consists of an $a$-twisted local operator $\mathcal{O}^a_1$ and an $a^2$-twisted local operator $\mathcal{O}^a_2$

$$
a \quad\longleftarrow\quad \underset{\mathcal{O}^a_1}{\bullet} \qquad\qquad a^2 \quad\longleftarrow\quad \underset{\mathcal{O}^a_2}{\bullet}
\tag{C.13}
$$

such that they are both uncharged under $\mathbb{Z}_3$ subgroup, but transform under $b$ as

$$
a \quad\longleftarrow\quad \underset{\mathcal{O}^a_1}{\bullet} \Big|_{b} \quad = \quad a \longleftarrow \overset{b}{\underset{b}{\vdash}}\!\!\!\underset{\mathcal{O}^a_2}{\overset{a^2}{\longleftarrow}\bullet}
\tag{C.14}
$$

After gauging $\mathbb{Z}_3$, the fact that they are uncharged under $\mathbb{Z}_3$ means that they are not in the twisted sector for under $\widehat{\mathbb{Z}}_3$, and hence become untwisted sector operators. Moreover, since they are in twisted sector for $\mathbb{Z}_3$, they carry non-trivial charges under $\widehat{\mathbb{Z}}_3$

$$
\bullet\;\;\mathcal{O}_1^a \;\Big|_{\widehat{a}} = \omega \;\Big|_{\widehat{a}}\;\bullet\;\mathcal{O}_1^a \;,\qquad \bullet\;\mathcal{O}_2^a\;\Big|_{\widehat{a}} = \omega^2 \;\Big|_{\widehat{a}}\;\bullet\;\mathcal{O}_2^a \tag{C.15}
$$

The two operators are still interchanged by the action of $b \in \widehat{S}_3$.

Now let us gauge $b$ to pass to $\mathsf{Rep}(S_3)$ symmetry. The operator

$$
\mathcal{O}_+^a := \mathcal{O}_1^a + \mathcal{O}_2^a , \tag{C.16}
$$

is gauge invariant and hence survives as an untwisted local operator. On the other hand, the operator

$$
\mathcal{O}_-^a := \mathcal{O}_1^a - \mathcal{O}_2^a , \tag{C.17}
$$

carries non-trivial gauge charge, and hence descends to the twisted sector for P. In total, the $\mathsf{Rep}(S_3)$ multiplet of generalized charge $\boldsymbol{Q}_{[a],1}$ comprises of two operators

$$
\bullet\;\mathcal{O}_+^a \;,\qquad \mathsf{P}\;\text{——}\;\bullet\;\mathcal{O}_-^a \tag{C.18}
$$

Since neither of these two operators was in the twisted sector for $b$, they are uncharged under P

$$
\bullet\;\mathcal{O}_+^a\;\Big|_{\mathsf{P}} = \Big|_{\mathsf{P}}\;\bullet\;\mathcal{O}_+^a\;,\qquad \mathsf{P}\;\text{——}\;\bullet\;\mathcal{O}_-^a\;\Big|_{\mathsf{P}} = \mathsf{P}\;\text{——}\!\!\Big|_{\mathsf{P}}\;\bullet\;\mathcal{O}_-^a \tag{C.19}
$$

Let us now consider the action of E on these operators. First, consider the action on $\mathcal{O}_+^a$. We have

$$
\bullet\;\mathcal{O}_+^a\;\Big|_{\mathsf{E}} \equiv \bullet\;\mathcal{O}_1^a + \mathcal{O}_2^a\;\Big|_{\widehat{a}} + \bullet\;\mathcal{O}_1^a + \mathcal{O}_2^a\;\Big|_{\widehat{a}^2}
$$

$$
= \Big|_{\widehat{a}}\;\bullet\;\omega\mathcal{O}_1^a + \omega^2\mathcal{O}_2^a + \Big|_{\widehat{a}^2}\;\bullet\;\omega^2\mathcal{O}_1^a + \omega\mathcal{O}_2^a \tag{C.20}
$$

which we can express in terms of

$$
\left.\begin{array}{c} \bullet \\ \mathcal{O}^a_+ \end{array}\right|_E \equiv \left.\uparrow \begin{array}{c} \bullet \\ \mathcal{O}^a_1 + \mathcal{O}^a_2 \end{array}\right|_{\widehat{a}} + \left.\uparrow \begin{array}{c} \bullet \\ \mathcal{O}^a_1 + \mathcal{O}^a_2 \end{array}\right|_{\widehat{a}^2}
$$

(C.21)

$$
\left.\begin{array}{c} P \\ \bullet \\ \mathcal{O}^a_- \end{array}\right|_E \equiv \left.\uparrow \begin{array}{c} \bullet \\ \mathcal{O}^a_1 - \mathcal{O}^a_2 \end{array}\right|_{\widehat{a}} - \left.\uparrow \begin{array}{c} \bullet \\ \mathcal{O}^a_1 - \mathcal{O}^a_2 \end{array}\right|_{\widehat{a}^2}
$$

as

$$
\left.\begin{array}{c} \bullet \\ \mathcal{O}^a_+ \end{array}\right|_E = -\frac{1}{2} \left.\begin{array}{c} \bullet \\ \mathcal{O}^a_+ \end{array}\right|_E + \left(\omega + \frac{1}{2}\right) \left.\begin{array}{c} P \\ \bullet \\ \mathcal{O}^a_- \end{array}\right|_E
$$

(C.22)

Similarly, we have

$$
\left. P \longrightarrow \begin{array}{c} \bullet \\ \mathcal{O}^a_- \end{array}\right|_E \equiv \begin{array}{c} \bullet \\ \mathcal{O}^a_1 - \mathcal{O}^a_2 \end{array} \left.\uparrow\right|_{\widehat{a}} + \begin{array}{c} \bullet \\ \mathcal{O}^a_1 - \mathcal{O}^a_2 \end{array} \left.\uparrow\right|_{\widehat{a}^2}
$$

$$
= \left.\uparrow \begin{array}{c} \bullet \\ \omega\mathcal{O}^a_1 - \omega^2\mathcal{O}^a_2 \end{array}\right|_{\widehat{a}} + \left.\uparrow \begin{array}{c} \bullet \\ \omega^2\mathcal{O}^a_1 - \omega\mathcal{O}^a_2 \end{array}\right|_{\widehat{a}^2}
$$

(C.23)

which can be expressed in terms of

$$
P \left.\longrightarrow\begin{array}{c} E \\ \bullet \\ \mathcal{O}^a_+ \end{array}\right|_E \equiv \left.\uparrow \begin{array}{c} \bullet \\ \mathcal{O}^a_1 + \mathcal{O}^a_2 \end{array}\right|_{\widehat{a}^2} - \left.\uparrow \begin{array}{c} \bullet \\ \mathcal{O}^a_1 + \mathcal{O}^a_2 \end{array}\right|_{\widehat{a}}
$$

(C.24)

$$
P \left.\longrightarrow\begin{array}{c} E \\ P \\ \bullet \\ \mathcal{O}^a_- \end{array}\right|_E \equiv - \left.\uparrow \begin{array}{c} \bullet \\ \mathcal{O}^a_1 - \mathcal{O}^a_2 \end{array}\right|_{\widehat{a}} - \left.\uparrow \begin{array}{c} \bullet \\ \mathcal{O}^a_1 - \mathcal{O}^a_2 \end{array}\right|_{\widehat{a}^2}
$$

as

$$P \quad \xrightarrow{\quad\bullet\quad} \Big| \quad = \quad -\big(\omega + \tfrac{1}{2}\big) \quad P \quad \xrightarrow{\quad}\Big|_{\bullet} \quad + \quad \tfrac{1}{2} \quad P \quad \xrightarrow{\quad}\Big|_{\bullet} \qquad (C.25)$$

with $\mathcal{O}^a_-$ (left), $\mathcal{O}^a_+$ (middle), $\mathcal{O}^a_-$ (right) labelled over $E$.

### C.2.3 $\quad Q_{[\mathrm{id}],E}$ multiplet

In terms of the original $S_3$ symmetry, this multiplet consists of two untwisted sector local operators $\mathcal{O}^E_1$ and $\mathcal{O}^E_2$ which are charged under $\mathbb{Z}_3$ subgroup as

$$\bullet\ \mathcal{O}^E_1 \Big|_a \ = \ \omega \ \Big|_a \bullet\ \mathcal{O}^E_1 \ , \qquad \bullet\ \mathcal{O}^E_2 \Big|_a \ = \ \omega^2 \ \Big|_a \bullet\ \mathcal{O}^E_2 \qquad (C.26)$$

and $b$ exchanges $\mathcal{O}^E_1$ and $\mathcal{O}^E_2$. After gauging $\mathbb{Z}_3$, these operators descend into twisted sector for $\widehat{\mathbb{Z}}_3$

$$\widehat{a} \quad \xleftarrow{\quad\bullet\quad}_{\mathcal{O}^E_1} \ , \qquad \widehat{a}^2 \quad \xleftarrow{\quad\bullet\quad}_{\mathcal{O}^E_2} \qquad (C.27)$$

These operators are uncharged under $\widehat{\mathbb{Z}}_3$ but are still exchanged by $b$.

Now let us gauge the residual $\mathbb{Z}_2$ symmetry. The combination

$$\mathcal{O}^E_+ = \mathcal{O}^E_1 + \mathcal{O}^E_2 \,, \qquad (C.28)$$

is gauge invariant, hence it descends to a local operator of the form

$$E \quad \xrightarrow{\qquad\bullet}_{\mathcal{O}^E_+} \qquad (C.29)$$

On the other hand, the combination

$$\mathcal{O}^E_- = \mathcal{O}^E_1 - \mathcal{O}^E_2 \,, \qquad (C.30)$$

has non-trivial gauge charge, hence it descends to a local operator of the form

$$E \quad \xrightarrow{\quad\bullet\quad}_{\mathcal{O}^E_-} \quad P \qquad (C.31)$$

Since these were not in twisted sector for $b$, they are uncharged under P

$$E \quad \xrightarrow{\quad\bullet}_{\mathcal{O}^E_+} \Big|_P \quad = \quad - \quad E \quad \xrightarrow{\quad}\Big|_{\bullet\ \mathcal{O}^E_+}{}_P$$

$$E \quad \xrightarrow{\quad\bullet\quad}_{\mathcal{O}^E_-}\Big|\ P \quad = \quad - \quad E \quad \xrightarrow{\quad}\Big|_{\bullet\ \mathcal{O}^E_-}\ P \qquad (C.32)$$

Again we remind the reader that the sign on the RHS arises due to the sign difference between (C.4) and (C.5), and not because these operators are charged under P.

Let us now determine the action of E on these operators. We have

$$ \tag{C.33} $$

The last two terms can be recognized as being

$$ \tag{C.34} $$

while the first two can be recognized as the difference

$$ \tag{C.35} $$

In total, we derive the action of E on $\mathcal{O}_+^E$ shown in (253). Similarly, one can derive (254).

### C.2.4 $Q_{[b],+}$ multiplet

In terms of the original $S_3$ symmetry, this multiplet comprises of three local operators $\mathcal{O}_b^{S_3}$, $\mathcal{O}_{ab}^{S_3}$ and $\mathcal{O}_{a^2b}^{S_3}$ lying in twisted sectors for $b$, $ab$ and $a^2b$ respectively. The action of an element $x \in S_3$ is

$$ x: \quad \mathcal{O}_y^{S_3} \to \mathcal{O}_{xyx^{-1}}^{S_3}, \qquad y \in \{b, ab, a^2b\}. \tag{C.36} $$

After gauging the $\mathbb{Z}_3$ subgroup, the multiplet has exactly the same properties. It comprises of three local operators

$$ \mathcal{O}_{\widehat{y}}^{\widehat{S}_3}, \qquad \widehat{y} \in \{b, \widehat{a}b, \widehat{a}^2b\}, \tag{C.37} $$

which are in the twisted sector for $\widehat{y}$. The action of an element $\widehat{x} \in \widehat{S}_3$ is

$$\widehat{x}: \quad \mathcal{O}_{\widehat{y}}^{\widehat{S}_3} \rightarrow \mathcal{O}_{\widehat{x}\widehat{y}\widehat{x}^{-1}}^{\widehat{S}_3}. \tag{C.38}$$

Let us now gauge the residual $\mathbb{Z}_2$ symmetry, which exchanges $\mathcal{O}_{\widehat{a}b}^{\widehat{S}_3}$ and $\mathcal{O}_{\widehat{a}^2 b}^{\widehat{S}_3}$, while leaving $\mathcal{O}_b^{\widehat{S}_3}$ invariant. The operator $\mathcal{O}_b^{\widehat{S}_3}$ descends to an untwisted sector local operator $\mathcal{O}^b$ which is charged under P, as shown in (256). The combination

$$\mathcal{O}_+^b := \mathcal{O}_{\widehat{a}b}^{\widehat{S}_3} + \mathcal{O}_{\widehat{a}^2 b}^{\widehat{S}_3}, \tag{C.39}$$

is gauge invariant, hence descends to an operator in the twisted sector for E. This operator is charged under P because of the presence of the factor of $b$ in the twisted sectors $ab$ and $a^2 b$ of $\mathcal{O}_{\widehat{a}b}^{\widehat{S}_3}$ and $\mathcal{O}_{\widehat{a}^2 b}^{\widehat{S}_3}$. This is reflected in the fact that there is no sign on the RHS of the second equation in (256). The sign coming from the fact that the operator is charged is canceled by the relative sign coming from (C.4) and (C.5). Finally, the combination

$$\mathcal{O}_-^b := \mathcal{O}_{\widehat{a}b}^{\widehat{S}_3} - \mathcal{O}_{\widehat{a}^2 b}^{\widehat{S}_3}, \tag{C.40}$$

has non-trivial gauge charge, hence descends to an operator sitting between lines E and P. For a similar reason as for $\mathcal{O}_+^b$, the operator $\mathcal{O}_-^b$ is also non-trivially charged under P, which is reflected in the positive sign on the RHS of the third equation in (256). The action of E can be derived in a similar way as for the $Q_{[\mathrm{id}],E}$ multiplet discussed above.

## C.3 $\mathrm{Rep}(S_3)$ phases from $S_3$ phases

In this subsection, we discuss how various important properties of $\mathrm{Rep}(S_3)$-symmetric gapped phases can be obtained by gauging the $S_3$ symmetry of $S_3$-symmetric gapped phases. From the SymTFT point of view, this gauging is localized on the symmetry boundary, while leaving the bulk SymTFT and the physical boundary invariant. The gauging modifies the symmetry boundary as

$$\mathfrak{B}^{\mathrm{sym}} = \mathcal{A}_{\mathrm{Dir}} \longrightarrow \mathcal{A}_{\mathrm{Neu}}. \tag{C.41}$$

**$S_3$ SSB phase to trivial $\mathrm{Rep}(S_3)$ phase.**  Let us first choose

$$\mathfrak{B}^{\mathrm{phys}} = \mathcal{A}_{\mathrm{Dir}}. \tag{C.42}$$

In terms of the original $S_3$ symmetry, it gives rise to the $S_3$ SSB phase discussed in section 4.6.1, which has 6 vacua permuted by the $S_3$ symmetry. Under the $\mathbb{Z}_3$ subgroup, the vacua form two orbits of 3 vacua each. These two groups are interchanged by $b$.

After gauging $\mathbb{Z}_3$, the vacua related by $\mathbb{Z}_3$ action get identified, and the resulting theory has 2 vacua that are interchanged by $b$. In other words, we obtain the $\mathbb{Z}_2$ SSB phase for $\widehat{S}_3$ symmetry.

Now let us gauge the residual $\mathbb{Z}_2$ symmetry. The result is that the remaining 2 vacua also get identified, and we obtain a phase for $\mathrm{Rep}(S_3)$ which has a single vacuum. This is the trivial $\mathrm{Rep}(S_3)$ phase discussed in section 5.3.1. Note that indeed the physical boundaries in both sections 4.6.1 and 5.3.1 are the same as C.42.

**$\mathbb{Z}_3 \subset S_3$ SSB phase to $\mathbb{Z}_2 \subset \mathrm{Rep}(S_3)$ SSB phase.**  Now let us choose

$$\mathfrak{B}^{\mathrm{phys}} = \mathcal{A}_{\mathrm{Neu}(\mathbb{Z}_2)}. \tag{C.43}$$

In terms of the original $S_3$ symmetry, it gives rise to the $\mathbb{Z}_3$ SSB phase discussed in section 4.6.2, which has 3 vacua permuted by the $\mathbb{Z}_3 \subset S_3$ symmetry, while $b$ permutes 2 vacua $v_1, v_2$ leaving 1 vacuum $v_0$ invariant. Note that the $D_1^{(b)}$ line can end along $v_0$, giving rise to a local operator $\mathcal{O}_b$ in $b$-twisted sector (221).

After gauging $\mathbb{Z}_3$, all the 3 vacua get identified. Thus we obtain the trivial phase for $\widehat{S}_3$ symmetry. The operator $\mathcal{O}_b$ is still in the $b$-twisted sector and can now be identified with the identity local operator regarded as the end of line $D_1^{(b)}$.

Let us now gauge the residual $\mathbb{Z}_2$ symmetry. The $b$-twisted sector operator $\mathcal{O}_b$ descends to a new untwisted sector local operator, implying that we have a total of 2 vacua in the resulting phase. Moreover, $\mathcal{O}_b$ has to be charged under P implying that we have $\mathbb{Z}_2$ SSB phase for $\text{Rep}(S_3)$ symmetry discussed in section 5.3.2. Note that indeed the physical boundaries in both sections 4.6.2 and 5.3.2 are the same as C.43.

**$\mathbb{Z}_2 \subset S_3$ SSB phase to $\text{Rep}(S_3)/\mathbb{Z}_2$ SSB phase.**    Now let us choose

$$\mathfrak{B}^{\text{phys}} = \mathcal{A}_{\text{Neu}(\mathbb{Z}_3)}. \tag{C.44}$$

In terms of the original $S_3$ symmetry, it gives rise to the $\mathbb{Z}_2$ SSB phase discussed in section 4.6.3, which has 2 vacua $v_0, v_1$ permuted by $b$, while $\mathbb{Z}_3 \subset S_3$ leaves both vacua invariant. Note that the $D_1^{(a)}$ line can end along $v_0$ and $v_1$, giving rise to two linearly independent local operators $\mathcal{O}_1^a, \mathcal{O}_2^a$ in $a$-twisted sector. Similarly, we have two linearly independent local operators $\mathcal{O}_1^{a^2}, \mathcal{O}_2^{a^2}$ in $a^2$-twisted sector.

After gauging $\mathbb{Z}_3$ symmetry, the above 4 twisted sector operators become new untwisted sector operators, and hence we obtain a total of 6 vacua. The new local operators are charged under dual $\widehat{\mathbb{Z}}_3$ symmetry as they are in the twisted sector before gauging. Moreover, $b$ also acts on these operators as

$$\mathcal{O}_1^a \leftrightarrow \mathcal{O}_2^{a^2}, \qquad \mathcal{O}_2^a \leftrightarrow \mathcal{O}_1^{a^2}, \tag{C.45}$$

as it interchanges $v_0$ and $v_1$, and simultaneously interchanges the two twisted sectors. In total, one can now easily see that the resulting phase is $\widehat{S}_3$ SSB phase.

Let us now gauge the residual $\mathbb{Z}_2$ symmetry, whose action on vacua of $\widehat{S}_3$ SSB phase has 3 orbits. Hence the resulting phase has 3 vacua, such that P is spontaneously unbroken in all 3 vacua. This can be identified with the $\text{Rep}(S_3)/\mathbb{Z}_2$ SSB phase discussed in section 5.3.3. Note that indeed the physical boundaries in both sections 4.6.3 and 5.3.3 are the same as C.44.

**Trivial $S_3$ phase to $\text{Rep}(S_3)$ SSB phase.**    Finally, let us choose

$$\mathfrak{B}^{\text{phys}} = \mathcal{A}_{\text{Neu}}. \tag{C.46}$$

In terms of the original $S_3$ symmetry, it gives rise to the trivial phase discussed in section 4.6.4, which has a single vacuum left invariant by all of $S_3$ symmetry. We have an operator in twisted sector for both $a$ and $a^2$, obtained by ending $D_1^{(a)}$ and $D_1^{(a^2)}$ along the identity operator.

After gauging $\mathbb{Z}_3$, these twisted sector operators descend into the untwisted sector. Thus the resulting phase has 3 vacua, and it can be recognized as the $\mathbb{Z}_3$ SSB phase for $\widehat{S}_3$ symmetry. The $b$ symmetry exchanges two vacua $v_1, v_2$, while leaving invariant a vacuum $v_0$. Due to this reason, we have a $b$-twisted sector operator $\mathcal{O}^b$ as $D_1^{(b)}$ can end along $v_0$.

Now let us gauge the residual $\mathbb{Z}_2$. The two vacua $v_1$ and $v_2$ get combined into a single vacuum

$$v_{12} \equiv v_1 + v_2. \tag{C.47}$$

The vacuum $v_0$ survives as an untwisted sector operator, and we obtain a new untwisted sector operator $\mathcal{O}^b$. Since we have

$$(\mathcal{O}^b)^2 = v_0, \tag{C.48}$$

the two vacua other than $v_{12}$ are

$$v_+ = \frac{v_0 + \mathcal{O}^b}{2}, \qquad v_- = \frac{v_0 - \mathcal{O}^b}{2}. \tag{C.49}$$

In total, we obtain 3 vacua $v_{12}$ and $v_\pm$, out of which $v_{12}$ is left invariant by P but $v_+$ and $v_-$ are interchanged because $\mathcal{O}^b$ carries a non-trivial charge under P. We thus obtain the Rep($S_3$) SSB phase discussed in section 5.3.4. Note that indeed the physical boundaries in both sections 4.6.4 and 5.3.4 are the same as C.46.

One can also observe the presence of relative Euler terms using the gauging procedure. Let us label the unit line operators transitioning between vacua $v_0$, $v_1$ and $v_2$ of the $\widehat{S}_3$-symmetric phase as

$$\mathbf{1}_{i,j}, \qquad i,j \in \{0,1,2\}, \tag{C.50}$$

and the unit line operators transitioning between vacua $v_{12}$, $v_+$ and $v_-$ of the Rep($S_3$)-symmetric phase as

$$\mathbf{1}_{i,j}, \qquad i,j \in \{12,+,-\}. \tag{C.51}$$

Then the $\mathbb{Z}_2$ residual gauging implies that these operators are related as

$$\begin{aligned}
\mathbf{1}_{12,12} &\equiv \mathbf{1}_{1,1} \oplus \mathbf{1}_{2,2}, \\
\mathbf{1}_{+,12} \oplus \mathbf{1}_{-,12} &\equiv \mathbf{1}_{0,1} \oplus \mathbf{1}_{0,2}, \\
\mathbf{1}_{+,+} \oplus \mathbf{1}_{-,-} &\equiv \mathbf{1}_{0,0}.
\end{aligned} \tag{C.52}$$

Then the linking action of $\mathbf{1}_{+,12}$ on $v_+$ is

$$\mathbf{1}_{+,12} \cdot v_+ = (\mathbf{1}_{+,12} \oplus \mathbf{1}_{-,12}) \cdot v_+ \equiv (\mathbf{1}_{0,1} \oplus \mathbf{1}_{0,2}) \cdot \left( \frac{v_0 + \mathcal{O}^b}{2} \right) = (\mathbf{1}_{0,1} \oplus \mathbf{1}_{0,2}) \cdot \frac{v_0}{2} = \frac{v_1 + v_2}{2} \equiv \frac{1}{2} v_{12}. \tag{C.53}$$

That is, we have a relative Euler term of $1/2$ in passing from $v_+$ to $v_{12}$. Similarly, we have a relative Euler term of $1/2$ in passing from $v_-$ to $v_{12}$. This reproduces the Euler terms found in section 5.3.4.

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
