# Peer review of "Gapped Phases with Non-Invertible Symmetries: (1+1)d"

_SciPost Physics, doi:SciPost Phys. 18, 032 (2025)_

## Round 2 · Referee Report · Anonymous (Referee 1) · 2024-3-28

Strengths

1- The paper contains a detailed discussion of a novel aspect that spontaneous breaking of non-invertible symmetries can lead to physically distinguishable vacua. 2- The paper is very comprehensive and rather detailed in its discussion of examples, starting with relatively simple and well-known ones and then turning its attention to more complicated and also new ones that have not been discussed in the literature so far 3- At many places, the paper uses different perspectives and approaches to reach the same conclusions. This is instructive and provides additional confidence in the validity of the results.

Weaknesses

1- Referring to the acceptance criteria: The paper does NOT contain a clear conclusion summarizing the results and offering perspectives for future work. One could argue though that the introduction itself serves as a suitable summary. 2- Beyond the introduction the authors are very sparse with references. For readers who are not entirely familiar with previous developments in this very dynamic area it is frequently very hard to judge whether a statement that is made is (i) well-established knowledge (not worth citing a reference [according to them]), (ii) a working hypothesis, or (iii) something that is going to be established later by the authors. One of many examples: "For invertible symmetries, i.e. when the symmetry is described by a finite (0-form) symmetry group G, the Euler terms can all still be tuned to zero" below equation (2.3), stated without explanation or reference. 3- Referring to point 2: What the authors regard as well-established knowledge might differ from what the reader regards as well-established knowledge, either because the reader lacks the detailed experience in the area or because they have different attitudes concerning the mathematical level of rigor of the statement. From this perspective it would be appropriate to be much more detailed about references that the authors have in mind in connection with certain statements and also be clearer about what is an assumption or working hypothesis and what will be shown subsequently. 4- Referring to point 3: The authors should imagine someone reading their paper in many years time, without having all the relevant background knowledge that active researchers in the field might currently have. In my opinion it should still be valuable then and be so as a stand-alone paper. For instance, the classification of fiber functors in (3.22) very clearly requires an explicit reference to where this (specific) result was obtained. Similarly, the statement "The irreducible topological boundary conditions of the SymTFT Z(S) are captured (modulo Euler terms) by Lagrangian algebras in the Drinfeld Center Z(S)" requires a reference. These are just two of many examples.

Report

The paper uses the recent SymTFT approach to classify gapped phases of matter in 1+1D and provides a detailed discussion of the relevant vacua and generalized order parameters, from a general perspective and in many concrete examples. One novel aspect is the realization that the spontaneous breakdown of non-invertible symmetries may lead to the existence of physically distinguishable vacua. Also the discussion of some examples seems to be new, especially of the symmetries $TY(Z_N)$.

The paper provides a very clear introduction into the topic and a nice summary of the general strategy and the results obtained. I also regard it as very valuable to see how the general theory of "generalized charges" that was obtained in a different paper applies concretely in 1+1D, a setting many people from various different scientific communities have experience and interest in. Even though the results for some of the examples are not truly new I personally regard it as important to see how they can be obtained within the powerful and very general framework of SymTFT and generalized charges.

The methodology and reasoning is very sound and explained in appropriate detail even though the referee was not able to validate all individual statements and computations in view of the available time. Concrete strengths and weaknesses have been highlighted before.

In my opinion, the paper opens a new pathway in an existing/emerging direction, with clear potential for multipronged follow-up work, thus satisfying one of the expectations for publications in SciPost Physics. It also satisfies acceptance criteria 1-5, with some reservations concerning the appropriate referencing to the literature (see above). Criterion 6 is clearly violated as was pointed out before.

In view of the well-written summary of the strategy and the results in the introduction I would still recommend the manuscript for publication after a minor revision.

Requested changes

1- Please address points 2-4 mentioned in the weaknesses above by being more specific about the origin of specific statements throughout the bulk of the text, including the citation of appropriate literature 2- Explicit examples of statements that would benefit from clarification or references are (and this list should not be viewed as exhaustive) * The fiber functor classification around (3.22) * Why are the two statements around (3.40) equivalent? * "one may obtain physically indistinguishable vacua even if non-invertible symmetries are spontaneously broken" just before Section 3.5 (this will be derived later and that should be stated explicitly) * below (4.12), what is "trivial" about this G-symmetry? * "A general fact of every 3d SymTFT Z(S) is that any arbitrary topological boundary condition of Z(S) can be obtained by gauging the S symmetry of BsymS . " * The statements below (4.19) * The statements in the second paragraph of Section 4.2 * The statements around (4.20) * Equation (4.29) 3- There is a long history of "string operators" and "string order", specifically in 1+1D and in connection with the notion of hidden symmetry breaking, that is currently completely absent from the references. The relevant original paper (which to my knowledge coined the term) appears to be https://doi.org/10.1103/PhysRevB.40.4709 and is, in my opinion, the minimum which should be cited 4- In (3.36) and the subsequent discussion there is an index $i$ in the coefficients $n_i^a$ that I found rather confusing (as it does not appear on the left hand side of the equation) and whose presence should be motivated explicitly 5- There is an $\omega$ appearing in (5.16) that appears to refer back to Section 4 while in the meantime $\omega$ was used for the anomaly of $Vec_G^\omega$. I would suggest to state again explicitly what $\omega$ is in this formula. 6- Papers that I would recommend citing at appropriate places: * https://doi.org/10.1103/PhysRevB.79.045316 7- There are a few typos that should be corrected * Below eq (3.32) there is no "following" rectangle identity but rather a figure that should be referenced * identity $\to$ identify above (5.41) and (5.44) * Below (7.17) (and maybe more generally) the footnote should be placed after the punctuation * Footnote 21 appears in an incomplete sentence and it lacks the desired reference * Appendix C would benefit from checking again for grammar, word twists, etc. 8- Probably quite a few papers cited in the references have been published in the meantime and should be updated.

---

## Round 2 · Referee Report · Anonymous (Referee 2) · 2024-6-12

Report

This paper provides a systematic study of 1+1d gapped phase with non-invertible symmetries. It provides sufficient details that expands a short companion paper on generalized Landau paradigm by the same set of authors. The results are novel, interesting, and the presentation is well-organized. The referee likes the paper a lot, and would strongly recommend the paper to be accepted by Scipost physics. 

That said, there are a few comments i wish the authors can implement before publication. Note that all the comments below are based on the latest arXiv version v3 (rather than the submitted version v2).

  1. I agree with all the comments from the first referee, and they should be implemented.

  2. To the referee’s knowledge, the SPT phases and phase transitions between SPT phases (even for invertible symmetries) lay outside the generalized Landau paradigm. See for instance the discussion in the middle of page 7 of https://arxiv.org/pdf/2312.16898. However, in the present work, the generalized charges involve extended operators from the twist sector, hence the SPT phase can be understood as condensation of string order parameters, and can be incorporated into the generalized Landau paradigm in the sense of the current paper. I think this should be emphasized somewhere in the paper.

  3. The authors emphasized that vacua in the presence of SSB of non-invertible symmetry are “physically distinguishable”, while whose of invertible symmetry are not. I do not fully agree with the later statement. Consider a mixed anomaly between G and H. For suitable mixed anomaly, the vacua from SSB of G carry different H-SPTs, and different SPTs are clearly distinguishable—for example different SPTs have different twisted sector charges, and also have different edge modes in the presence of boundaries. I think by physical distinguishability, the authors mean the relative Euler terms, hence it is somewhat a more restricted sense of physical distinguishability.

  4. I think it is worth to mention that only “simple” symmetry boundary condition A_S of the SymTFT is considered, which means the paper does not consider multi-fusion category. (However mult-fusion category appeared below eq 3.9, is the assumption relaxed here, and only here? )

  5. Related to 4, I think it is also worth to mention that only “simple” physical topological boundary condition A_{phys} of the SymTFT is considered, which means the paper considers the “maximal” symmetry of the TQFT. The reason I brought this up is because given a TQFT, the choice of SymTFT is not unique. In particular, we can consider the SymTFT of a sub-symmetry. In this paper, I think the 2d S-symmetric TQFT classified by the SymTFT has S as its maximal 0-form symmetry. Relaxing the simpleness of the physical boundary clearly makes the symmetry non-maximal. For example, one can consider a topological domain wall between the two 2d TQFTs associated with each simple TFT, and domain wall is not associated with any object in S. Moreover, simple physical boundary condition is also related to only one ambiguous \lambda in eq 3.12. If the physical boundary condition is non-simple, there can be multiple \lambda’s unfixed.

  6. The simpleness of the topological boundary conditions are mentioned on page 27, but I think it can be mentioned much earlier (say in the intrduction).

  7. On page 11, it is said that the TY($Z_N$) are non-group-theoretical. This is not precise—TY($Z_N$) is non-group theoretical only when N is not a prefect square, as discussed in detail in ref 64.

  8. What is the precise relation between eq 3.32 and eq 3.34? Is the latter a special case of the former? Should’t the representation of tube algebra completely characterize the generalized charge, or not?

  9. In footnote 16, it is said that A_S and A’_S have the same localized symmetry S. Can the authors give an example? Moreover, what does “localized symmetry” mean?

Recommendation

Ask for minor revision

---

## Round 4 · Referee Report · Anonymous (Referee 2) · 2024-8-19

Report

The authors have addressed my comments, and I recommend to publish it as an article in Scipost Physics.

Recommendation

Publish (surpasses expectations and criteria for this Journal; among top 10%)

---

## Round 4 · Referee Report · Anonymous (Referee 1) · 2024-9-26

Report

The authors mostly improved the manuscript along the lines suggested. While I still think that generally more effort should have been put into acknowledging relevant pre-2020 contributions to the field and also making references more specific, i.e. tie them to the concrete discussion of examples and results that are reviewed instead of citing them in bulk in the introductory section, I am willing to leave it with requesting the inclusion of additional references to the discussion of string order parameters. This was already strongly suggested in my previous report but to assist the authors this is now made more concrete below.

Requested changes

At the beginning of Section 4.2 there seems to be substantial scope for including additional references to the use of string order parameters to describe SPT phases that are protected by invertible symmetries. Historically, and in terms of generality the most relevant works in the context of the present paper are probably * https://doi.org/10.1103/PhysRevB.3.3918 (disorder variable in the Ising model) * https://doi.org/10.1007/BF02097239 (Z2xZ2 hidden symmetry breaking - original case underlying the Kennedy-Tasaki transformation) * https://doi.org/10.1088/0953-8984/4/36/019 (Z2xZ2 hidden symmetry breaking - more general case) * https://doi.org/10.1103/PhysRevLett.100.167202 (general discussion about the relation between string order and symmetries) * https://doi.org/10.1103/PhysRevLett.109.050402 (string-like order parameters for SPT phases) * https://doi.org/10.1103/PhysRevB.86.125441 (generalized string order parameters for SPT phases) * https://doi.org/10.1103/PhysRevB.88.085114 (hidden symmetry breaking picture for general finite abelian groups) * https://doi.org/10.1103/PhysRevB.88.125115 (similar but with an emphasis on discussing orbits of phases under Kennedy-Tasaki-like dualities)

Hidden symmetry breaking refers to the characterization of SPT phases in terms of string-order parameters that can be mapped to local order parameters using a non-local duality. Especially, the last two papers include a detailed discussion of the interplay between both local and non-local order parameters in a relatively general setting and how they transform into each other under non-local dualities.

All of this appears highly relevant in the context of the present paper and should, in my opinion, be cited in the intro to section 4.2 and in the appropriate examples (like Z2 or Z2xZ2) to give prospective readers a comprehensive and accurate picture of previous developments.

Concerning https://doi.org/10.1103/PhysRevB.79.045316, my impression is that this is one of the first papers discussing the condensation of anyons on the boundary of a topological phase, predating the discussion in [128] in terms of Lagrangian algebras by 8 years. A place where it could be cited is the beginning of the second paragraph in appendix B.

Recommendation

Ask for minor revision

---

## Round 4 · Author Response

We thank both referees for their detailed and insightful reports.
We list below the changes made according to each referee's suggestions.

Report 1:

We thank the referee for the extremely detailed reading of the paper and all the comments.
We have addressed most of the report's requests when they were clearly stated. Regarding reference
https://doi.org/10.1103/PhysRevB.79.045316 we were not sure where the referee wants us to cite this.

Report 2:

We thank the referee for the detailed report and comments.

Regarding the point 2: We have emphasized that the SPT phases are strictly speaking outside of the original Landau paradigm.

Regarding point 3: that is a very good point and we have made our statements clearer to distinguish this from the G-phases that the referee pointed out.

Regarding point 4: In subsequent papers we discuss reducible boundary conditions for topological orders which are important for gapless phases, and a detailed discussion can be found there 2403.00905. In the present paper we indeed focus on simple boundary conditions. As for eqn (3.9), the multi-fusion category being discussed there is the category formed by lines of a 2d TFT that contains multiple vacua, and *not* the category formed by lines of an irreducible 2d topological boundary condition of a 3d TFT.

Regarding point 5: The irreducibility of physical boundaries is emphasized in bold on page 27.

We have addressed all other questions in the text at the referred locations.

---

## Round 5 · Referee Report · Anonymous (Referee 1) · 2024-11-2

Report

All requests have been addressed, I am happy for this article to be published.

Recommendation

Publish (easily meets expectations and criteria for this Journal; among top 50%)

---

## Round 5 · Referee Report · Anonymous (Referee 2) · 2024-11-27

Report

The authors have addressed all the comments and I would recommend it for publication.

Recommendation

Publish (surpasses expectations and criteria for this Journal; among top 10%)

---

## Round 5 · Author Response

We thank both referees for their reports.
We addressed the requests of report 2 by including all the mentioned references.

---

## Editorial Decision

published